# REINFORCEMENT LEARNING FOR FINITE SPACE MEAN-FIELD TYPE GAMES

## ABSTRACT

Mean field type games (MFTGs) describe Nash equilibria between large coalitions: each coalition consists of a continuum of cooperative agents who maximize the average reward of their coalition while interacting non-cooperatively with a finite number of other coalitions. Although the theory has been extensively developed, we are still lacking efficient and scalable computational methods. Here, we develop reinforcement learning methods for such games in a finite space setting with general dynamics and reward functions. We start by proving that MFTG solution yields approximate Nash equilibria in finite-size coalition games. We then propose two algorithms. The first is based on quantization of mean-field spaces and Nash Q-learning. We provide convergence and stability analysis. We then propose a deep reinforcement learning algorithm, which can scale to larger spaces. Numerical experiments in 5 environments with mean-field distributions of dimension up to 200 show the scalability and efficiency of the proposed method.

## 1 INTRODUCTION

Game theory has found a large number of applications, from economics and finance to biology and epidemiology. The most common notion of solution is the concept of Nash equilibrium, in which no agent has any incentive to deviate unilaterally (Nash, 1951). At the other end of the spectrum is the concept of social optimum, in which the agents cooperate to maximize a total reward over the population. These notions have been extensively studied for finite-player games, see e.g. (Fudenberg & Tirole, 1991). Computing exactly Nash equilibria in games with a large number of players is known to be a very challenging problem (Daskalakis et al., 2009).

To address this challenge, the concept of mean field games (MFGs) has been introduced in (Lasry & Lions, 2007; Huang et al., 2006), relying on intuitions from statistical physics. The main idea is to consider an infinite population of agents, replacing the finite population with a probability distribution, and to study the interactions between one representative player with this distribution. Under suitable conditions, the solution to an MFG provides an approximate Nash equilibrium for the corresponding finite-player game. While MFGs typically focus on the solution concept of Nash equilibrium, mean field control (MFC) problems focus on the solution concept of social optimum (Bensoussan et al., 2013). The theory of these two types of problems has been extensively developed, in particular using tools from stochastic analysis and partial differential equations, see e.g. (Bensoussan et al., 2013; Gomes & Saúde, 2014; Carmona & Delarue, 2018) for more details.

However, many real-world situations involve agents that are not purely cooperative or purely non-cooperative. In many scenarios, the agents form coalitions: they cooperate with agents of the same group and compete with other agents of other groups. In the limit where the number of agents is infinite while the number of coalitions remains finite, this leads to the concept of mean-field type games (MFTGs) (Tembine, 2017). Various applications have been developed, such as blockchain token economics (Barreiro-Gomez & Tembine, 2019), risk-sensitive control (Tembine, 2015) or more broadly in engineering (Barreiro-Gomez & Tembine, 2021). Similar problems have been studied under the terminology of mean field games among teams (Subramanian et al., 2023) and team-against-team mean field problems (Sanjari et al., 2023; Yüksel & Başar, 2024). The case of zero-sum MFTG has received special interest (Başar & Moon, 2021; Cosso & Pham, 2019; Guan et al., 2024), but the framework of MFTGs also covers general sum games with more than two (mean-field) coalitions. MFTGs are different from MFGs because the agents are cooperative within

coalitions, while MFGs are about purely non-cooperative agents. They are also different from MFC problems, in which the agents are purely cooperative. As a consequence, computational methods and learning algorithms for MFGs and MFC problems cannot be applied to compute Nash equilibria between mean-field coalitions in MFTGs. Last, graphon games (Caines & Huang, 2019) and mixed mean field control games (Angiuli et al., 2023a) correspond to limit scenarios with infinitely many mean-field groups. In such games, each player has a negligible impact on the rest of the population, which is not the case in MFTGs, see (Tembine, 2017), so new methods are required for MFTGs.

Inspired by the recent successes of RL in two-player games such as Go (Silver et al., 2016) and poker (Brown et al., 2020), RL methods have been adapted to solve MFGs and MFC problems, see e.g. (Subramanian & Mahajan, 2019; Guo et al., 2019; Elie et al., 2020; Cui & Koeppl, 2021) and (Gu et al., 2021; Carmona et al., 2023; Angiuli et al., 2023b) respectively, among many other references. We refer to (Laurière et al., 2022) and the references therein for more details. Such methods compute the solutions to mean field problems. A related topic is mean field multi-agent reinforcement learning (MFMARL) (Yang et al., 2018), which studies finite-agent systems and replaces the interactions between agents with the mean of neighboring agents' states and actions. Extensions include situations with multiple types and partial observation (Ganapathi Subramanian et al., 2020; 2021). However, the MFMARL setting differs substantially from MFTGs: (1) it does not take into account a general dependence on the mean field (i.e., the whole population distribution), (2) it aims directly for the finite-agent problem while using a mean-field approximation in an empirical way, and (3) it is not designed to tackle Nash equilibria between coalitions. The works most related to ours applied RL to continuous space linear-quadratic MFTGs by exploiting the specific structure of the equilibrium policy in these games (Carmona et al., 2020; uz Zaman et al., 2024; Zaman et al., 2024). In these settings, policies can be represented exactly with a small number of parameters. In contrast, we focus on finite space MFTGs with general dynamics and reward functions, for which there has been no RL algorithm thus far to the best of our knowledge.

**Main contributions.** Our main contributions are as follows:

1. We prove that solving an MFTG provides an $\epsilon$-Nash equilibrium for a game between finite-size coalitions (Theorem 2.4), which justifies studying MFTGs for finite-player applications.
2. We propose a tabular RL method based on quantization of the mean-field spaces and Nash Q-learning (Hu & Wellman, 2003). We prove the convergence of this algorithm, analyzing the error due to the discretization (Theorem 3.2).
3. We propose a deep RL algorithm based on DDPG (Lillicrap et al., 2016) which does not require quantization and hence is more scalable to problems with a large number of states.
4. We illustrate both methods in 5 environments with distribution in dimension up to 200. Since this paper is the first to propose RL algorithms for (finite space) MFTGs with general dynamics and rewards, there is no standard baseline to compare with. We thus carry out a comparison with two baselines inspired by independent learning.

The rest of the paper is organized as follows. In Section 2, we define the finite-agent problem with coalitions and then its mean-field limit, and establish their connection. We then reformulate the MFTG problem in the language of mean field MDPs. In Section 3, we present an algorithm based on the idea of Nash Q-learning, and we analyze it. In Section 4, we present our deep RL algorithm for MFTG, without discretization of the mean-field spaces. Numerical experiments are provided in Section 5. Section 6 is dedicated to a summary and a discussion. The appendices contain proofs and additional numerical results.

## 2 DEFINITION OF THE MODEL

In this section, we define the finite-population $m$-coalition game and the limiting MFTG with $m$ (central) players. We will use the terminology **agent** for an individual in a coalition and **central player** for the player who chooses the policy to be used by her coalition. We will sometimes write player instead of central player.

### 2.1 FINITE-POPULATION $m$-COALITION GAME

We consider a game between $m$ groups of many agents. Each group is called a **coalition** and behaves cooperatively within itself. Alternatively, we can say that there are $m$ central players, and each of

them chooses the behaviors to be used in their respective coalition. For each $i \in [m]$, let $S^i$ and $A^i$ be respectively the finite state space and the finite action space for the individual agents in coalition $i$. Let $N_i$ denote the number of individual agents in coalition $i$. Let $\Delta(S^i)$ and $\Delta(A^i)$ be the sets of probability distributions on $S^i$ and $A^i$, respectively. Agent $j$ in coalition $i$ has a state $x_t^{ij}$ at time $t$. The state of coalition $i$ is characterized by the empirical distribution $\mu_t^{i,\bar{N}} = \frac{1}{N_i} \sum_{j=1}^{N_i} \delta_{x_t^{ij}} \in \Delta(S^i)$, and the state of the whole population is characterized by the joint empirical distribution: $\mu_t^{\bar{N}} = (\mu_t^{1,\bar{N}}, \ldots, \mu_t^{m,\bar{N}})$. The state of every agent $j \in [N_i]$ in coalition $i$ evolves according to a transition kernel $p^i : S^i \times A^i \times \prod_{i'=1}^m \Delta(S^{i'}) \to \Delta(S^i)$. If the agent takes action $a_t^{ij}$ and the distribution is $\mu_t^{\bar{N}}$, then: $x_{t+1}^{ij} \sim p^i(\cdot|x_t^{ij}, a_t^{ij}, \mu_t^{\bar{N}})$. We assume that the states of all agents in all coalitions are sampled independently. During this transition, the agent obtains a reward $r^i(x_t^{ij}, a_t^{ij}, \mu_t^{\bar{N}})$ given by a function $r^i : S^i \times A^i \times \prod_{i'=1}^m \Delta(S^{i'}) \to \mathbb{R}$. All the agents in coalition $i$ independently pick their actions according to a common policy $\pi^i : S^i \times \Delta(S^1) \times \cdots \times \Delta(S^m) \to \Delta(A^i)$, i.e., $a_t^{ij}$ for all $j \in [N_i]$ are i.i.d. with distribution $\pi^i(\cdot|x_t^{ij}, \mu_t^{\bar{N}})$. Notice that the arguments include the individual state and the distribution of each coalition. We denote by $\Pi^i$ the set of such policies. The average social reward for the central player of population $i$ is defined as: $J^{i,\bar{N}}(\pi^1, \ldots, \pi^m) = \frac{1}{N_i} \sum_{j=1}^{N_i} \mathbb{E}[\sum_{t \geq 0} \gamma^t r_t^{ij}]$, where $\gamma \in [0, 1)$ is a discount factor and the one-step reward at time $t$ is $r_t^{ij} = r_t^i(x_t^{ij}, a_t^{ij}, \mu_t^{\bar{N}})$. We focus on the solution corresponding to a Nash equilibrium between the central players.

**Definition 2.1 (Nash equilibrium for finite-population $m$-coalition type game)** *A policy profile $(\pi_*^1, \ldots, \pi_*^m) \in \Pi^1 \times \cdots \times \Pi^m$ is a **Nash equilibrium** for the above finite-population game if: for all $i \in [m]$, for all $\pi^i \in \Pi^i$, $J^{i,\bar{N}}(\pi^i; \pi_*^{-i}) \leq J^{i,\bar{N}}(\pi_*^i; \pi_*^{-i})$, where $\pi_*^{-i}$ denotes the vector of policies for central players in other coalitions except $i$.*

In a Nash equilibrium, there is no incentive for unilateral deviations at the coalition level. When each $N_i$ goes to infinity, we obtain a game between $m$ central players in which each player controls a population distribution. Such games are referred to as **mean-field type games** (MFTG for short).

## 2.2 Mean-field type game

Informally, as $N_i \to +\infty$, the state $\mu_t^{i,\bar{N}}$ of coalition $i$ has a limiting distribution $\mu_t^i \in \Delta(S^i)$ for each $i \in [m]$, and the state $\mu_t^{\bar{N}}$ of the whole population converges to $\mu_t = (\mu_t^1, \ldots, \mu_t^m) \in \Delta(S^1) \times \cdots \times \Delta(S^m)$. We will refer to the limiting distributions as the **mean-field** distributions. Based on propagation-of-chaos type results, we expect all the agents' states to evolve independently, interacting only through the mean-field distributions. It is thus sufficient to understand the behavior of one representative agent per coalition. A representative agent in mean-field coalition $i$ has a state $x_t^i \in S^i$ which evolves according to: $x_{t+1}^i \sim p^i(\cdot|x_t^i, a_t^i, \mu_t)$, $a_t^i \sim \pi^i(\cdot|x_t^i, \mu_t)$, where $\pi^i \in \Pi^i$ is the policy for coalition $i$. We consider that this policy is chosen by a **central player** and then applied by all the infinitesimal **agents** in coalition $i$. The total reward for coalition $i$ is: $J^i(\pi^1, \ldots, \pi^m) = \mathbb{E}\left[ \sum_{t \geq 0} \gamma^t r^i(x_t^i, a_t^i, \mu_t) \right]$, where, intuitively, the expectation takes into account the average over all the agents of coalition $i$. Then, the goal is to find a Nash equilibrium between the $m$ central players.

**Definition 2.2 (Nash equilibrium for $m$-player MFTG)** *A policy profile $(\pi_*^1, \ldots, \pi_*^m) \in \Pi^1 \times \cdots \times \Pi^m$ is a **Nash equilibrium** for the above MFTG if: for all $i \in [m]$, for all $\pi^i \in \Pi^i$, $J^i(\pi^i; \pi_*^{-i}) \leq J^i(\pi_*^i; \pi_*^{-i})$, where $\pi_*^{-i}$ denotes the vector of policies for players in other coalitions except $i$.*

In other words, in a Nash equilibrium, the central players have no incentive to deviate unilaterally. This can also be expressed through the notion of exploitability, which quantifies to what extent a policy profile is far from being a Nash equilibrium, see (Heinrich et al., 2015; Perrin et al., 2020).

**Definition 2.3 (Exploitability)** *The **exploitability** of a policy profile $(\pi^1, \ldots, \pi^m) \in \Pi^1 \times \cdots \times \Pi^m$ is $\mathcal{E}(\pi^1, \ldots, \pi^m) = \sum_{i=1}^m \mathcal{E}^i(\pi^1, \ldots, \pi^m)$, where the $i$-th central player's exploitability is: $\mathcal{E}^i(\pi^1, \ldots, \pi^m) = \max_{\tilde{\pi}^i \in \Pi^i} J^i(\tilde{\pi}^i; \pi^{-i}) - J^i(\pi^i; \pi^{-i})$.*

Notice that $\mathcal{E}^i(\pi^1, \ldots, \pi^m)$ quantifies how much player $i$ can be better off by playing an optimal policy against $\pi^{-i}$ instead of $\pi^i$. In particular $\mathcal{E}(\pi^1, \ldots, \pi^m) = 0$ if and only if $(\pi^1, \ldots, \pi^m)$ is a

Nash equilibrium for the MFTG. More generally, we will use the exploitability to quantify how far $(\pi^1, \ldots, \pi^m)$ is from being a Nash equilibrium.

The main motivation behind the MFTG is that its Nash equilibrium provides an approximate Nash equilibrium in the finite-population $m$-coalition game, and the quality of the approximation increases with the number of agents. In particular, we can show that solving an MFTG provides an $\epsilon$-Nash equilibrium for a game between finite-size coalitions. The following assumptions are classical in the literature on MFC and MFTGs, see e.g. (Cui et al., 2024; Guan et al., 2024).

**Assumption 1** **(a)** *For each $i \in [m]$, the reward function $r^i(x, a, \mu)$ is bounded by a constant $C_r > 0$ and Lipschitz w.r.t. $\mu$ with constant $L_r$.*
**(b)** *The transition probability $p(x'|x, a, \mu)$ satisfies the following Lipschitz bound: $\|p(\cdot|x, a, \mu) - p(\cdot|x, a, \tilde{\mu})\|_1 \leq L_p d(\mu, \tilde{\mu})$ for every $x \in S^i$, $a \in A^i$, and $\mu, \tilde{\mu} \in \Delta(S^i)$.*
**(c)** *The policies $\pi(a|x, \mu)$ satisfy the following Lipschitz bound: $\|\pi(\cdot|x, \mu) - \pi(\cdot|x, \tilde{\mu})\|_1 \leq L_\pi d(\mu, \tilde{\mu})$ for every $x \in S^i$, and $\mu, \tilde{\mu} \in \Delta(S^i)$.*

**Theorem 2.4 (Approximate Nash equilibrium)** *Suppose that Assm. 1 holds. Let $(\pi_*^1, \ldots, \pi_*^m) \in \Pi^1 \times \cdots \times \Pi^m$ be a Nash equilibrium for the MFTG. When the discount factor $\gamma$ satisfies $\gamma(1 + L_\pi + L_p) < 1$, then $\max_{\tilde{\pi}^i} J^{i,\bar{N}}(\tilde{\pi}^i; \pi_*^{-i}) \leq J^{i,\bar{N}}(\pi_*^i; \pi_*^{-i}) + \varepsilon(N)$, for all $i \in [m]$, with $\varepsilon(N) = C \max_{i \in [m]} \left\{ |S^i| \sqrt{|A^i|} / \sqrt{N_i} \right\}$, where $C$ is a constant.*

In other words, if all the agents use the policy coming from the MFTG corresponding to their coalition, then each coalition can increase its total reward only marginally (at least when the number of agents is large enough). The proof is deferred to Appx. A. In contrast with e.g. (Saldi et al., 2018, Theorem 4.1), our result provides not only asymptotic convergence but also a rate of convergence.

## 2.3 REFORMULATION WITH MEAN-FIELD MDPs

Our next step towards RL methods is to rephrase the MFTG in the framework of Markov decision processes (MDPs). Since the game involves the population's states represented by probability distributions, the MDPs will be of mean-field type. We will thus rely on the framework of mean-field Markov decision processes (MFMDP) (Motte & Pham, 2022; Carmona et al., 2023). But in contrast with these prior works, we consider a game between MFMDPs, which is more challenging than a single MFMDP. The key remark is that, since $x_t^i$ has distribution $\mu_t^i$ and $a_t^i$ has distribution $\pi^i(\cdot|x_t^i, \mu_t)$, the expected one-step reward can be expressed as a function $\bar{r}^i$ of the $i$-th policy and the distributions:

$$\bar{r}^i(\mu_t, \bar{\pi}_t^i) = \sum_{x \in S^i} \mu_t^i(x) \sum_{a \in A^i} \bar{\pi}_t^i(a|x) r^i(x, a, \mu_t),$$

where $\bar{\pi}_t^i = \pi_t^i(\cdot|\cdot, \mu_t)$. This will help us to rewrite the problem posed to central player $i$, as an MDP. Before doing so, we introduce the following notations: $\bar{S} = \bigtimes_{i=1}^m \bar{S}^i$ is the (mean-field) state space, where $\bar{S}^i = \Delta(S^i)$ is the (mean-field) state space of population $i$. The (mean-field) state is $\bar{s}_t = \mu_t \in \bar{S}$; $\bar{A}^i = \Delta(A^i)^{|S^i|}$ is the (mean-field) action space; $\bar{r}^i : \bar{S} \times \bar{A}^i \to \mathbb{R}$ is as defined above; $\bar{p} : \bar{S} \times \bar{A}^1 \times \cdots \times \bar{A}^m \to \bar{S}$ is defined such that: $\bar{p}(\bar{s}_t, \bar{a}_t^1, \ldots, \bar{a}_t^m) = \bar{s}_{t+1}$ where, if $\bar{s}_t = (\mu_t^1, \ldots, \mu_t^m)$ and $\bar{a}_t^i = \pi^i(\cdot|\cdot, \mu_t^i)$, then $\bar{s}_{t+1} = (\mu_{t+1}^1, \ldots, \mu_{t+1}^m)$, where we recall that $\mu_{t+1}^i$ is the distribution of $x_{t+1}^i$. In other words, $\bar{p}$ encodes the transitions of the mean-field state, which depends on all the central players' (mean-field) actions. To stress the fact that the transitions are deterministic, we will sometimes use the notation $\bar{F} = \bar{p}$ to stress that this is a transition function (at the mean-field level). A **(mean-field) policy** is now a function $\bar{\pi}^i : \bar{S} \to \bar{A}^i$. In other words, the central player first chooses a function $\bar{\pi}^i$ of the mean field. When applied on $\mu_t$, $\bar{\pi}^i(\mu_t)$ returns a policy for the individual agent, i.e., $\bar{\pi}^i(\mu_t) : S^i \ni x_t^i \mapsto \bar{\pi}^i(\mu_t, x_t^i) = \pi^i(\cdot|x_t^i, \mu_t) \in \Delta(A^i)$. Although this approach may seem quite abstract, it allows us to view the problem posed to the $i$-th central player as a "classical" MDP (modulo the fact that the state is a vector of probability distributions). We can then borrow tools from reinforcement learning to solve this MDP.

**Remark 2.5** *Notice that an action for central player $i$, i.e., an element $\bar{a}^i$ of $\bar{A}^i$. From the point of view of an agent in coalition $i$, it is a decentralized policy. Then $\bar{\pi}^i$ is a mean-field policy for the central player, whose input is a mean field. This generalizes the approach proposed in (Carmona*

*et al., 2023) to the case of multiple controllers. It is different from e.g. (Yang et al., 2018), in which there is no central player and no mean-field policies. This allows to represent coalitions' behaviors that react to other coalitions' mean fields.*

### 2.4 STAGE GAME EQUILIBRIA

We now rephrase the notion of MFTG equilibrium using the notion of value function, which will lead to a connection with the concept of stage-game. To make the model more general, we also assume that the reward of coalition $i$ could function depend on the actions of all central players.

The central player of coalition $i$ aims to choose a policy $\bar{\pi}^i$ to maximize the discounted sum of rewards: $\bar{v}^i_{\bar{\pi}}(\bar{s}) = \bar{v}^i(\bar{s}, \bar{\pi}) := \mathbb{E}_{\bar{\pi}}\left[\sum_{t=0}^{\infty} \gamma^t \bar{r}^i(\bar{s}_t, \bar{a}^i_t)\right]$, where $\bar{\pi} = (\bar{\pi}^1, \ldots, \bar{\pi}^m)$ is the policy profile and $\bar{s}_0 = \bar{s}$, $\bar{s}_{t+1} \sim \bar{p}(\cdot|\bar{s}_t, \bar{a}^1_t, \ldots, \bar{a}^m_t)$, $\bar{a}^j_t \sim \bar{\pi}^i(\cdot|\bar{s}_t)$, $j = 1, \ldots, m$, $t \geq 0$.

We can now rephrase the notion of Nash equilibrium for the MFTG (Def. 2.2) in this framework.

**Definition 2.6 (Nash equilibrium for MFTG rephrased)** *An MFTG **Nash equilibrium** $\bar{\pi}_* = (\bar{\pi}^1_*, \ldots, \bar{\pi}^m_*)$ is such that for all $i = 1, \ldots, m$: $\bar{v}^i(\bar{s}, \bar{\pi}_*) \geq \bar{v}^i(\bar{s}, (\bar{\pi}^i, \bar{\pi}^{-i}_*))$, $\forall \bar{s} \in \bar{S}, \forall \bar{\pi}^i \in \bar{\Pi}^i$.*

To simplify the notation, we let $\bar{a} = (\bar{a}^1, \ldots, \bar{a}^m)$, $\bar{\pi}^{-i}(\mathrm{d}\bar{a}^{-i}|\bar{s}) = \prod_{j \neq i} \bar{\pi}^j(\mathrm{d}\bar{a}^j|\bar{s})$, $\bar{a}^{-i} \in \bar{A}^{-i} = \prod_{j \neq i} \bar{A}^j$. The Q-function for central player $i$ is defined as: $\bar{Q}^i_{\bar{\pi}}(\bar{s}, \bar{a}) = \mathbb{E}_{\bar{\pi}}\left[\sum_{t=0}^{\infty} \gamma^t \bar{r}^i(\bar{s}_t, \bar{a}^i)|\bar{s}_0 = \bar{s}, \bar{a}_0 = \bar{a}\right]$.. We now introduce an (MF)MDP for central player $i$ when the other players' policies are fixed. We define the following MDP, denoted by MDP($\bar{\pi}^{-i}$).

**Definition 2.7 (MDP($\bar{\pi}^{-i}$))** *An MDP for a central player $i$ against fixed policies of other players is a tuple $(\bar{S}, \bar{A}^i, \bar{p}_{\bar{\pi}^{-i}}, \bar{r}_{\bar{\pi}^{-i}}, \gamma)$ where $\bar{p}_{\bar{\pi}^{-i}}(\bar{s}'|\bar{s}, \bar{a}^i) = \int_{\bar{A}^{-i}} \bar{p}(\bar{s}'|\bar{s}, \bar{a})\bar{\pi}^{-i}(\mathrm{d}\bar{a}^{-i}|\bar{s})$, $\bar{r}_{\bar{\pi}^{-i}}(\bar{s}, \bar{a}^i) = \bar{r}^i(\bar{s}, \bar{a}^i)$.*

Next, we define the notion of stage game, which is a Nash equilibrium for a one-step problem. This serves as an intermediate goal in Nash Q-learning, to learn a global-in-time Nash equilibrium.

**Definition 2.8 (Stage game and stage Nash equilibrium)** *Given a (mean-field) state $\bar{s} \in \bar{S}$ and a policy profile $\bar{\pi} = (\bar{\pi}^1, \ldots, \bar{\pi}^m)$, the (mean-field) **stage game** induced by $\bar{s}$ and $\bar{\pi}$ is a static game in which player $i$ takes an action $\bar{a}^i \in \bar{A}^i$, $i = 1, \ldots, m$ and gets the reward $\bar{Q}^i_{\bar{\pi}}(\bar{s}, \bar{a}^1, \ldots, \bar{a}^m)$. Player $i$ is allowed to use a mixed strategy $\sigma^i \in \Delta(\bar{A}^i)$. A **Nash equilibrium** for this stage game is a strategy profile $\sigma_* = (\sigma^1_*, \ldots, \sigma^m_*)$ such that, for all $\sigma^i \in \Delta(\bar{A}^i)$,*

$$\sigma^1_* \cdots \sigma^m_* \bar{Q}^i_{\bar{\pi}}(\bar{s}) \geq \sigma^1_* \cdots \sigma^{i-1}_* \sigma^i \sigma^{i+1}_* \cdots \sigma^m_* \bar{Q}^i_{\bar{\pi}}(\bar{s})$$

*where we define $\sigma^1 \cdots \sigma^m \bar{Q}^i_{\bar{\pi}}(\bar{s}) := \bar{r}^i(\bar{s}, \sigma^i) + \gamma \int_{\bar{S}} \int_{\bar{A}} \bar{v}^i(\bar{s}', \bar{\pi})\bar{p}(\mathrm{d}\bar{s}'|\bar{s}, \bar{a})\sigma(\mathrm{d}\bar{a}|\bar{s})$, with $\bar{A} = \bar{A}^1 \times \cdots \times \bar{A}^m$, $\sigma(\mathrm{d}\bar{a}|\bar{s}) = \prod_{i=1}^m \sigma^i(\mathrm{d}\bar{a}^i|\bar{s})$, and $\bar{r}^i(\bar{s}, \sigma^i) := \mathbb{E}_{\bar{a}^i \sim \sigma^i} \bar{r}^i(\bar{s}, \bar{a}^i)$.*

We now define a mean-field version of the NashQ function introduced by Hu & Wellman (2003). Intuitively, it quantifies the reward that player $i$ gets when the system starts in a given state, all the player uses the stage-game equilibrium strategies for the first action, and then play according to a fixed policy profile for all remaining time steps.

**Definition 2.9 (NashQ function)** *Given a Nash equilibrium $(\sigma^1_*, \ldots, \sigma^m_*)$, the **NashQ function** of player $i$ is defined as: $\mathrm{Nash}\bar{Q}^i_{\bar{\pi}}(\bar{s}) := \sigma^1_* \cdots \sigma^m_* \bar{Q}^i_{\bar{\pi}}(\bar{s})$.*

We conclude by showing the link between Defs. 2.2 and 2.8 (the proof is in Appx. B).

**Proposition 2.10** *The following statements are equivalent: **(i)** $\bar{\pi}_* = (\bar{\pi}^1_*, \ldots, \bar{\pi}^m_*)$ is a Nash equilibrium for the MFTG with equilibrium payoff $(\bar{v}^1_{\bar{\pi}_*}, \ldots, \bar{v}^m_{\bar{\pi}_*})$; **(ii)** For every $\bar{s} \in \bar{S}$, $(\bar{\pi}^1_*(\bar{s}), \ldots, \bar{\pi}^m_*(\bar{s}))$ is a Nash equilibrium in the stage game induced by state $\bar{s}$ and policy profile $\bar{\pi}_*$*

## 3 NASH Q-LEARNING AND TABULAR IMPLEMENTATION

In this section, we present an adaptation of the celebrated Nash Q-learning of Hu & Wellman (2003) to solve MFTG. It should be noted that the original Nash Q-learning algorithm Hu & Wellman (2003) is for finite state and action spaces and to the best of our knowledge, extensions to continuous spaces have been proposed only in special cases, such as Vamvoudakis (2015); Casgrain et al. (2022), but there is no extension to continuous spaces for general games that could be applied to MFTGs. The main difficulty is the computation of the solution to the stage-game at each iteration, which relies on the fact that the action space is finite. So this algorithm cannot be applied directly to solve MFTGs.

In order to implement this method using tabular RL, we will start by discretizing the simplexes following the idea in Carmona et al. (2023). This allows us to analyze the algorithm fully. However, this approach is not scalable in terms of the number of states, which is why in Section 4, we will present a deep RL method that does not require simplex discretization.

### 3.1 DISCRETIZED MFTG

Since $S^i$ and $A^i$ are finite, $\bar{S}^i = \Delta(S^i)$ and $\Delta(A^i)$ are (finite-dimensional) simplexes. We endow $\bar{S}$ and $\Delta(A^i)$ with the distances $d_{\bar{S}}(\bar{s}, \bar{s}') = \sum_{i \in [m]} d(\bar{s}^i, \bar{s}'^i) = \sum_{i \in [m]} \sum_{x \in S^i} |\mu^i(x) - \mu'^i(x)|$, and $d_{A^i}(\bar{a}^i(\bar{s}), \bar{a}'^i(\bar{s})) = \sum_{x,a} |\pi^i(a|x, \bar{s}) - \pi'^i(a|x, \bar{s})|$, where $\bar{s}^i = \mu^i$, $\bar{a}^i(\bar{s}) = \pi^i(\cdot|\cdot, \bar{s})$. In the action space $\bar{A}^i = \{\bar{S} \to \Delta(A^i)\}$, we define the distance $d_{\bar{A}^i}(\bar{a}^i, \bar{a}'^i) = \sup_{\bar{s} \in \bar{S}} d_{A^i}(\bar{a}^i(\bar{s}), \bar{a}'^i(\bar{s}))$. However, $\bar{S}$ and $\bar{A}^i$ are not finite. To apply the tabular Q-learning algorithm, we replace $\bar{S}$ and $\bar{A}^i$ with finite sets. For $i = 1, \ldots, m$, let $\check{S}^i \subset \bar{S}^i$ and $\check{\Delta}(A^i) \subset \Delta(A^i)$ be finite approximations of $\bar{S}^i$ and $\Delta(A^i)$. We then define the (mean-field) **finite state space and action space** $\check{S} = \Pi_{i=1}^m \check{S}^i \subset \bar{S}$ and $\check{A}^i = \{\check{a}^i : \check{S} \to \check{\Delta}(A^i)\}$. Let $\epsilon_S = \max_{\bar{s} \in \bar{S}} \min_{\check{s} \in \check{S}} d_{\bar{S}}(\bar{s}, \check{s})$ and $\epsilon_A = \max_i \max_{\bar{a}^i \in \bar{A}^i} \min_{\check{a}^i \in \check{A}^i} d_{\bar{A}^i}(\bar{a}^i, \check{a}^i)$, which characterize the fineness of the discretization. The policy space of each player $i$ is $\check{\Pi}^i = \{\check{\pi}^i : \check{S} \to \Delta(\check{A}^i)\}$. We will also use the projection operator $\text{Proj}_{\check{S}} : \bar{S} \to \check{S}$, which maps $\bar{s}$ to the closest point in $\check{S}$ (ties broken arbitrarily). This will ensure the state takes value in $\check{S}$. Specifically, given a state $\check{s}_t$ and a joint action $(\check{a}_t^1, \ldots, \check{a}_t^m)$, we generate $\bar{s}_{t+1} = \bar{F}(\check{s}_t, \check{a}_t^1, \ldots, \check{a}_t^m)$. Then, we project $\bar{s}_{t+1}$ back to $\check{S}$ and denote the projected state by $\check{s}_{t+1} = \text{Proj}_{\check{S}}(\bar{s}_{t+1})$. This finite space setting can be regarded as a special case of an $m$-player stochastic game, and the Theorem 2 in (Fink, 1964) guarantees the existence of a Nash equilibrium.

### 3.2 NASH Q-LEARNING ALGORITHM

We briefly describe the tabular Nash Q-learning algorithm, similar to the algorithm of Hu & Wellman (2003). The main idea is that, instead of using classical $Q$-learning updates, which involve only the player's own $Q$-function, the players will use the $\text{Nash}Q$ function for a stage game.

At each step $t$, the players use their current estimate of the $Q$-functions to define a stage game. They compute the Nash equilibrium, say $(\check{\sigma}^1, \ldots, \check{\sigma}^m) \in \prod_{i=1}^m \check{\Pi}^i$, and deduce the associated $\text{Nash}Q$ function, which is then used to update their estimates of the $Q$-functions.

At each step $t$, player $i$ observes $\check{s}$ and takes an action according to a behavior policy chosen to ensure exploration. Then, she observes the reward, actions of each player, and the next state $\check{s}'$. She then solves the stage game with rewards $(\check{Q}_t^1(\check{s}'), \ldots, \check{Q}_t^m(\check{s}'))$, where $\check{Q}_t^i(\check{s}') : (\bar{a}^1, \ldots, \bar{a}^m) \mapsto \check{Q}_t^i(\check{s}', \bar{a}^1, \ldots, \bar{a}^m)$. Let $(\check{\pi}_*^{i,1}(\check{s}'), \ldots, \check{\pi}_*^{i,m}(\check{s}'))$ be the Nash equilibrium obtained on player $i$'s belief. The NashQ function of player $i$ is defined as: $\text{Nash}\check{Q}_t^i(\check{s}') = \check{\pi}_*^{i,1} \cdots \check{\pi}_*^{i,m} \check{Q}_t^i(\check{s}')$. From here, she updates the Q-values according to the following rule, where $\alpha_t$ is a learning rate:

$$\check{Q}_{t+1}^i(\check{s}, \check{a}^1, \ldots, \check{a}^m) = (1 - \alpha_t)\check{Q}_t^i(\check{s}, \check{a}^1, \ldots, \check{a}^m) + \alpha_t(\check{r}_t^i + \beta \text{Nash}\check{Q}_t^i(\check{s}')). \tag{1}$$

It is noted that in each iteration, the Q-values of each player are updated asynchronously based on the observation. The detailed algorithm is described in Algo. 1 in Appx. D due to space constraints.

### 3.3 NASH Q-LEARNING ANALYSIS

We will see that $\check{Q}_t^i$ from Algo. 1 converges to $\check{Q}_{\check{\boldsymbol{\pi}}_*}^i$ under the following assumption, which is classical in the literature on NashQ-learning, see e.g. Hu & Wellman (2003); Yang et al. (2018). We use it for the proof although it seems that in practice the algorithm works well even when this assumption does not hold.

**Assumption 2** (a) *Every state $\check{s} \in \check{S}$ and action $\check{a}^i \in \check{A}^i$, $i = 1, \ldots, m$, are visited infinitely often.*
(b) $\alpha_t$ *satisfies the following two conditions for all* $t, \check{s}, \check{a}^1, \ldots, \check{a}^m$: **1.** $0 \leq \alpha_t(\check{s}, \check{a}^1, \ldots, \check{a}^m) < 1$, $\sum_{t=0}^{\infty} \alpha_t(\check{s}, \check{a}^1, \ldots, \check{a}^m) = \infty$, $\sum_{t=0}^{\infty} \alpha_t^2(\check{s}, \check{a}^1, \ldots, \check{a}^m) < \infty$, *the latter two hold uniformly and with probability 1.* **2.** $\alpha_t(\check{s}, \check{a}^1, \ldots, \check{a}^m) = 0$, *if* $(\check{s}, \check{a}^1, \ldots, \check{a}^m) \neq (\check{s}_t, \check{a}_t^1, \ldots, \check{a}_t^m)$.
(c) *One of the following two conditions holds:* **1.** *Every stage game* $(\check{Q}_t^1(\check{s}'), \ldots, \check{Q}_t^m(\check{s}'))$ *for all* $t$ *and* $\check{s}$*, has a global optimal point, and players' payoff in this equilibrium are used to update their Q-functions.* **2.** *Every stage game* $(\check{Q}_t^1(\check{s}'), \ldots, \check{Q}_t^m(\check{s}'))$ *for all* $t$ *and* $\check{s}$*, has a saddle point, and players' payoff in this equilibrium are used to update their Q-functions.*

Here, a **global optimal point** is a joint policy of the stage game such that each player receives her highest payoff following this policy. A **saddle point** is a Nash equilibrium policy of the stage game, and each player would receive a higher payoff provided at least one of the other players takes a policy different from the Nash equilibrium policy.

**Theorem 3.1 (NashQ-learning convergence)** *Under Assm. 2, $\check{Q}_t = (\check{Q}_t^1, \ldots, \check{Q}_t^m)$, updated by (1) converges to the Nash equilibrium Q-functions $\check{Q}_{\check{\boldsymbol{\pi}}_*} = (\check{Q}_{\check{\boldsymbol{\pi}}_*}^1, \ldots, \check{Q}_{\check{\boldsymbol{\pi}}_*}^m)$.*

We omit the proof of Theorem 3.1 as it is essentially the same as in (Hu & Wellman, 2003). We then focus on the difference between the approximated Nash Q-function, $\check{Q}_t^i(\text{Proj}_{\check{S}}(\bar{s}), \text{Proj}_{\check{A}^1}(\bar{a}^1), \ldots, \text{Proj}_{\check{A}^m}(\bar{a}^m))$ and the true Nash Q-function, $\bar{Q}_{\check{\boldsymbol{\pi}}_*}^i(\bar{s}, \bar{a}^1 \ldots \bar{a}^m)$, in the infinite space $\bar{S} \times \bar{A}^i \times \cdots \times \bar{A}^m$. For this proof, we use the following assumption, which is an extension to the multi-player setting of the assumptions in (Carmona et al., 2023).

**Assumption 3** (a) *For each $i$, $\bar{r}^i$ is bounded and Lipschitz continuous w.r.t. $(\bar{s}_t, \bar{a}_t^i)$ with constant $L_{\bar{r}^i}$. $\bar{F}$ is Lipschitz continuous w.r.t. $(\bar{s}, \bar{a}^1, \ldots, \bar{a}^m)$ with constant $L_{\bar{F}}$ in expectation.*
(b) *$\bar{v}_{\bar{\boldsymbol{\pi}}}^i$ is Lipschitz continuous w.r.t. $\bar{s}$ with constant $L_{\bar{v}_{\bar{\boldsymbol{\pi}}}^i}$.*

Assm. 3 (a) can be achieved with suitable conditions on the game. The boundedness of the reward function, together with the discount factor $0 < \gamma < 1$, can also lead to the boundedness of the payoff function $\bar{v}_{\check{\boldsymbol{\pi}}_*}^i$. For classical MDPs, Lipschitz continuity of the value function can be derived from assumptions on the model as in (Motte & Pham, 2022).

To alleviate the notation, we let: $\text{Proj}(\bar{s}, \bar{a}^1 \ldots \bar{a}^m) = (\text{Proj}_{\check{S}}(\bar{s}), \text{Proj}_{\check{A}^1}(\bar{a}^1), \ldots, \text{Proj}_{\check{A}^m}(\bar{a}^m))$.

**Theorem 3.2 (Discrete problem analysis)** *Let $\epsilon > 0$. Suppose Assm. 3 holds and there is a unique pure policy $\bar{\boldsymbol{\pi}}_*^p$ for the MFTG for each $i$ and $\bar{s} \in \bar{S}$, the function $v_{\bar{\boldsymbol{\pi}}_*^p}^i(\bar{s})$ is a global optimal point for the stage game $\bar{Q}_{\bar{\boldsymbol{\pi}}_*^p}^i(\bar{s})$. Then, if $t$ is large enough, for each $i$, $\bar{s} \in \bar{S}$, $i = 1, 2, \cdots$, we have $|\check{Q}_t^i(\text{Proj}(\bar{s}, \bar{a}^1 \ldots \bar{a}^m)) - \bar{Q}_{\bar{\boldsymbol{\pi}}_*^p}^i(\bar{s}, \bar{a}^1 \ldots \bar{a}^m)| \leq \epsilon'$, where $\epsilon' = \epsilon + C_1 \epsilon_A + C_2 \epsilon_S$, with $\epsilon_S$ and $\epsilon_A$ defined above, respectively, $C_1 = \frac{1}{1-\gamma}(L_{\bar{r}^i} + \gamma L_{\bar{v}_{\bar{\boldsymbol{\pi}}_*}^i} L_{\bar{F}} m)$ and $C_2 = \frac{\gamma}{1-\gamma} L_{\bar{v}_{\bar{\boldsymbol{\pi}}_*}^i} + L_{\bar{r}^i} + \gamma L_{\bar{v}_{\bar{\boldsymbol{\pi}}_*}^i} L_{\bar{F}}$.*

Note the first $\epsilon$ in the bound $\epsilon'$ can be chosen arbitrarily small provided $t$ is large enough. The second and the third term are controlled by $\epsilon_A$ and $\epsilon_S$ and can be small if we choose a finer simplex approximation. The proof is provided in Appx. C.

## 4 DEEP RL FOR MFTG

While the above extension of the NashQ learning algorithm has the advantage of being fully analyzable and enjoying convergence guarantees, it is not scalable to large state and action spaces. Indeed, it requires discretizing the simplexes of distributions on states and actions. The number of points increases exponentially in the number of states and actions, which makes the algorithm intractable

for very fine discretizations. Furthermore, each step relies on solving a stage game and computing a Nash equilibrium is a difficult task for large games, even they are static.

For this reason, we now present a deep RL algorithm whose main advantages are that it does not require discretizing the simplexes and does not require solving any stage-game. The state and action distributions are represented as vectors (containing the probability mass functions) and passed as inputs to neural networks for the policies and the value functions. At the level of the central player for coalition $i$, one action is an element $\bar{a}^i \in \bar{A}^i$. Although it corresponds to a mixed policy at the level of the individual agent, it represents one action for the central player. We focus on learning deterministic central policies, meaning functions that map a mean-field state $\bar{s}$ to a mean-field action $\bar{a}^i$. To this end, we use a variant of the deep deterministic policy gradient algorithm (DDPG) (Lillicrap et al., 2016), as shown in Algo. 2 in Appx. D. Our algorithm substantially differs from the DDPG algorithm as the two players' behaviors are coupled. Each player interacts with a dynamic environment that the other player also influences. Unlike the tabular Nash Q-learning algorithm, it is generally difficult to have a rigorous proof of convergence due to the complexity of deep neural networks. Although the theoretical convergence of some algorithms has been studied, such as deep Q-learning (Fan et al., 2020), deterministic policy gradient (Xiong et al., 2022), and actor-critic algorithms with multi-layer neural networks (Tian et al., 2024), to the best of our knowledge, the convergence of DDPG under assumptions that could be applied to our setting has not been established. Also, in the case of MFTGs, we would need to analyze the convergence to a Nash equilibrium, which is more complex than the solution to an MDP. Therefore, we leave for future work the theoretical analysis and focus on numerical analysis: we use several numerical metrics to measure the performance of DDPG-MFTG Algo. 2, as detailed in the next section.

## 5 NUMERICAL EXPERIMENTS

**Metrics.** To assess the convergence of our algorithms, we use several metrics. First, we check the testing rewards of each central player (i.e., the total reward for each coalition, averaged over the testing set of initial distributions). But this is not sufficient to show that the policies form a Nash equilibrium of the MFTG. For this, we compute the exploitability. This requires training a best response (BR) policy for each player independently, which is also done with deep RL, using the DDPG method. Last, we also check the evolution of the distributions to make sure that they match what we expect to happen in the Nash equilibrium. The pseudo-codes for evaluating a policy profile and computing the exploitability are respectively provided in Algs. 4 and 5 in Appx. E.

**Training and testing sets.** The training set consists of randomly generated tuples of distributions, and each element of the tuple represents the initial distribution of a player. The testing set consists of a finite number of tuples of distributions that are not in the training set. Details of the training and testing sets are described case by case.

**Baseline.** To the best of our knowledge, there are no RL algorithms that can be applied to the type of MFTG problems we study here. In the absence of standard baselines, we will use two types of baselines, for each of our algorithms. For small-scale examples, we discretize the mean-field state and action spaces and employ DNashQ-MFTG. Here, we use as a baseline an algorithm where each coalition runs an independent mean field type Q-learning (after suitable discretization of the simplexes). We call this method Independent Learning-Mean Field Type Game (IL-MFTG for short). For larger scale examples with many states, we use the DDPG-based methods described in the previous section. In this case, we use as a baseline an ablated DDPG method in which each central player can only see her own (mean-field) state. For both our algorithms and the baselines, the exploitability is computed using our original class of policies, see Algo. 5.

**Games.** We present here 3 examples. Two more are presented in Appx. F.4 and F.5. Table 1 in Appx. summarizes the average improvements obtained by our method (at least 30% in each game).

**Example 1: 1D Population Matching Grid Game** There are $m = 2$ populations. The agent's state space is a 3-state 1D grid world. The possible actions are moving left, staying, and moving right, with individual noise perturbing the movements. The rewards encourage Coalition 1 to stay where it is initialized but also to consider avoiding Coalition 2, and encourage Coalition 2 to match Coalition 1. For the model details and the training and testing distributions, see Appx. F.1. We implement **DNashQ-MFTG** to solve this game. The numerical results are presented in Fig. 1. We make the following observations. **Testing reward curves:** Fig 1 (left) shows the testing rewards.

In this game setting, the Nash equilibrium is that Coalition 1 stays where it is but considers the impact of Coalition 2 at the same time, while Coalition 2 matches with Coalition 1 perfectly. The testing reward for Coalition 1 increases during the first two thousand episodes. The testing reward for Coalition 2 increases at the first three thousand episodes and fluctuates below 0 due to the noise in the dynamics. **Exploitability curves:** Fig. 1 (middle) shows the averaged exploitabilities over the testing sets and players. The game reaches Nash equilibrium around 4000 episodes, with slight fluctuations after that. However, the independent learner remains high exploitability. **Distribution plots:** Fig. 1 (right) illustrates the distribution evolution during the game. After training, Coalition 1 mainly stays where it is while Coalition 2 tries to match with Coalition 1. See Appx. F.1 for details.

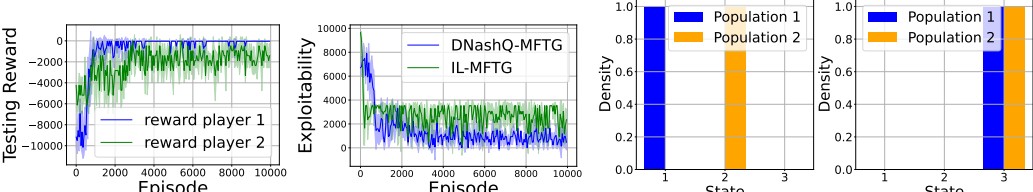

Figure 1: **Ex. 1:** Left and middle: averaged testing rewards and exploitabilities resp. (mean $\pm$ stddev). Right: one realization of population evolution at $t = 0$ and $4$ for one testing distribution.

**Example 2: Four-room with crowd aversion** There are $m = 2$ populations. The agent's state space is a 2D grid world composed of $4$ rooms of size $5 \times 5$ connected by $4$ doors, as shown in Fig. 2 (right). The policies' inputs are thus of dimension $2 \times 4 \times 5 \times 5 = 200$. The reward function encourages the two populations to spread as much as possible (to maximize the entropy of the distribution) while avoiding each other; furthermore, Coalition 2 has a penalty for going to rooms other than the one she started in. See Appx. F.2 for details of the reward and the training and testing distributions. We implement **DDPG-MFTG** to solve this game. The numerical results are presented in Fig. 2. We make the following observations. **Testing reward curves:** Fig. 2 (left, top) shows the testing rewards. **Exploitability curves:** Fig. 2 (left, bottom) shows the average exploitabilities over the testing set and players. The DDPG-MFTG algorithm performs better. **Distribution plots:** Figs. 2 (right) illustrates the distribution evolution during the game for a (pair of) initial distributions and for the policy obtained by DDPG-MFTG algorithm and the baseline. We see that the populations spread well in any case, but with DDPG-MFTG, Coalition 1 can see where Coalition 2 is and then decides to avoid that room. This explains the better performance of the DDPG-MFTG algorithm.

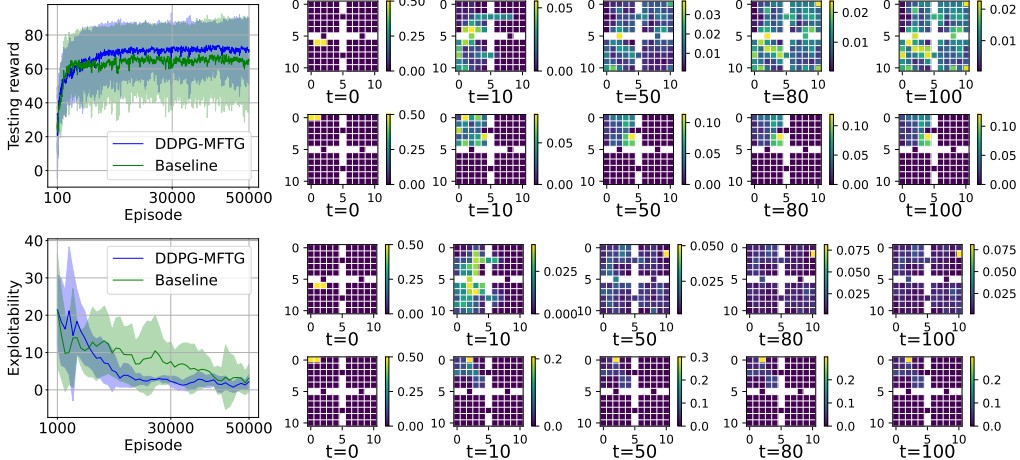

Figure 2: **Ex. 2:** Left, top and bottom: averaged testing rewards and exploitabilities resp. (mean $\pm$ stddev). Right, two top rows: distribution evolution of the two populations with our method. Right two bottom rows: distribution evolution with the baseline. Colorbars indicate density values.

**Example 3: Predator-prey 2D with 4 groups** We now present an example with more coalitions. There are $m = 4$ populations. The player's state space is a $5 \times 5$-state 2D grid world with walls on the boundaries (no periodicity). The reward functions represent the idea that Coalition 1 is a predator of Coalition 2. Coalition 2 avoids Coalition 1 and chases Coalition 3, which avoids Coalition 2 while chasing Coalition 4. Coalition 4 tries to avoid Coalition 3. There is also a cost for moving. See Appx. F.3 for details of the reward and the training and testing distributions. We implement **DDPG-MFTG** to solve this game. The numerical results are presented in Fig. 3. We make the following observations. The **testing reward curves** (Fig. 8 in Appx.) do not show a clear increase for the same reason as the previous example. **Exploitability curves:** Fig. 3 (left) shows the averaged exploitabilities over the testing set and players. Initially, the baseline and DDPG-MFTG have similar exploitability for the first several thousand episodes. However, after that period, the baseline maintains higher exploitability than DDPG-MFTG. The exploitability of DDPG-MFTG is close to 0 but still fluctuates between 0 and 100. This instability is because Deep RL can only approximate the best response and cannot achieve it with absolute accuracy. **Distribution plots:** Fig. 3 (right) shows the distribution evolution during testing. Coalition 1 chases Coalition 2. Coalition 2 tries to catch Coalition 3 while avoiding Coalition 1. Coalition 3 tries to catch Coalition 4 while escaping from Coalition 2. Coalition 4 simply escapes from Coalition 3. Testing rewards are shown in Appx. F.3.

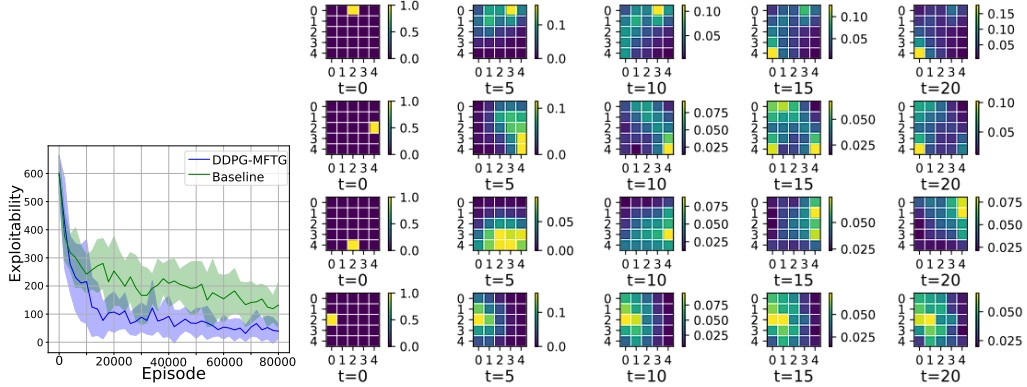

Figure 3: **Ex. 3:** Left: averaged exploitabilities (mean $\pm$ stddev). Right: populations evolution, one coalition per row and one time per column: $t = 0, 5, 10, 15, 20$. Colorbars indicate density values.

## 6 CONCLUSION

**Summary.** In this work, we made both theoretical and numerical contributions. First, we proved that the Nash equilibrium for a mean-field type game provides an approximate Nash equilibrium for a game between coalitions of finitely many agents, and we obtained a rate of convergence. We then proposed the first (to our knowledge) value-based RL methods for MFTGs: a tabular RL and a deep RL algorithm. We applied them to several MFTGs. Our proposed methods provide a way to approximately compute the Nash equilibrium of finite number players, which is known to be hard to solve numerically. We proved the convergence of the tabular algorithm, and through extensive experiments, we illustrated the scalability of the deep RL method.

**Related works.** Carmona et al. (2020); uz Zaman et al. (2024); Zaman et al. (2024) studied RL for MFTGs of LQ form only, with specific methods when the policy is deterministic and linear, while our algorithms are for generic MFTGs with discrete spaces. (Motte & Pham, 2022; Carmona et al., 2023) focused on single MFMDPs while we consider a game between MFMDPs. Subramanian & Mahajan (2019); Guo et al. (2019); Elie et al. (2020); Cui & Koeppl (2021) propose RL for MFGs but are limited to population-independent policies. Perrin et al. (2022) studied population-dependent policies, but only for MFGs, in which players are infinitesimal; their method cannot solve MFTG because each player has a macroscopic impact on the other groups.

**Limitations and future directions.** We did not provide proof of convergence for the deep RL algorithm due to the difficulties related to analyzing deep neural networks and because we aim for Nash equilibria and not just MDPs. Furthermore, we would like to apply our algorithms to more realistic examples and investigate further the difference with the baseline. We are also interested in applying other deep RL algorithms and seeing their performance in MFTGs of increasing complexities.

**Reproducibility statement.** We have included all the relevant details to ensure reproducibility. Appx. D and E give pseudo-codes for all the algorithms, including how we evaluate the performance of our method using the exploitability metric. Appx. F gives all the detailed definition of the environments, provides extra numerical results and also gives all the details about the implementation such as neural network architectures and hyperparameter choices for training. and Appx. G shows sweeps over hyperparameters to illustrate the sensitivity of our algorithms.

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

## A  PROOF OF APPROXIMATE NASH PROPERTY

We prove Theorem 2.4.

**Proof:** For each $i \in [m]$, we first define the distance between two distributions $\mu_t^i, \tilde{\mu}_t^i \in \Delta(S^i)$ to be

$$d(\mu_t^i, \tilde{\mu}_t^i) = ||\mu_t^i - \tilde{\mu}_t^i||_1 = \sum_{x \in S^i} |\mu_t^i(x) - \tilde{\mu}_t^i(x)|$$

For $\mu_t, \tilde{\mu}_t \in \Delta(S^1) \times \cdots \times \Delta(S^m)$, we also define

$$d(\mu_t, \tilde{\mu}_t) = \max_i d(\mu_t^i, \tilde{\mu}_t^i)$$

We first derive a bound for $\mathbb{E}||\mu_0^{i,\bar{N}} - \mu_0^i||_1$. The idea is inspired by the Lemma 7 in (Guan et al., 2024). Since $x_0^{ij}$ are i.i.d. from $\mu_0^i$, for all $x \in S^i$,

$$\mathbb{E}||\mu_0^{i,\bar{N}} - \mu_0^i||_2^2 = \mathbb{E}\left[\sum_{x \in S^i} \left(\frac{1}{N_i}\sum_{j=1}^{N_i} \delta_{x_0^{ij}}(x) - \mu_0^i(x)\right)^2\right]$$

$$= \mathbb{E}\left[\sum_{x \in S^i} \frac{1}{N_i^2}\left(\sum_{j=1}^{N_i}\left(\delta_{x_0^{ij}}(x) - \mu_0^i(x)\right)\right)^2\right]$$

$$= \sum_{x \in S^i} \frac{1}{N_i^2}\mathbb{E}\left[\left(\sum_{j=1}^{N_i}\left(\delta_{x_0^{ij}}(x) - \mu_0^i(x)\right)\right)^2\right]$$

$$= \sum_{x \in S^i} \frac{1}{N_i^2}\text{Var}\left(\sum_{j=1}^{N_i} \delta_{x_0^{ij}}(x)\right)$$

$$= \frac{1}{N_i^2}\sum_{x \in S^i}\sum_{j=1}^{N_i}\text{Var}\left(\delta_{x_0^{ij}}(x)\right) \quad \text{as } x_0^{ij} \text{ are i.i.d.}$$

$$= \frac{1}{N_i^2}\sum_{j=1}^{N_i}\sum_{x \in S^i}\left(\mathbb{E}\left[\delta_{x_0^{ij}}^2(x)\right] - \left(\mu_0^i(x)\right)^2\right)$$

$$= \frac{1}{N_i^2}\sum_{j=1}^{N_i}\sum_{x \in S^i}\left(\mu_0^i(x) - \left(\mu_0^i(x)\right)^2\right) \quad \text{as } \mathbb{E}\left[\delta_{x_0^{ij}}^2(x)\right] = \mu_0^i(x)$$

$$\leq \frac{1}{N_i^2}\sum_{j=1}^{N_i}\sum_{x \in S^i}\mu_0^i(x) = \frac{1}{N_i} \tag{2}$$

So we have:

$$\mathbb{E}||\mu_0^{i,\bar{N}} - \mu_0^i||_1 \leq \sqrt{|S^i|}\mathbb{E}||\mu_0^{i,\bar{N}} - \mu_0^i||_2 \leq \sqrt{\frac{|S^i|}{N_i}}$$

the second inequality above is due to the Jensen's inequality. Thus, for each $i \in [m]$, as $N_i \to +\infty$, we have

$$\mathbb{E}d(\mu_0^{\bar{N}}, \mu_0) \to 0 \text{ a.e.}$$

Next, we consider the distance between the joint state-action distribution of population $i$ at time $t$ and its empirical distribution. We denote the joint state-action distribution of population $i$ at time $t$ to be

$$\nu_t^i(x, a) = \mu_t^i(x)\pi_t^i(a|x, \mu_t)$$

and the empirical state-action distribution of population $i$ at time $t$ to be

$$\nu_t^{i,\bar{N}} = \frac{1}{N_i}\sum_{j=1}^{N_i} \delta_{x_t^{ij}, a_t^{ij}}$$

then, we have

$$\mathbb{E} \sum_{x,a} |\nu_t^i(x,a) - \nu_t^{i,\bar{N}}(x,a)|$$

$$= \mathbb{E} \sum_{x,a} |\mu_t^i(x)\pi_t^i(a|x,\mu_t) - \mu_t^{i,\bar{N}}(x)\pi_t^i(a|x,\mu_t)$$

$$+ \mu_t^{i,\bar{N}}(x)\pi_t^i(a|x,\mu_t) - \mu_t^{i,\bar{N}}(x)\pi_t^i(a|x,\mu_t^{\bar{N}})$$

$$+ \mu_t^{i,\bar{N}}(x)\pi_t^i(a|x,\mu_t^{\bar{N}}) - \nu_t^{i,\bar{N}}(x,a)|$$

$$\leq \mathbb{E} \sum_{x,a} |\pi_t^i(a|x,\mu_t)(\mu_t^i(x) - \mu_t^{i,\bar{N}}(x))|$$

$$+ \mathbb{E} \sum_{x,a} |\mu_t^{i,\bar{N}}(x)(\pi_t^i(a|x,\mu_t) - \pi_t^i(a|x,\mu_t^{\bar{N}}))|$$

$$+ \mathbb{E} \sum_{x,a} \left| \frac{1}{N_i}\sum_{j=1}^{N_i} \delta_{x_t^{ij}}(x) \left( \pi_t^i(a|x,\mu_t^{\bar{N}}) - \frac{\frac{1}{N_i}\sum_{j=1}^{N_i}\delta_{x_t^{ij},a_t^{ij}}(x,a)}{\frac{1}{N_i}\sum_{j=1}^{N_i}\delta_{x_t^{ij}}(x)} \right) \right|$$

$$\leq \mathbb{E} \sum_{x,a} |\pi_t^i(a|x,\mu_t)||(\mu_t^i(x) - \mu_t^{i,\bar{N}}(x))|$$

$$+ \mathbb{E} \sum_{x,a} |\mu_t^{i,\bar{N}}(x)||(\pi_t^i(a|x,\mu_t) - \pi_t^i(a|x,\mu_t^{\bar{N}}))|$$

$$+ \mathbb{E} \sum_{x,a} \left| \frac{1}{N_i}\sum_{j=1}^{N_i} \delta_{x_t^{ij}}(x) \left( \pi_t^i(a|x,\mu_t^{\bar{N}}) - \frac{\frac{1}{N_i}\sum_{j=1}^{N_i}\delta_{x_t^{ij},a_t^{ij}}(x,a)}{\frac{1}{N_i}\sum_{j=1}^{N_i}\delta_{x_t^{ij}}(x)} \right) \right|$$

$$\leq \mathbb{E} \sum_{x} |\mu_t^i(x) - \mu_t^{i,\bar{N}}(x)|$$

$$+ \mathbb{E} \sum_{x} |\mu_t^{i,\bar{N}}(x)| L_\pi d(\mu_t, \mu_t^{\bar{N}})$$

$$+ \sum_{x,a} \mathbb{E} \left| \frac{1}{N_i}\sum_{j=1}^{N_i} \delta_{x_t^{ij}}(x) \left( \pi_t^i(a|x,\mu_t^{\bar{N}}) - \frac{\frac{1}{N_i}\sum_{j=1}^{N_i}\delta_{x_t^{ij},a_t^{ij}}(x,a)}{\frac{1}{N_i}\sum_{j=1}^{N_i}\delta_{x_t^{ij}}(x)} \right) \right|$$

$$\leq (1 + L_\pi)\mathbb{E} d(\mu_t, \mu_t^{\bar{N}})$$

$$+ \sum_{x,a} \mathbb{E} \left| \frac{1}{N_i}\sum_{j=1}^{N_i} \delta_{x_t^{ij}}(x) \left( \pi_t^i(a|x,\mu_t^{\bar{N}}) - \frac{\frac{1}{N_i}\sum_{j=1}^{N_i}\delta_{x_t^{ij},a_t^{ij}}(x,a)}{\frac{1}{N_i}\sum_{j=1}^{N_i}\delta_{x_t^{ij}}(x)} \right) \right|$$

Given $\{x_t^{ij}\}_{j=1}^{N_i}$, let $N_i^t(x) = \sum_{j=1}^{N_i}\delta_{x_t^{ij}}(x) = N_i\mu_t^{i,\bar{N}}(x)$. We can decompose $S^i$ into $S^i = S_+^i \cup S_0^i$, where $S_+^i = \{x \in S^i : N_i^t(x) > 0\}$ and $S_0^i = \{x \in S^i : N_i^t(x) = 0\}$. For $x \in S_0^i$, we have $\mu_t^{i,\bar{N}}(x) = 0$ and $\nu_t^{i,\bar{N}}(x,a) = 0$, so

$$\mathbb{E} \left| \mu_t^{i,\bar{N}}(x)\pi_t^i(a|x,\mu_t^{\bar{N}}) - \nu_t^{i,\bar{N}}(x,a) \right| = 0$$

For a fixed $x \in S_+^i$, since $a_t^{ij}$ are i.i.d. with distribution $\pi^i(\cdot|x,\mu_t^{\bar{N}})$, we have

$$\mathbb{E}_{a_t^{ij}} \left[ \frac{\sum_{j=1}^{N_i}\delta_{x_t^{ij},a_t^{ij}}(x,a)}{\sum_{j=1}^{N_i}\delta_{x_t^{ij}}(x)} \right] = \pi_t^i(a|x,\mu_t^{\bar{N}}).$$

Thus, similarly to (2), for $x \in S_+^i$ we have

$$
\mathbb{E}_{a_t^{ij}} \left\| \pi_t^i(\cdot|x, \mu_t^{\bar{N}}) - \frac{\frac{1}{N_i}\sum_{j=1}^{N_i} \delta_{x_t^{ij}, a_t^{ij}}(x, \cdot)}{\frac{1}{N_i}\sum_{j=1}^{N_i} \delta_{x_t^{ij}}(x)} \right\|_2^2
$$

$$
= \mathbb{E}_{a_t^{ij}} \left[ \sum_{a \in A^i} \left( \pi_t^i(a|x, \mu_t^{\bar{N}}) - \frac{1}{N_i(x)} \sum_{j=1}^{N_i} \delta_{x_t^{ij}, a_t^{ij}}(x, a) \right)^2 \right]
$$

$$
\leq \left[ \frac{1}{N_i^t(x)} \right],
$$

and

$$
\mathbb{E}_{a_t^{ij}} \left\| \pi_t^i(\cdot|x, \mu_t^{\bar{N}}) - \frac{\frac{1}{N_i}\sum_{j=1}^{N_i} \delta_{x_t^{ij}, a_t^{ij}}(x, \cdot)}{\frac{1}{N_i}\sum_{j=1}^{N_i} \delta_{x_t^{ij}}(x)} \right\|_1 \leq \frac{\sqrt{|A^i|}}{\sqrt{N_i^t(x)}}
$$

Thus,

$$
\sum_{x,a} \mathbb{E} \left| \frac{1}{N_i} \sum_{j=1}^{N_i} \delta_{x_t^{ij}}(x) \left( \pi_t^i(a|x, \mu_t^{\bar{N}}) - \frac{\frac{1}{N_i}\sum_{j=1}^{N_i} \delta_{x_t^{ij}, a_t^{ij}}(x, a)}{\frac{1}{N_i}\sum_{j=1}^{N_i} \delta_{x_t^{ij}}(x)} \right) \right|
$$

$$
\leq \sum_x \mathbb{E} \left[ \mu_t^{i, N_1 \dots N_m}(x) \frac{\sqrt{|A^i|}}{\sqrt{N_i^t(x)}} \right]
$$

$$
= \sum_x \mathbb{E} \sqrt{\frac{\mu_t^{i, N_1 \dots N_m}(x)|A^i|}{N_i}} \leq \frac{|S^i|\sqrt{|A^i|}}{\sqrt{N_i}}
$$

Therefore, we have

$$
\mathbb{E} \sum_{x,a} |\nu_t^i(x, a) - \nu_t^{i, \bar{N}}(x, a)| \leq (1 + L_\pi)\mathbb{E}d(\mu_t, \mu_t^{\bar{N}}) + \frac{|S^i|\sqrt{|A^i|}}{\sqrt{N_i}}
$$

On the other hand, for any $t \geq 1$, we have

$$
\mu_{t+1}^i(x') = \sum_{x,a} p(x'|x, a, \mu_t)\nu_t^i(x, a)
$$

and

$$
\mu_{t+1}^{i, \bar{N}}(x') = \sum_{x,a} p(x'|x, a, \mu_t^{\bar{N}})\nu_t^{i, \bar{N}}(x, a).
$$

Moreover,

$$\mathbb{E}\|\mu_{t+1}^i - \mu_{t+1}^{i,\bar{N}}\|_1$$

$$= \mathbb{E}\sum_{x'}|\mu_{t+1}^i(x') - \mu_{t+1}^{i,\bar{N}}(x')|$$

$$= \mathbb{E}\sum_{x'}|\sum_{x,a}p(x'|x,a,\mu_t)\nu_t^i(x,a) - \sum_{x,a}p(x'|x,a,\mu_t^{\bar{N}})\nu_t^{i,\bar{N}}(x,a)|$$

$$\leq \mathbb{E}\sum_{x'}|\sum_{x,a}p(x'|x,a,\mu_t)\nu_t^i(x,a) - \sum_{x,a}p(x'|x,a,\mu_t)\nu_t^{i,\bar{N}}(x,a)|$$

$$+ \mathbb{E}\sum_{x'}|\sum_{x,a}p(x'|x,a,\mu_t)\nu_t^{i,\bar{N}}(x,a) - \sum_{x,a}p(x'|x,a,\mu_t^{\bar{N}})\nu_t^{i,\bar{N}}(x,a)|$$

$$\leq \sum_{x,a}\mathbb{E}|\nu_t^i(s,a) - \nu_t^{i,\bar{N}}(s,a)|$$

$$+ \mathbb{E}\sum_{x'}\sum_{x,a}|(p(x'|x,a,\mu_t) - p(x'|x,a,\mu_t^{\bar{N}}))\nu_t^{i,\bar{N}}(x,a)|$$

$$\leq \sum_{x,a}\mathbb{E}|\nu_t^i(s,a) - \nu_t^{i,\bar{N}}(s,a)| + \mathbb{E}\sum_{x,a}L_p d(\mu_t^i,\mu_t^{i,\bar{N}})\nu_t^{i,\bar{N}}(x,a)$$

$$\leq (1 + L_\pi + L_p)\mathbb{E}d(\mu_t,\mu_t^{\bar{N}}) + |S^i|\sqrt{|A^i|}\frac{1}{\sqrt{N_i}}$$

Thus, for $t \geq 1$

$$\mathbb{E}d(\mu_{t+1},\mu_{t+1}^{\bar{N}}) \leq (1 + L_\pi + L_p)\mathbb{E}d(\mu_t,\mu_t^{\bar{N}}) + \frac{|S|\sqrt{|A|}}{\sqrt{N}} \tag{3}$$

where $\frac{|S|\sqrt{|A|}}{\sqrt{N}} = \max_i\{\frac{|S^i|\sqrt{|A^i|}}{\sqrt{N_i}}\}_{i=1}^m$. Therefore,

$$\mathbb{E}d(\mu_t,\mu_t^{\bar{N}}) \leq (1 + L_\pi + L_p)^t\mathbb{E}d(\mu_0,\mu_0^{\bar{N}}) + M(t)\frac{|S|\sqrt{|A|}}{\sqrt{N}}$$

where $M(t) = \frac{(1+L_\pi+L_p)^t-1}{L_\pi+L_p}$.

We can also rewrite the reward functions using $\nu_t^i$ and $\nu_t^{i,\bar{N}}$ as:

$$J^i(\pi^1,\ldots,\pi^m) = \mathbb{E}\left[\sum_{t\geq 0}\gamma^t r^i(x_t^i,a_t^i,\mu_t)\right]$$

$$= \sum_{t\geq 0}\gamma^t\sum_x\mu_t^i(x)\sum_a\pi_t^i(a|x,\mu_t)r^i(x,a,\mu_t)$$

$$= \sum_{t\geq 0}\gamma^t\sum_{x,a}\nu_t^i(x,a)r^i(x,a,\mu_t)$$

and

$$J^{i,\bar{N}}(\pi^1,\ldots,\pi^m) = \mathbb{E}\left[\frac{1}{N_i}\sum_{j=1}^{N_i}\sum_{t\geq 0}\gamma^t r^i(x_t^{ij},a_t^{ij},\mu_t^{\bar{N}})\right]$$

$$= \sum_{t\geq 0}\gamma^t\sum_{x,a}\mathbb{E}\left[\nu_t^{i,\bar{N}}(x,a)r^i(x,a,\mu_t^{\bar{N}})\right].$$

Given a joint policy $(\pi^1, \ldots, \pi^m) \in \Pi^1 \times \cdots \times \Pi^m$, we have

$$|J^{i,\bar{N}}(\pi^1, \ldots, \pi^m) - J^i(\pi^1, \ldots, \pi^m)|$$

$$= |\sum_{t \geq 0} \gamma^t \sum_{x,a} \mathbb{E}\left[\nu_t^{i,\bar{N}}(x,a) r^i(x,a,\mu_t^{\bar{N}})\right] - \sum_{t \geq 0} \gamma^t \sum_{x,a} \nu_t^i(x,a) r^i(x,a,\mu_t)|$$

$$\leq |\sum_{t \geq 0} \gamma^t \sum_{x,a} \mathbb{E}\left[\nu_t^{i,\bar{N}}(x,a) r^i(x,a,\mu_t^{\bar{N}})\right] - \sum_{t \geq 0} \gamma^t \sum_{x,a} \mathbb{E}\left[\nu_t^{i,\bar{N}}(x,a) r^i(x,a,\mu_t)\right]|$$

$$+ |\sum_{t \geq 0} \gamma^t \sum_{x,a} \mathbb{E}\left[\nu_t^{i,\bar{N}}(x,a) r^i(x,a,\mu_t)\right] - \sum_{t \geq 0} \gamma^t \sum_{x,a} \nu_t^i(x,a) r^i(x,a,\mu_t)|$$

$$\leq \left|\sum_{t \geq 0} \gamma^t \mathbb{E} \sum_{x,a} \left[\nu_t^{i,\bar{N}}(x,a)\left(r^i(x,a,\mu_t^{\bar{N}}) - r^i(x,a,\mu_t)\right)\right]\right|$$

$$+ \sum_{t \geq 0} \gamma^t \sum_{x,a} C_r \mathbb{E}\left|\nu_t^{i,\bar{N}}(x,a) - \nu_t^i(x,a)\right|$$

$$\leq \sum_{t \geq 0} \gamma^t L_r \mathbb{E}d(\mu_t^{\bar{N}}, \mu_t) + \sum_{t \geq 0} \gamma^t C_r(1 + L_\pi)\mathbb{E}d(\mu_t^{\bar{N}}, \mu_t) + \sum_{t \geq 0} \gamma^t C_r |S^i|\sqrt{|A^i|}\frac{1}{\sqrt{N_i}}$$

$$\leq \sum_{t \geq 0} \gamma^t (L_r + C_r(1 + L_\pi))\mathbb{E}d(\mu_t^{\bar{N}}, \mu_t) + \sum_{t \geq 0} \gamma^t C_r |S^i|\sqrt{|A^i|}\frac{1}{\sqrt{N_i}}$$

$$\leq \sum_{t \geq 0} (L_r + C_r(1 + L_\pi))\gamma^t (1 + L_\pi + L_p)^t \mathbb{E}d(\mu_0^{\bar{N}}, \mu_0)$$

$$+ \sum_{t \geq 0} (L_r + C_r(1 + L_\pi))\gamma^t M(t)\frac{|S|\sqrt{|A|}}{\sqrt{N}} + \sum_{t \geq 0} \gamma^t C_r \frac{|S|\sqrt{|A|}}{\sqrt{N}}$$

When the discount factor $\gamma$ satisfies

$$\gamma(1 + L_\pi + L_p) < 1 \tag{4}$$

we have

$$\sum_{t \geq 0} |(L_r + C_r(1 + L_\pi))\gamma^t (1 + L_\pi + L_p)^t < \infty$$

$$\sum_{t \geq 0} (L_r + C_r(1 + L_\pi))\gamma^t M(t) < \infty, \quad \sum_{t \geq 0} \gamma^t C_r < \infty$$

Thus,

$$|J^{i,\bar{N}}(\pi^1, \ldots, \pi^m) - J^i(\pi^1, \ldots, \pi^m)| \leq M\frac{|S|\sqrt{|A|}}{\sqrt{N}} \tag{5}$$

where

$$M = \sum_{t \geq 0} (L_r + C_r(1 + L_\pi))\gamma^t (1 + L_\pi + L_p)^t$$

$$+ \sum_{t \geq 0} (L_r + C_r(1 + L_\pi))\gamma^t M(t) + \sum_{t \geq 0} \gamma^t C_r$$

is finite.

Let $(\pi_*^1, \ldots, \pi_*^m) \in \Pi^1 \times \cdots \times \Pi^m$ be a Nash equilibrium for the mean-field type game and $\tilde{\pi}^i$ be the policy for an agent in coalition $i$ of the finite-population $m$-coalition game such that

$$J^{i,\bar{N}}(\tilde{\pi}^i; \pi_*^{-i}) = \max_{\pi^i \in \Pi^i} J^{i,\bar{N}}(\pi^i; \pi_*^{-i}),$$

we have

$$
\begin{aligned}
J^{i,\bar{N}}(\tilde{\pi}^i; \pi_*^{-i}) - J^{i,\bar{N}}(\pi_*^i; \pi_*^{-i}) &= J^{i,\bar{N}}(\tilde{\pi}^i; \pi_*^{-i}) - J^i(\tilde{\pi}^i; \pi_*^{-i}) \\
&\quad + J^i(\tilde{\pi}^i; \pi_*^{-i}) - J^i(\pi_*^i; \pi_*^{-i}) \\
&\quad + J^i(\pi_*^i; \pi_*^{-i}) - J^{i,\bar{N}}(\pi_*^i; \pi_*^{-i}) \\
&\leq |J^{i,\bar{N}}(\tilde{\pi}^i; \pi_*^{-i}) - J^i(\tilde{\pi}^i; \pi_*^{-i})| \\
&\quad + |J^i(\pi_*^i; \pi_*^{-i}) - J^{i,\bar{N}}(\pi_*^i; \pi_*^{-i})| \\
&\leq \frac{2M|S|\sqrt{|A|}}{\sqrt{N}}
\end{aligned}
$$

The last two inequalities are due to the definition of $\pi_*^i$ and (5). $\qquad\square$

## B  CONNECTION BETWEEN MFTG AND STAGE-GAME NASH EQUILIBRIA

We prove Proposition 2.10.

**Proof: Proof of $\Leftarrow$:** If (ii) is true, without loss of generality, we consider player $i$. we have for $\bar{s} \in \bar{S}$,

$$
\bar{v}_{\bar{\boldsymbol{\pi}}_*}^i(\bar{s}) \geq \bar{\pi}_*^1(\bar{s}) \cdots \bar{\pi}_*^{i-1}(\bar{s}) \bar{\pi}^i(\bar{s}) \bar{\pi}_*^{i+1}(\bar{s}) \cdots \bar{\pi}_*^m(\bar{s}) \bar{Q}_{\bar{\boldsymbol{\pi}}}^i(\bar{s})
$$

$$
= \bar{r}^i(\bar{s}, \bar{\pi}^i(\bar{s})) + \gamma \int_{\bar{S}} \int_{\bar{\boldsymbol{A}}} \bar{p}(\mathrm{d}\bar{s}'|\bar{s}, \bar{a}^1, \dots, \bar{a}^m) \bar{\pi}_*^1(\mathrm{d}\bar{a}^1|\bar{s}) \cdots \bar{\pi}^i(\mathrm{d}\bar{a}^i|\bar{s}) \cdots \bar{\pi}_*^m(\mathrm{d}\bar{a}^m|\bar{s}) \bar{v}_{\bar{\boldsymbol{\pi}}_*}^i(\bar{s}')
$$

By iteration and substituting $\bar{v}_{\bar{\boldsymbol{\pi}}_*}^i(\bar{s}')$ with the above inequality, we have

$$
\bar{v}_{\bar{\boldsymbol{\pi}}_*}^i(\bar{s}) \geq \bar{v}_{\bar{\boldsymbol{\pi}}'}^i(\bar{s})
$$

for all $\bar{\pi}^i \in \bar{\Pi}^i$, where $\bar{\boldsymbol{\pi}}' = (\bar{\pi}_*^1, \dots, \bar{\pi}^i, \dots \bar{\pi}_*^m)$. Since $i$ is arbitrary, by the definition of Nash equilibrium, we have $(\bar{\pi}_*^1, \dots, \bar{\pi}_*^m)$ is a Nash equilibrium for the MFTG.

**Proof of $\Rightarrow$:** If (i) is true, then $\bar{\pi}_*^i$ is also the optimal policy for the MDP($\bar{\boldsymbol{\pi}}_*^{-i}$). For each $\bar{s}$, $\bar{\pi}_*^i(\bar{s})$ maximizes

$$
\bar{r}_{\bar{\boldsymbol{\pi}}^{-i}}(\bar{s}, \bar{a}^i) + \gamma \int_{\bar{S}} \bar{p}_{\bar{\boldsymbol{\pi}}^{-i}}(\mathrm{d}\bar{s}'|\bar{s}, \bar{a}^i) \bar{v}_{\bar{\boldsymbol{\pi}}^i}^i(\bar{s}') \tag{6}
$$

So $\bar{\pi}_*^i(\bar{s})$ is the best response of player $i$ in stage game $(\bar{Q}_{\bar{\boldsymbol{\pi}}_*}^1(\bar{s}), \dots, \bar{Q}_{\bar{\boldsymbol{\pi}}_*}^m(\bar{s}))$. The result also applies to other players, so $(\bar{\pi}_*^1(\bar{s}), \dots, \bar{\pi}_*^m(\bar{s}))$ is a Nash equilibrium in the stage game $(\bar{Q}_{\bar{\boldsymbol{\pi}}_*}^1(\bar{s}), \dots, \bar{Q}_{\bar{\boldsymbol{\pi}}_*}^m(\bar{s}))$. $\qquad\square$

## C  ANALYSIS OF DISCRETIZED NASHQ LEARNING

We now prove Theorem 3.2.

**Proof:** Let $\check{\boldsymbol{\pi}}_*^p$ be a unique pure policy for the discretized MFTG such that for each $i$ and $\check{s} \in \check{S}$, the payoff function $v_{\check{\boldsymbol{\pi}}_*^p}^i(\check{s})$ is a global optimal point for the stage game $\check{Q}_{\check{\boldsymbol{\pi}}_*^p}^i(\check{s})$.

$$
|\check{Q}_t^i(\mathrm{Proj}_{\check{S}}(\bar{s}), \mathrm{Proj}_{\check{A}^1}(\bar{a}^1), \dots, \mathrm{Proj}_{\check{A}^m}(\bar{a}^m)) - \bar{Q}_{\bar{\boldsymbol{\pi}}_*^p}^i(\bar{s}, \bar{a}^1 \dots \bar{a}^m)|
$$

$$
\leq |\check{Q}_t^i(\mathrm{Proj}_{\check{S}}(\bar{s}), \mathrm{Proj}_{\check{A}^1}(\bar{a}^1), \dots, \mathrm{Proj}_{\check{A}^m}(\bar{a}^m)) - \check{Q}_{\check{\boldsymbol{\pi}}_*^p}^i(\mathrm{Proj}_{\check{S}}(\bar{s}), \mathrm{Proj}_{\check{A}^1}(\bar{a}^1), \dots, \mathrm{Proj}_{\check{A}^m}(\bar{a}^m))|
$$

$$
+ |\check{Q}_{\check{\boldsymbol{\pi}}_*^p}^i(\mathrm{Proj}_{\check{S}}(\bar{s}), \mathrm{Proj}_{\check{A}^1}(\bar{a}^1), \dots, \mathrm{Proj}_{\check{A}^m}(\bar{a}^m)) - \bar{Q}_{\bar{\boldsymbol{\pi}}_*^p}^i(\mathrm{Proj}_{\check{S}}(\bar{s}), \mathrm{Proj}_{\check{A}^1}(\bar{a}^1), \dots, \mathrm{Proj}_{\check{A}^m}(\bar{a}^m))|
$$

$$
+ |\bar{Q}_{\bar{\boldsymbol{\pi}}_*^p}^i(\mathrm{Proj}_{\check{S}}(\bar{s}), \mathrm{Proj}_{\check{A}^1}(\bar{a}^1), \dots, \mathrm{Proj}_{\check{A}^m}(\bar{a}^m)) - \bar{Q}_{\bar{\boldsymbol{\pi}}_*^p}^i(\bar{s}, \bar{a}^1 \dots \bar{a}^m)|
$$

$$
\tag{7}
$$

From Theorem 3.1, when $t$ is large enough, we have

$$
|\check{Q}_t^i(\mathrm{Proj}_{\check{S}}(\bar{s}), \mathrm{Proj}_{\check{A}^1}(\bar{a}^1), \dots, \mathrm{Proj}_{\check{A}^m}(\bar{a}^m)) - \check{Q}_{\check{\boldsymbol{\pi}}_*^p}^i(\mathrm{Proj}_{\check{S}}(\bar{s}), \mathrm{Proj}_{\check{A}^1}(\bar{a}^1), \dots, \mathrm{Proj}_{\check{A}^m}(\bar{a}^m))| < \epsilon. \tag{8}
$$

We now consider the second term on the RHS of (7). Using the notation

$$
(\mathrm{Proj}_{\check{S}}(\bar{s}), \mathrm{Proj}_{\check{A}^1}(\bar{a}^1), \dots, \mathrm{Proj}_{\check{A}^m}(\bar{a}^m)) = (\check{s}, \check{a}^1, \dots, \check{a}^m).
$$

and

$$\check{F}(\check{s}, \check{a}^1, \ldots, \check{a}^m) = \text{Proj}(\bar{F}(\check{s}, \check{a}^1, \ldots, \check{a}^m))$$

then we have

$$
\begin{aligned}
&|\check{Q}^i_{\check{\boldsymbol{\pi}}^p_*}(\check{s}, \check{a}^1, \ldots, \check{a}^m) - \bar{Q}^i_{\bar{\boldsymbol{\pi}}^p_*}(\check{s}, \check{a}^1, \ldots, \check{a}^m)| \\
&\leq \gamma \mathbb{E}[v^i_{\check{\boldsymbol{\pi}}^p_*}(\check{F}(\check{s}, \check{a}^1, \ldots, \check{a}^m)) - v^i_{\bar{\boldsymbol{\pi}}^p_*}(\bar{F}(\check{s}, \check{a}^1, \ldots, \check{a}^m))] \\
&\leq \gamma \mathbb{E}[v^i_{\check{\boldsymbol{\pi}}^p_*}(\check{F}(\check{s}, \check{a}^1, \ldots, \check{a}^m)) - v^i_{\bar{\boldsymbol{\pi}}^p_*}(\check{F}(\check{s}, \check{a}^1, \ldots, \check{a}^m))] \\
&\quad + \gamma \mathbb{E}[v^i_{\bar{\boldsymbol{\pi}}^p_*}(\check{F}(\check{s}, \check{a}^1, \ldots, \check{a}^m)) - v^i_{\bar{\boldsymbol{\pi}}^p_*}(\bar{F}(\check{s}, \check{a}^1, \ldots, \check{a}^m))] \\
&\leq \gamma \mathbb{E}[v^i_{\check{\boldsymbol{\pi}}^p_*}(\check{F}(\check{s}, \check{a}^1, \ldots, \check{a}^m)) - v^i_{\bar{\boldsymbol{\pi}}^p_*}(\check{F}(\check{s}, \check{a}^1, \ldots, \check{a}^m))] + \gamma L_{\bar{v}_{\bar{\boldsymbol{\pi}}_*}} \epsilon_S \\
&\leq \gamma \mathbb{E}[|\text{Nash}\check{Q}^i_{\check{\boldsymbol{\pi}}^p_*}(\check{F}(\check{s}, \check{a}^1, \ldots, \check{a}^m)) - \text{Nash}\bar{Q}^i_{\bar{\boldsymbol{\pi}}^p_*}(\check{F}(\check{s}, \check{a}^1, \ldots, \check{a}^m))|] + \gamma L_{\bar{v}_{\bar{\boldsymbol{\pi}}_*}} \epsilon_S
\end{aligned}
\tag{9}
$$

where we used the assumption that $\bar{v}^i_{\boldsymbol{\pi}_*}$ is Lipschitz continuous w.r.t. $\bar{s}$ with constant $L_{\bar{v}_{\boldsymbol{\pi}_*}}$. Namely,

$$|\bar{v}^i_{\boldsymbol{\pi}_*}(\bar{s}) - \bar{v}^i_{\boldsymbol{\pi}_*}(\bar{s}')| \leq L_{\bar{v}_*} d_{\bar{S}}(\bar{s}, \bar{s}')$$

Let $\check{F}(\check{s}, \check{a}^1, \ldots, \check{a}^m) = \check{s}'$, and $(\bar{a}^1_*, \ldots, \bar{a}^m_*)$, $(\check{a}^1_*, \ldots, \check{a}^m_*)$ such that

$$\text{Nash}\check{Q}^i_{\check{\boldsymbol{\pi}}^p_*}(\check{F}(\check{s}, \check{a}^1, \ldots, \check{a}^m)) = \check{Q}^i_{\check{\boldsymbol{\pi}}^p_*}(\check{s}', \check{a}^1_*, \ldots, \check{a}^m_*)$$
$$\text{Nash}\bar{Q}^i_{\bar{\boldsymbol{\pi}}^p_*}(\check{F}(\check{s}, \check{a}^1, \ldots, \check{a}^m)) = \bar{Q}^i_{\bar{\boldsymbol{\pi}}^p_*}(\check{s}', \bar{a}^1_*, \ldots, \bar{a}^m_*)$$

consider the term

$$
\begin{aligned}
&\check{Q}^i_{\check{\boldsymbol{\pi}}^p_*}(\check{s}', \check{a}^1_*, \ldots, \check{a}^m_*) - \bar{Q}^i_{\bar{\boldsymbol{\pi}}^p_*}(\check{s}', \bar{a}^1_*, \ldots, \bar{a}^m_*) \\
&= \check{Q}^i_{\check{\boldsymbol{\pi}}^p_*}(\check{s}', \check{a}^1_*, \ldots, \check{a}^m_*) - \check{Q}^i_{\check{\boldsymbol{\pi}}^p_*}(\check{s}', \text{Proj}_{\check{A}^1}(\bar{a}^1_*), \ldots, \text{Proj}_{\check{A}^m}(\bar{a}^m_*)) \\
&\quad + \check{Q}^i_{\check{\boldsymbol{\pi}}^p_*}(\check{s}', \text{Proj}_{\check{A}^1}(\bar{a}^1_*), \ldots, \text{Proj}_{\check{A}^m}(\bar{a}^m_*)) - \bar{Q}^i_{\bar{\boldsymbol{\pi}}^p_*}(\check{s}', \text{Proj}_{\check{A}^1}(\bar{a}^1_*), \ldots, \text{Proj}_{\check{A}^m}(\bar{a}^m_*)) \\
&\quad + \bar{Q}^i_{\bar{\boldsymbol{\pi}}^p_*}(\check{s}', \text{Proj}_{\check{A}^1}(\bar{a}^1_*), \ldots, \text{Proj}_{\check{A}^m}(\bar{a}^m_*)) - \bar{Q}^i_{\bar{\boldsymbol{\pi}}^p_*}(\check{s}', \bar{a}^1_*, \ldots, \bar{a}^m_*) \\
&\geq -||\check{Q}^i_{\check{\boldsymbol{\pi}}^p_*} - \bar{Q}^i_{\bar{\boldsymbol{\pi}}^p_*}||_\infty + \bar{r}^i(\check{s}', \text{Proj}_{\check{A}^i}(\bar{a}^i_*)) - \bar{r}^i(\check{s}', \bar{a}^i_*) \\
&\quad + \gamma \mathbb{E}v^i_{\check{\boldsymbol{\pi}}^p_*}(\bar{F}(\check{s}', \text{Proj}_{\check{A}^1}(\bar{a}^1_*), \ldots, \text{Proj}_{\check{A}^m}(\bar{a}^m_*))) - \gamma \mathbb{E}v^i_{\bar{\boldsymbol{\pi}}^p_*}(\bar{F}(\check{s}', \bar{a}^1_*, \ldots, \bar{a}^m_*)) \\
&\geq -||\check{Q}^i_{\check{\boldsymbol{\pi}}^p_*} - \bar{Q}^i_{\bar{\boldsymbol{\pi}}^p_*}||_\infty - L_{\bar{r}^i} d(\bar{a}^i_*, \text{Proj}_{\check{A}^i}(\bar{a}^i_*)) - \gamma L_{\bar{v}^i_{\bar{\boldsymbol{\pi}}_*}} L_{\bar{F}} \sum_{i=1}^m d(\bar{a}^i_*, \text{Proj}_{\check{A}^i}(\bar{a}^i_*))
\end{aligned}
\tag{10}
$$

the last inequality is due to the Lipschitz continuous assumptions on $\bar{r}^i$ and $\bar{F}$. Namely,

$$|\bar{r}^i(\bar{s}, \bar{a}^i) - \bar{r}^i(\bar{s}', \bar{a}'^i)| \leq L_{\bar{r}^i}\left(d_{\bar{S}}(\bar{s}, \bar{s}') + d_{\bar{A}^i}(\bar{a}^i, \bar{a}'^i)\right)$$

and

$$\mathbb{E}|\bar{F}(\bar{s}, \bar{a}^1, \ldots, \bar{a}^m) - \bar{F}(\bar{s}', \bar{a}'^1, \ldots, \bar{a}'^m)| \leq L_{\bar{F}}\left(d_{\bar{S}}(\bar{s}, \bar{s}') + \sum_{i \in [m]} d_{\bar{A}^i}(\bar{a}^i, \bar{a}'^i)\right)$$

On the other hand,

$$
\begin{aligned}
&\check{Q}^i_{\check{\boldsymbol{\pi}}^p_*}(\check{s}', \check{a}^1_*, \ldots, \check{a}^m_*) - \bar{Q}^i_{\bar{\boldsymbol{\pi}}^p_*}(\check{s}', \bar{a}^1_*, \ldots, \bar{a}^m_*) \\
&= \check{Q}^i_{\check{\boldsymbol{\pi}}^p_*}(\check{s}', \check{a}^1_*, \ldots, \check{a}^m_*) - \bar{Q}^i_{\bar{\boldsymbol{\pi}}^p_*}(\check{s}', \check{a}^1_*, \ldots, \check{a}^m_*) + \bar{Q}^i_{\bar{\boldsymbol{\pi}}^p_*}(\check{s}', \check{a}^1_*, \ldots, \check{a}^m_*) - \bar{Q}^i_{\bar{\boldsymbol{\pi}}^p_*}(\check{s}', \bar{a}^1_*, \ldots, \bar{a}^m_*) \\
&\leq ||\check{Q}^i_{\check{\boldsymbol{\pi}}^p_*} - \bar{Q}^i_{\bar{\boldsymbol{\pi}}^p_*}||_\infty
\end{aligned}
\tag{11}
$$

Thus, we have

$$
\begin{aligned}
&|\check{Q}^i_{\check{\boldsymbol{\pi}}^p_*}(\check{s}', \check{a}^1_*, \ldots, \check{a}^m_*) - \bar{Q}^i_{\bar{\boldsymbol{\pi}}^p_*}(\check{s}', \bar{a}^1_*, \ldots, \bar{a}^m_*)| \\
&\leq \gamma(||\check{Q}^i_{\check{\boldsymbol{\pi}}^p_*} - \bar{Q}^i_{\bar{\boldsymbol{\pi}}^p_*}||_\infty + L_{\bar{r}^i}\epsilon_A + \gamma L_{\bar{v}^i_{\bar{\boldsymbol{\pi}}_*}} L_{\bar{F}} m\epsilon_A) + \gamma L_{\bar{v}_{\bar{\boldsymbol{\pi}}_*}} \epsilon_S
\end{aligned}
\tag{12}
$$

Therefore, we have

$$||\check{Q}^i_{\check{\boldsymbol{\pi}}^p_*} - \bar{Q}^i_{\bar{\boldsymbol{\pi}}^p_*}||_\infty \leq \frac{\gamma}{1 - \gamma}\left(L_{\bar{r}^i}\epsilon_A + \gamma L_{\bar{v}^i_{\bar{\boldsymbol{\pi}}_*}} L_{\bar{F}} m\epsilon_A + L_{\bar{v}_{\bar{\boldsymbol{\pi}}_*}} \epsilon_S\right)
\tag{13}$$

For the last term on the RHS of (7), we have

$$
|\bar{Q}^i_{\boldsymbol{\pi}^p_*}(\mathrm{Proj}_{\breve{S}}(\bar{s}), \mathrm{Proj}_{\breve{A}^1}(\bar{a}^1), \ldots, \mathrm{Proj}_{\breve{A}^m}(\bar{a}^m)) - \bar{Q}^i_{\bar{\boldsymbol{\pi}}^p_*}(\bar{s}, \bar{a}^1 \ldots \bar{a}^m)|
$$

$$
\leq |\bar{r}^i(\mathrm{Proj}_{\breve{S}}(\bar{s}), \mathrm{Proj}_{\breve{A}^i}(\bar{a}^i)) - \bar{r}^i(\bar{s}, \bar{a}^i)|
$$

$$
+ \gamma \mathbb{E}[\bar{v}^i_{\boldsymbol{\pi}^p_*}(\bar{F}(\mathrm{Proj}_{\breve{S}}(\bar{s}), \mathrm{Proj}_{\breve{A}^1}(\bar{a}^1), \ldots, \mathrm{Proj}_{\breve{A}^m}(\bar{a}^m))) - \bar{v}^i_{\boldsymbol{\pi}^p_*}(\bar{F}(\bar{s}, \bar{a}^1 \ldots \bar{a}^m))]
$$

$$
\leq L_{\bar{r}^i}(d_{\bar{S}}(\mathrm{Proj}_{\breve{S}}(\bar{s}), \bar{s}) + d_{\bar{A}^i}(\mathrm{Proj}_{\breve{A}^i}(\bar{a}^i), \bar{a}^i))
$$

$$
+ \gamma L_{\bar{v}^i_{\bar{\boldsymbol{\pi}}_*}} \mathbb{E}(\bar{F}(\mathrm{Proj}_{\breve{S}}(\bar{s}), \mathrm{Proj}_{\breve{A}^1}(\bar{a}^1), \ldots, \mathrm{Proj}_{\breve{A}^m}(\bar{a}^m)) - \bar{F}(\bar{s}, \bar{a}^1 \ldots \bar{a}^m))
$$

$$
\leq L_{\bar{r}^i}(\epsilon_S + \epsilon_A) + \gamma L_{\bar{v}^i_{\bar{\boldsymbol{\pi}}_*}} L_{\bar{F}}(\epsilon_S + m\epsilon_A)
$$

(14)

Finally, we get the result by combining inequalities (8), (13), and (14) together. $\qquad\square$

# D  PSEUDO-CODES FOR THE MAIN ALGORITHMS

Algorithm 1 shows the DNashQ-MFTG algorithm. Algorithm 2 shows the DDPG-MFTG algorithm.

---

**Algorithm 1** Discretized Nash Q-learning for Mean Field Type Game (**DNashQ-MFTG**)

---

1: **Inputs:** A series of learning rates $\alpha_t \in (0,1)$, $t \geq 0$, and exploration levels $\epsilon_t$, $t \geq 0$
2: **Outputs:** Nash Q-functions $\check{Q}_N^i$ for $i = 1, \ldots, m$
3: Initialization: $\check{Q}_{0,0}^i(\check{s}, \check{a}^1, \ldots, \check{a}^m) = 0$ for all $\check{s} \in \check{S}$ and $\check{a}^i \in \check{A}^i$;
4: **for** $k = 0, 1, \ldots, N-1$ **do**
5:    Initialize state $\check{s}_0$
6:    **for** $t = 0, \ldots, T-1$ **do**
7:       Generate a random number $\zeta_t \sim \mathcal{U}[0,1]$
8:       **if** $\zeta_t \geq \epsilon_t$ **then**
9:          Solve the stage game $\check{Q}_{k,t}^i(\check{s}_t)$ and get strategy profile $(\check{\pi}_*^{i,1}, \ldots, \check{\pi}_*^{i,m})$ for $i = 1, \ldots, m$
10:          Sample $\check{a}_t^i \sim \check{\pi}_*^{i,i}$ for $i = 1, \ldots, m$
11:       **else**
12:          Sample action $\check{a}_t^i$ uniformly from $\check{A}^i$ for $i = 1, \ldots, m$
13:       **end if**
14:       Observe $r_t^1, \ldots, r_t^m$, $\check{a}_t^1, \ldots, \check{a}_t^m$, and $\check{s}_{t+1} = \mathrm{Proj}_{\check{S}}(\bar{F}(\check{s}_t, \check{a}_t^1, \ldots, \check{a}_t^m))$
15:       Solve the stage game $\check{Q}_{k,t}^i(\check{s}_{t+1})$ and get strategy profile $(\check{\pi}_*^{'i,1}, \ldots, \check{\pi}_*^{'i,m})$ for $i = 1, \ldots, m$
16:       Compute $\mathrm{Nash}\check{Q}_{k,t}^i(\check{s}_{t+1}) = \check{\pi}_*^{'i,1} \ldots \check{\pi}_*^{'i,m} \check{Q}_{k,t}^i(\check{s}_{t+1})$
17:       Copy $\check{Q}_{k,t+1}^i = \check{Q}_{k,t}^i$ for $i = 1, \ldots, m$ and update $\check{Q}_{k,t+1}^i$ by:
$$\check{Q}_{k,t+1}^i(\check{s}_t, \check{a}^1, \ldots, \check{a}^m) = (1-\alpha_t)\check{Q}_{k,t}^i(\check{s}_t, \check{a}^1, \ldots, \check{a}^m) + \alpha_t(r_t^i + \beta \mathrm{Nash}\check{Q}_{k,t}^i(\check{s}_{t+1}))$$
18:    **end for**
19:    Copy $\check{Q}_{k+1,0}^i = \check{Q}_{k,T-1}^i$ for $i = 1, \ldots, m$
20: **end for**

---

---

**Algorithm 2** DDPG for MFTG

---

1: **Inputs:** A number of episodes $N$; a length $T$ for each episode; a minibatch size $N_{\text{batch}}$; a learning rate $\tau$.

2: **Outputs:** Policy functions for each central player represented by $\pi^i_{\omega_i}$.

3: Initialize parameters $\theta_i$ and $\omega_i$ for critic networks $Q^i_{\theta_i}$ and actor networks $\pi^i_{\omega_i}$, $i = 1, ..., m$

4: Initialize $\theta'_i \leftarrow \theta_i$ and $\omega'_i \leftarrow \omega_i$ for target networks $Q^{i'}_{\theta'_i}$ and $\pi^{i'}_{\omega'_i}$, $i = 1, ..., m$

5: Initialize replay buffer $R_{\text{buffer}}$

6: **for** $k = 0, 1, ..., N - 1$ **do**

7:     Initialize distribution $\bar{s}_0$

8:     **for** $t = 0, 1, \ldots, T - 1$ **do**

9:         Select actions $\bar{a}^i_t = \pi^i_{\omega_i}(\bar{s}_t) + \epsilon_t$, where $\epsilon_t$ is the exploration noise, for $i = 1, ..., m$

10:         Execute $\bar{a}^i_t$, observe reward $\bar{r}^i(\bar{s}_t, \bar{a}^i_t)$, for $i = 1, ..., m$

11:         Observe $\bar{s}_{t+1}$

12:         Store transition $(\bar{s}_t, \bar{a}^1_t, ..., \bar{a}^m_t, \bar{r}^1_t, ..., \bar{r}^m_t, \bar{s}_{t+1})$ in $R_{\text{buffer}}$

13:         Sample a random minibatch of $N_{\text{batch}}$ transitions $(\bar{s}_j, \bar{a}^1_j, ..., \bar{a}^m_j, \bar{r}^1_j, ..., \bar{r}^m_j, \bar{s}_{j+1})$ from $R_{\text{buffer}}$

14:         Set $y^i_j = \bar{r}^i_j + \gamma Q^{i'}_{\theta'_i}(\bar{s}_{j+1}, \pi^{i'}_{\omega'_i}(\bar{s}_{j+1}))$ for $i = 1, ..., m$, $j = 1, ..., N_{\text{batch}}$

15:         Update the critic networks by minimizing the loss: $L^i(\theta_i) = \frac{1}{N_{\text{batch}}} \sum_j (y^i_j - Q^i_{\theta_i}(\bar{s}_j, \bar{a}^i_j))^2$, for $i = 1, ..., m$

16:         Update the actor policies using the sampled policy gradients $\nabla_{\omega_i} v^i$, for $i = 1, ..., m$:

$$\nabla_{\omega_i} v^i(\omega_i) \approx \frac{1}{N_{\text{batch}}} \sum_j \nabla_{\bar{a}^i} Q^i_{\theta_i}(\bar{s}_j, \pi^i_{\omega_i}(\bar{s}_j)) \nabla_{\omega_i} \pi^i_{\omega_i}(\bar{s}_j)$$

17:         Update target networks: $\theta'_i \leftarrow \tau\theta_i + (1 - \tau)\theta'_i$, $\omega'_i \leftarrow \tau\omega_i + (1 - \tau)(\omega'_i)$, for $i = 1, ..., m$.

18:     **end for**

19: **end for**

---

## E  PSEUDO-CODES FOR THE EVALUATION METRICS

In this section, we present pseudo-codes used for evaluation.

- Algorithm 3 shows how to do the inference of DNash-MFTG given the Q-functions of agents.

- Algorithm 4 explains the way to evaluate policies.

- Algorithm 5 presents the general structure of computing exploitability.

- Algorithm 6 presents a detailed version of computing the exploitability.

---

**Algorithm 3** DNashQ-MFTG inference

1: **Inputs:** Nash Q-functions $\check{Q}_N^i$ for $i = 1, \ldots, m$; number of steps $T$
2: **Outputs:** $\upsilon^i = (\check{s}_0^i, \check{a}_0^i, r_0^i, \ldots, \check{s}_{T-1}^i, \check{a}_{T-1}^i, r_{T-1}^i)$ for $i = 1, \ldots, m$
3: Initialize $\check{s}_0$ and trajectory $\upsilon^i$
4: **for** $t = 0, \ldots, T - 1$ **do**
5:     Solve the stage game $\check{Q}_N^i(\check{s}_t)$ and get strategy profile $(\check{\pi}_*^{i,1}, \ldots, \check{\pi}_*^{i,m})$ for $i = 1, \ldots, m$
6:     Sample $\check{a}_t^i \sim \check{\pi}_*^{i,i}$ for $i = 1, \ldots, m$
7:     Observe $r_t^1, \ldots, r_t^m$ and $\check{s}_{t+1} = \text{Proj}_{\check{S}}(\bar{F}(\check{s}_t, \check{a}_t^1, \ldots, \check{a}_t^m))$
8:     Store $(\check{s}_t^i, \check{a}_t^i, r_t^i)$ to $\upsilon^i$
9: **end for**
10: **return** Trajectory $\upsilon^i$

---

**Algorithm 4** Policies evaluation

1: **Inputs:** Policy profile $\bar{\pi} = (\bar{\pi}^1, \ldots, \bar{\pi}^m)$, testing set of initial distributions $\mathcal{D}_{\text{test}}$
2: **Outputs:** Values $J^i(\bar{\pi})$
3: Initialize $V^i = 0$, $i = 1, \ldots, m$
4: **for** $\mu_0 \in \mathcal{D}_{\text{test}}$ **do**
5:     Run an episode starting from initial distribution $\mu_0$ and using policies $\bar{\pi}$
6:     Let $V_{\mu_0}^i$ be the total reward, $i = 1, \ldots, m$
7:     Let $V^i = V^i + V_{\mu_0}^i$, $i = 1, \ldots, m$
8: **end for**
9: Let $J^i = \frac{1}{|\mathcal{D}_{\text{test}}|} V^i$
10: Return $J^i$, $i = 1, \ldots, m$

---

**Algorithm 5** Exploitability computation

1: **Inputs:** Policy profile $\bar{\pi} = (\bar{\pi}^1, \ldots, \bar{\pi}^m)$, training set of initial distributions $\mathcal{D}_{\text{train}}$, testing set of initial distributions $\mathcal{D}_{\text{test}}$
2: **Outputs:** Exploitabilities $E^i(\bar{\pi})$, $i = 1, \ldots, m$
3: **for** $i = 1, \ldots, m$ **do**
4:     Compute BR $\bar{\pi}^{i*} = \arg\max_{\tilde{\bar{\pi}}^i} J^i(\tilde{\bar{\pi}}^i; \bar{\pi}^{-i})$ using RL with testing set $\mathcal{D}_{\text{test}}$
5:     Compute $M^i = J^i(\bar{\pi}^{i*}; \bar{\pi}^{-i})$ using Algo. 4 with policy profile $(\bar{\pi}^{i*}; \bar{\pi}^{-i})$ and $\mathcal{D}_{\text{test}}$
6:     Compute $V^i = J^i(\bar{\pi}^i; \bar{\pi}^{-i})$ using Algo. 4 with policy profile $(\bar{\pi}^i; \bar{\pi}^{-i})$ and $\mathcal{D}_{\text{test}}$
7:     Let $E^i = M^i - V^i$
8: **end for**
9: Return $E^i$, $i = 1, \ldots, m$

---

**Algorithm 6** Exploitability computation

---

1: **Inputs:** Policy profile $\bar{\pi} = (\bar{\pi}^1, \ldots, \bar{\pi}^m)$, testing set of initial distributions $\mathcal{D}_{\text{test}}$,
2: **Outputs:** Exploitabilities $E^i(\bar{\pi})$, $i = 1, \ldots, m$
3: Initialize $M^i = 0$, $E^i = 0$, $i = 1, \ldots, m$
4: **for** $i = 1, \ldots, m$ **do**
5:     **for** $\mu_0$ in $\mathcal{D}_{\text{test}}$ **do**
6:         Initialize replay buffer and optimizers
7:         **for** $j = 1, \ldots, N$ **do**
8:             Compute BR $\bar{\pi}_j^{i*} = \arg\max_{\tilde{\bar{\pi}}^i} J^i(\tilde{\bar{\pi}}^i; \bar{\pi}^{-i})$ using RL with the initial distribution $\mu_0$
9:             Compute $M_j^i = J^i(\bar{\pi}_j^{i*}; \bar{\pi}^{-i})$ using Algo. 4 with policy profile $(\bar{\pi}_j^{i*}; \bar{\pi}^{-i})$ and $\mu_0$
10:             $M^i = M^i + M_j^i$
11:         **end for**
12:         $M^i = M^i/N$
13:         Compute $V^i = J^i(\bar{\pi}^i; \bar{\pi}^{-i})$ using Algo. 4 with policy profile $(\bar{\pi}^i; \bar{\pi}^{-i})$ and $\mu_0$
14:         $E^i = E^i + M^i - V^i$
15:     **end for**
16:     $E^i = \frac{1}{|\mathcal{D}_{\text{test}}|} E^i$
17: **end for**
18: Return $E^i$, $i = 1, \ldots, m$

---

# F  DETAILS ON NUMERICAL EXPERIMENTS

## F.1  EXAMPLE 1: 1D TARGET MOVING GRID GAME

**Model.**   The model is as follows:

- **Number of populations:** $m = 2$.
- **State space:** $S^i = S = \{1, 2, 3, \ldots, G\}$ for $i = 1, 2$, which represents locations.
- **Action space:** $A^i = A = \{0, -1, 1\}$ for $i = 1, 2$, represents the agent will stay, move left, or move right respectively
- **Individual dynamics:** $x_{t+1}^i = x_t^i + a_t^i + \xi_t^i$, where $(\xi_t^i)_{n \geq 1}$ is a sequence of i.i.d. random variables taking values in $A^i$ and sampled from a predefined distribution as noises. We use periodic boundary conditions, meaning that agents who move left (resp. right) while in the $0$ (resp. $G$) state end up on the other side, at the $G$ (resp. $0$) state.
- **Mean-field transitions:** we can formulate the element in $k$-th row, $\ell$-th column in the $G \times G$ transition matrix $\bar{P}^i(\bar{s}_t^i, \bar{a}_t^i)$ is equal to $p^i(\bar{s}_{t+1}^i = k | \bar{s}_t^i = \ell, \bar{a}_t^i, \xi_t^i)$
- **Rewards:** Population 1 receives a high penalty when it moves, while Population 2 tries to match with Population 1's current position. We use the following rewards:

$$\bar{r}^1(\bar{s}, \bar{a}_t^1) = -c_1(\|\bar{a}_{\text{stay}}^1 - \bar{a}_t^1\|_2) - c_2(\bar{s}^1 \times \bar{s}^2), \quad \bar{r}^2(\bar{s}) = -c_1(\|\bar{s}^1 - \bar{s}^2\|_2)$$

where $c_1 = 1000$ and $c_2 = 10$. As a consequence, we expect that, at Nash equilibrium, Coalition 1 stays where it is but also tries to avoid coalition 2, while Coalition 2 matches Coalition 1 perfectly.

**Training and testing sets.**   In this example, we use $G = 3$ points in the 1D grid. (Scaling up to larger spaces would require a huge amount of memory due to the required discretization of the state space. This motivates the deep RL algorithm we use in the next examples.) We use the following sets of initial distributions for training and testing.

- Training distributions: We employ a random sampling technique to generate the training distribution at the beginning of each training episode. Specifically, we first sample each element in the state matrix from a uniform distribution over the interval $[0, 1)$ and then divide each element by the total sum of the matrix to normalize it.
- Testing distributions: we use the following pairs:
  $\mathcal{D}_{\text{test}} = \{((1.0, 0.0, 0.0), (0.0, 0.0, 1.0)), ((0.0, 0.0, 1.0), (1.0, 0.0, 0.0)),$
  $((0.0, 1.0, 0.0), (0.0, 1.0, 0.0))\}$

**Parameters and Hyper-parameters**   In the tabular case, we take the following hyper-parameters for both inner Q learning and outer Nash Q earning:

- a learning rate $\alpha_t = \frac{1}{n_t(\bar{s}_t, \bar{a}^1, \bar{a}^2)}$ where $n_t(\bar{s}_t, \bar{a}^1, \bar{a}^2)$ is the number of time that tuple $(\bar{s}_t, \bar{a}^1, \bar{a}^2)$ visited.
- $\epsilon_t = \epsilon_{end} + (\epsilon_{start} - \epsilon_{end}) \exp(-\frac{t}{T})$, where $T$ is the total training episode, $\epsilon_{end} = 0.01$, and $\epsilon_{start} = 0.99$.
- $\xi_t \sim \{0.99, 0.005, 0.005\}$

**Evaluation**   We evaluate the policy of each player by computing exploitability in Algo. 6. We perform tabular Q learning to solve an MDP to generate the best response.

**Baseline**   The baseline for DNashQ-MFTG is different from other examples. Each coalition learns the game independently through Q-learning after the same discretization as our DNashQ-MFTG, while for the exploitability computation, we still perform standard Q-learning with full observation of mean-field states to generate the best response.

We show more examples of distribution evolution in Fig. 4.

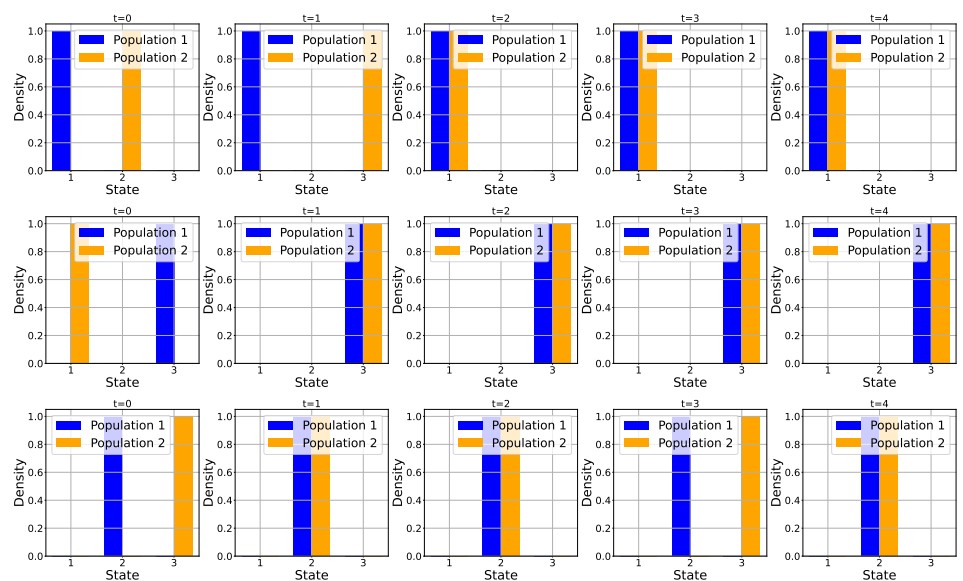

Figure 4: 1D Target Moving Grid Game: Population evolution of testing distribution at $t = 0, 1, 2, 3, 4$. From up to bottom are the evolution of testing distribution 1,2,3.

### F.2 EXAMPLE 2: FOUR-ROOM WITH CROWD AVERSION

**Model.** We consider a 2-dimensional grid world with four rooms and obstacles. Each room has only one door that connects to the next room and has $5 \times 5$ states.

- **Number of populations:** $m = 2$.
- **State space:** $S = \{0, \dots, N_x^1\} \times \{0, \dots, N_x^2\}$, where we set $N_x^1 = N_x^2 = 10$.
- **Action space:** $A = \{(-1, 0), (1, 0), (0, 0), (0, 1), (0, -1)\}$, which represents move left, move right, stay, move up, and move down, respectively.
- **Transitions:** At time $n$, the agent at position $s_n = (x, y)$ chooses an action $a_n$, the next state is computed according to

$$s_{n+1} = \begin{cases} s_n + a_n + \epsilon_{n+1}, & \text{if } s_n + a_n + \epsilon_{n+1} \text{ is not in a forbidden state} \\ s_n, & \text{otherwise} \end{cases} \quad (15)$$

where $\{\epsilon_n\}_n$ is a sequence of i.i.d. random variables taking values in $A$, representing the random disturbance.

The mean-field distribution $\bar{s}_t^i(x, y)$ is computed according to

$$\bar{s}_{t+1}^i(x, y) = \bar{s}_t^i(x, y)\bar{a}^i((0,0)|(x,y)) + \bar{s}_t^i(x, y-1)\bar{a}^i((0,1)|(x, y-1))$$
$$+ \bar{s}_t^i(x, y+1)\bar{a}^i((0,-1)|(x, y+1)) + \bar{s}_t^i(x+1, y)\bar{a}^i((-1,0)|(x+1, y))$$
$$+ \bar{s}_t^i(x-1, y)\bar{a}^i((1,0)|(x-1, y))$$

where $\bar{s}_t^i(a, b)$ is the density of Coalition i at the location $(a, b)$ at time step $t$.

- **One-step reward function:**

$$\bar{r}^1(\bar{s}_t^1, \bar{s}_t^2) = -\bar{s}_t^1 \cdot \log(\bar{s}_t^1 + \bar{s}_t^2)/\log(100)$$

$$\bar{r}^2(\bar{s}_t^1, \bar{s}_t^2) = -\bar{s}_t^2 \cdot \log(\bar{s}_t^1 + \bar{s}_t^2)/\log(100) - 30 \times \left( \bar{s}_t^2(2, 5) + \bar{s}_t^2(8, 5) + \bar{s}_t^2(5, 2) + \bar{s}_t^2(5, 8) \right)$$

where $\cdot$ is the inner product.

- **Time horizon:** $N_T = 40$.

**Training and testing sets**  For the training set, each player chooses locations among the four rooms with the sum of probability density equal to 1 as the initial distribution. We used three pairs of distributions as the testing set. Each of them is a uniform distribution among selected locations. The testing distributions are illustrated in Fig. 5.

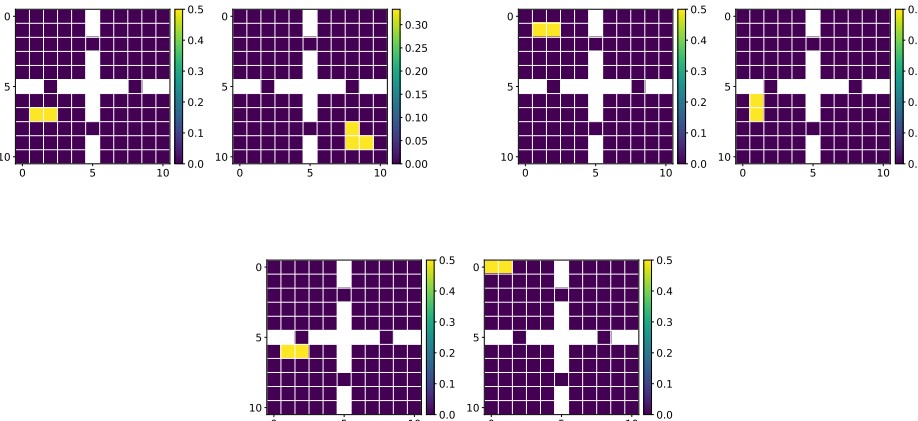

Figure 5: 3 pairs of testing distributions. For each pair, the left one is the initial distribution of Player 1, and the right one is the initial distribution of Player 2.

**Neural network architecture and hyper-parameters**  In the actor network, each state vector is initially flattened and fed into a fully connected network with a Tanh activation function, resulting in a 200-dimensional output for each. These outputs are then concatenated and processed through a two-layer fully connected network, each with 200 hidden neurons, utilizing ReLU and Tanh activation functions. The final output dimension is $|S| \times |A|$. The output is then normalized using the softmax function. The critic network follows a similar architecture. During the training, we use the Adam optimizer with the actor network learning rate equal to $5 \times 10^{-5}$ and the critic network learning rate equal to 0.0001. The standard deviation used in the Ornstein–Uhlenbeck process is 0.08. We also use target networks to stabilize the training and the update rate is 0.005. The replay buffer is of size 100000, and the batch size is 32. The model is trained using one GPU with 256GB memory, and it takes at most seven days to finish 50000 episodes.

### F.3   EXAMPLE 3: PREDATOR-PREY 2D WITH 4 GROUPS

**Model.**  In this $5 \times 5$ dimensional grid world, The transition dynamics and the action space are the same as in Example 2. In this game, we have one coalition acting as predator and another coalition as prey. Their reward function can be formulated as follows:

$$\bar{r}^1(\bar{s}_t, \bar{a}^1) = c_1 r_{\text{move}}(\bar{s}^1, \bar{a}^1) + c_2 \bar{s}^1 \cdot \bar{s}^2$$

$$\bar{r}^4(\bar{s}_t, \bar{a}^4) = c_1 r_{\text{move}}(\bar{s}_t^4, \bar{a}^4) - c_2 \bar{s}^3 \cdot \bar{s}^4$$

The remaining two coalitions act as predator and prey at the same time, with rewards:

$$\bar{r}^2(\bar{s}_t, \bar{a}^3) = c_1 r_{\text{move}}(\bar{s}^2, \bar{a}^2) + c_2(\bar{s}^2 \cdot \bar{s}^3 - \bar{s}^1 \cdot \bar{s}^2)$$

$$\bar{r}^3(\bar{s}_t, \bar{a}^3) = c_1 r_{\text{move}}(\bar{s}^3, \bar{a}^3) + c_2(\bar{s}^3 \cdot \bar{s}^4 - \bar{s}^2 \cdot \bar{s}^3),$$

where $c_1 = c_2 = 100$. Each episode has a time horizon $T = 21$ and $\gamma = 0.99$.

**Training and testing set**  For the training set, we sample each element in the grid world from a uniform distribution over the interval $[0, 1)$ and then divide each element by the total sum of the matrix to normalize it. Testing set can be found in Fig. 7.

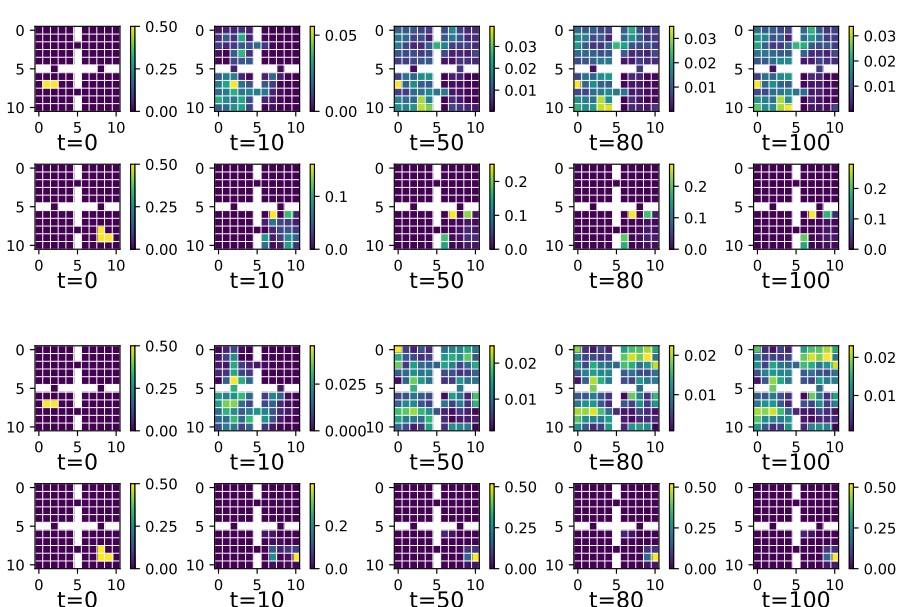

Figure 6: **Ex. 2:** populations evolution 2. The top two rows show the distribution evolution of the two players. The bottom two rows show the corresponding distribution evolution of the baseline model.

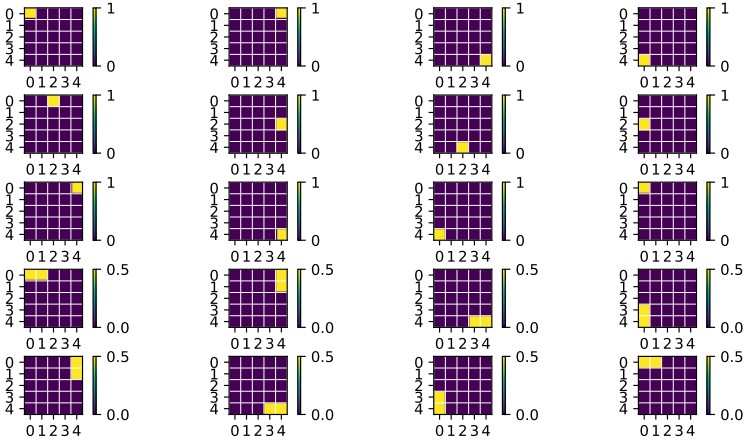

Figure 7: 5 sets of testing distributions for predator-prey 2D with 4 groups. Each row shows one set of testing distribution for 4 coalitions. For each row, from the left to the right are the coalition 1 to coalition 4.

**Neural network architecture and hyper-parameters** The architectures of the actor and critic networks are the same as those used in the discrete planning 2D (Appx. F.4). We use the Adam optimizer, with learning rates set to 0.0005 for the actor network and 0.001 for the critic network. The Ornstein-Uhlenbeck noise standard deviation is set to 0.8. Target networks are updated at a rate of 0.0025. The replay buffer has a capacity of 50,000 and a batch size of 64. This experiment was run on a GPU with 64GB of memory, taking two days to complete 80,000 episodes of training.

**Numerical results.** We conducted this experiment over 5 runs, with each run corresponding to a specific testing distribution from the testing set. For each run, we averaged the exploitability of all players to determine the run's exploitability. We then calculated the mean and standard deviation of exploitability across the 5 runs. Additionally, for the testing reward, we calculated the mean and standard deviation for each player over the 5 runs. Fig. 8 shows the testing rewards.

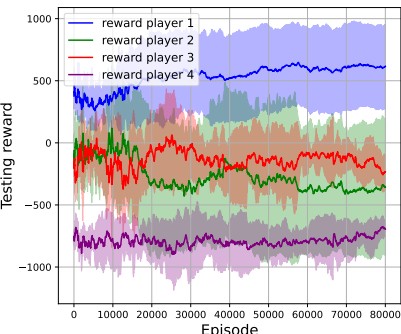

Figure 8: **Ex. 3:** testing rewards.

### F.4 EXAMPLE 4: DISTRIBUTION PLANNING IN 2D

There are $m = 2$ populations. The agent's state space is a $5 \times 5$ state 2D grid world, with the center as a forbidden state. The possible actions are: move up/down/left/right or stay, and there is no individual noise perturbing the movements. The rewards encourage each population to match a target distribution (hence the name "planning"): Population 1 and 2 move respectively towards the top left and bottom right corners, with a uniform distribution over fixed locations (see Fig. 11). We describe the model details and the training and testing distributions below. We implement **DDPG-MFTG** to solve this game. The numerical results are presented in Figs. 9 and 10. We make the following observations. **Testing reward curves:** Fig. 9 (left) shows the testing rewards. In this game setting, the Nash equilibrium for each coalition is to move to its target position without interacting with the other coalition. We observe that the testing rewards increase and then stabilize with minimal oscillation. The reward curve of the baseline stays below the one using DDPG-MTFG. **Exploitability curves:** Fig. 11 (right) shows the averaged exploitabilities over the testing set and players. We observe that the game reaches Nash equilibrium around 15000 episodes. The baseline shows higher exploitability than the **DDPG-MFTG** algorithm. **Distribution plots:** Fig. 10 illustrates the distribution evolution during the game. With the policy learned using **DDPG-MFTG**, each player deterministically moves to the target position in several steps and avoids overlapping with the other player during movement.

**Model.**

- **Number of populations:** $m = 2$.
- **State space:** $S = \{0, \ldots, N_x^1\} \times \{0, \ldots, N_x^2\}$, where we set $N_x^1 = N_x^2 = 4$.
- **Action space:** $A = \{(-1,0), (1,0), (0,0), (0,1), (0,-1)\}$, which represents move left, move, right, stay, move up, and move down, respectively.
- **Transitions:** At time $n$, the agent at position $s_n = (x, y)$ chooses an action $a_n$, the next state is computed according to

$$s_{n+1} = \begin{cases} s_n + a_n, & \text{if } s_n + a_n \text{ is not in a forbidden state} \\ s_n, & \text{otherwise} \end{cases} \tag{16}$$

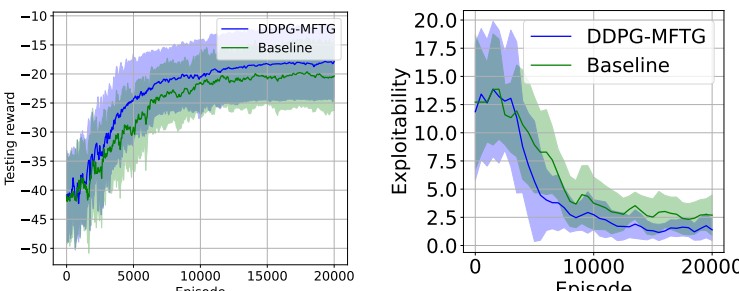

Figure 9: Left: Testing rewards. Right: exploitabilities.



Figure 10: Distribution planning in 2D: The top row and the bottom respectively show the distribution evolution of player 1 and 2 using the policy learned by DDPG-MFTG.

The mean-field distribution $\bar{s}_t^i(x, y)$ is computed according to

$$
\begin{aligned}
\bar{s}_{t+1}^i(x, y) = {} & \bar{s}_t^i(x, y)\bar{a}^i((0,0)|(x,y)) + \bar{s}_t^i(x, y-1)\bar{a}^i((0,1)|(x,y-1)) \\
& + \bar{s}_t^i(x, y+1)\bar{a}^i((0,-1)|(x,y+1)) + \bar{s}_t^i(x+1, y)\bar{a}^i((-1,0)|(x+1,y)) \\
& + \bar{s}_t^i(x-1, y)\bar{a}^i((1,0)|(x-1,y))
\end{aligned}
$$

where $\bar{s}_t^i(a, b)$ is the density of Population i at the location $(a, b)$ at time step $t$.

- **One-step reward function:** Each central player $i$ aims to make the population match a target distribution $m_i$ while maximizes the reward. For each player $i$, the reward of each step is

$$
\bar{r}^i(\bar{s}_t^1, \bar{s}_t^2, \bar{a}^i) = c_1 r_{\text{move}}(\bar{s}^i, \bar{a}^i) + c_2 r(\bar{s}^i, m_i) + c_3 r(\bar{s}^1, \bar{s}^2),
$$

where $r_{\text{move}}(\bar{s}^i, \bar{a}^i) = -\bar{s}^i \cdot ||\bar{a}^i||$ is the cost for moving, $r(\bar{s}^i, m_i) = -\text{dist}(\bar{s}^i, m_i)$ is the distance to a target distribution, $r(\bar{s}^1, \bar{s}^2) = -\bar{s}^1 \cdot \bar{s}^2$ is the inner product of the two population distributions. $c_i$ is the coefficient, for $i = 1, 2, 3$. Here, $c_1 = 1$, $c_2 = 2$, and $c_3 = 5$.

- **Time horizon:** $N_T = 10$.

**Training and testing sets.** The training set consists of a randomly sampled location with a probability density 1 representing the initial state. See Fig. 12 for testing distribution.

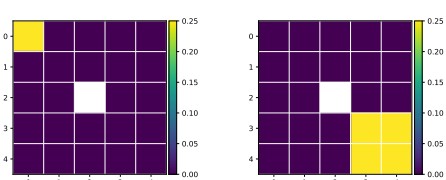

Figure 11: Target distributions for player 1 (left) and player 2 (right).

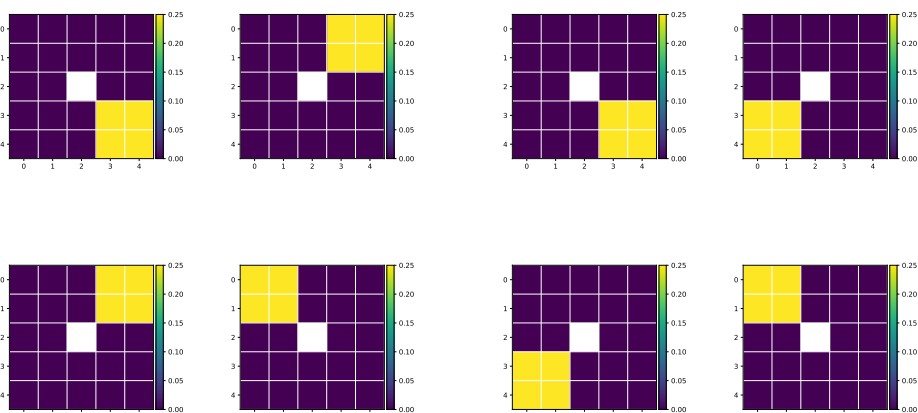

Figure 12: 4 pairs of testing distributions. For each pair, the left one is the initial distribution of player 1, and the right one is the initial distribution of player 2.

**Neural network architecture and hyper-parameters**  In the actor network, each state vector is initially flattened and fed into a fully connected network with a ReLU activation function, resulting in a 200-dimensional output for each. These outputs are then concatenated and processed through a two-layer fully connected network with 200 hidden neurons, utilizing ReLU and Tanh activation functions. The final output dimension is $|S| \times |A|$. The output is then normalized using the softmax function. The critic network follows a similar architecture, where we use ReLU in the last layer. During the training, we use the Adam optimizer with the actor-network learning rate equal to $5 \times 10^{-5}$ and the critic-network learning rate equal to 0.0001. Both learning rates are reduced by half after around 6000 and 12000 episodes. The standard deviation used in the Ornstein–Uhlenbeck process is 0.08 and is also reduced by half after around 6000 and 12000 episodes. We also use target networks to stabilize the training and the update rate is 0.005. The replay buffer is of size 50000, and the batch size is 128. The model is trained using one GPU with 256GB memory, and it takes at most two days to finish 20000 episodes.

### F.5 EXAMPLE 5: CYBER SECURITY

We now present another example in a cyber security setting inspired by Kolokoltsov & Bensoussan (2016) in the context of MFGs and Carmona et al. (2023) in the context of discrete-time MFC. The original formulation has only one type of player, namely computers defending themselves against a virus. Here, we add another group, corresponding to attackers, and we study the MFTG. The defenders cooperate with each other, and likewise for the attackers. Each of the two populations competes with the other to maximize its reward.

**Model.**  The model is as follows:

- **Number of populations:** $m = 2$, defenders and attackers.

- **State spaces:** For defenders, $S^{def} = \{DI, DS, UI, US\}$, standing respectively for defended and infected, defended and susceptible of infection, undefended and infected, undefended and susceptible of infection. For attackers, $S^{att} = \{active, inactive\}$. When an attacker is active, it is able to infect the defenders. As we will see below, the defenders' transitions from susceptible to infected are affected by the proportion of active attackers.

- **Action space:** The central player of each population can influence the state in the following way. The defenders' central player can influence the transition probability between defended and undefended. The attackers' central player can influence the transition probability between active/inactive. The action space is $A = \{0, 1\}$ for both populations. The central player of a population chooses 0 if they are satisfied with the current level, and 1 if they want to switch to the other level.

- **Dynamics:** We describe the model in continuous time and then its discrete-time version. When the central player chooses to change the current state for the agents in a specific state, the update occurs at an update probability $\lambda$ for both two populations. If the agents use pure policies, then at each of the states, the central player for defenders (resp. attackers) only chooses one action per state and applies it to all the agents among the defenders (resp. attackers) at that state. If the agents use mixed control, then for each state, the central player for defenders (resp. attackers) chooses a distribution over actions and each agent among the defenders (resp. attackers) in this state picks independently an action according to the chosen distribution. When infected, each defender agent may recover at a rate $q_{rec}^D$ or $q_{rec}^U$ depending on whether it is defended or not. Also, a defender agent may be infected either directly by an attacker agent at rate $v_H q_{inf}^D$ or $v_H q_{inf}^U$ depending on whether it is defended or not. A defender can also get infected by undefended infected (UI) defenders, at rate $\beta_{UU}\mu(UI)$ or $\beta_{UD}\mu(UI)$ depending on whether it is undefended or not; it can also get infected by defended infected (DI) defenders, at rate $\beta_{DU}\mu(DI)$ or $\beta_{DD}\mu(DI)$, depending on whether it is undefended or not. Here $v_H$ stands for the proportion of attackers who are active. In short, the transition rate matrix is given by:

$$P_{\mu,a}^{def} = \begin{pmatrix} \cdots & P_{DS \to DI}^{\mu,a} & \lambda a & 0 \\ q_{rec}^D & \cdots & 0 & \lambda a \\ \lambda a & 0 & \cdots & q_{rec}^U \\ 0 & \lambda a & P_{US \to UI}^{\mu,a} & \cdots \end{pmatrix}$$

where

$$P_{DS \to DI}^{\mu,a} = v_H q_{inf}^D + \beta_{DD}\mu_t(DI) + \beta_{UD}\mu_t(UI)$$
$$P_{US \to UI}^{\mu,a} = v_H q_{inf}^U + \beta_{UU}\mu_t(UI) + \beta_{DU}\mu_t(DI)$$

The attackers' transition matrix is:

$$P_{\mu,a}^{att} = \begin{pmatrix} \cdots & \lambda a \\ \lambda a & \cdots \end{pmatrix}.$$

In these matrices, $a$ represents the action for the corresponding distribution. The summation of every row of these transition rate matrices is 0. From this continuous-time model, we derive a discrete-time version, which will fit in the MDP framework. We consider a time step size $\Delta t$. We formulate the transition probability matrices as follows:

$$\mu_{t+1}^{def} = \mu_t^{def\top}(I + P_{\mu,a}^{def}\Delta t)$$
$$\mu_{t+1}^{att} = \mu_t^{att\top}(I + P_{\mu,a}^{att}\Delta t),$$

where $I$ denotes the identity matrix, and $I + P_{\mu,a}^{def}\Delta t$ and $I + P_{\mu,a}^{att}\Delta t$ represent the transition probability matrices for defenders and attackers respectively. For transition probability matrices, the summation of each row equals 1.

- **Reward functions:** We use:

$$R_t^{def} = -20\mu_t(DI) - 10\mu_t(DS) - 10\mu_t(UI)$$
$$R_t^{att} = 10\mu_t(DI) + 10\mu_t(UI) - 10\mu_t(active)$$

For this game, we consider a terminal horizon $T$ and we accumulate the rewards every time step until $T$. The steps are of length $\Delta t$.

- **Parameter values:** We use $\beta_{UU} = 0.3$, $\beta_{UD} = 0.4$, $\beta_{DU} = 0.3$, $\beta_{DD} = 0.4$, $\lambda = 0.8$, $q_{rec}^D = 0.5$, $q_{rec}^U = 0.4$, $q_{inf}^D = 0.4$, $q_{inf}^U = 0.3$. In the experiments, we take $T = 10$ and $\Delta t = 0.1$.

**Training and Testing Data Sets.** We used different initialization methods for training in the inner loop and outer loop. For the outer loop, we used a uniform random sampler which samples a random number in the interval $[0, 1)$ according to uniform distribution in each entry in the initial state. Then we normalized the initial state to make the sum of the distribution of each population equals to 1. For exploitability computation, we run the inner loop for a fixed initial distribution so we set the training initial distribution the same as the testing initial distribution. For the experiments using our algorithm, **DDPG-MFTG**, and the baseline, there are eight different fixed initial distributions, and we run each of them three times with different random seeds. So the exploitability curve for each population is plotted based on the average of numerical results from 24 experiments. The eight initial testing distributions are:

$$
\begin{aligned}
\mathcal{D}_{\text{test}} = \big\{ &\big((0.8, 0.05, 0.05, 0.1), (0.3, 0.7)\big), \big((0.5, 0.2, 0.2, 0.1), (0.3, 0.7)\big), \\
&\big((0.1, 0.1, 0.7, 0.1), (0.7, 0.3)\big), \big((0.5, 0.0, 0.5, 0.0), (0.55, 0.45)\big), \\
&\big((0.25, 0.15, 0.35, 0.25), (0.55, 0.45)\big), \big((0.35, 0.35, 0.15, 0.15), (0.3, 0.7)\big), \\
&\big((0.45, 0.35, 0.15, 0.05), (0.2, 0.8)\big), \big((0.15, 0.25, 0.25, 0.35), (0.7, 0.3)\big) \big\}
\end{aligned}
$$

**Neural Network Architecture and Hyper-Parameters.** We implement the same neural network structure for both populations. For the actor network, the input is the concatenated vector of states for the two populations, and there are two hidden layers with 100 neurons in each layer. All activation functions in both hidden layers and the output layer are set to be sigmoid. The output dimension of the actor network is the same as the dimension for the action vector for the defenders/attackers, depending on which population the network is used for. For the critic network, the input is the concatenated vector of the states for the two populations, and together with the action vector for the defenders/attackers, depending on which population the critic network is used for. The architecture of hidden layers and corresponding activation functions is same as the actor network. For the output layer of the critic network, the activation function is an identity function, and the output is a single value. For both the inner loop and outer loop, the learning rates are set as follows: for the defenders, the learning rates for actor and critic are respectively 0.0006 and 0.0009; for the attackers, the learning rates for actor and critic are respectively $6 \times 10^{-5}$ and $9 \times 10^{-5}$. The replay buffer is of size 5000 and the batch size is 64. The model is trained using one CPU with 256GB memory, and it takes around 24 hours to finish 100 episodes. The exploitability is calculated every 4 episodes within the first 20 episodes, and then every 10 episodes from 20 to 100 episodes. The length of an inner loop to learn the best response is 400 episodes.

**Numerical results.** We implement **DDPG-MFTG** to solve this game. The numerical results are presented in Figs. 13 and 14. We make the following observations based on the provided plots. **Exploitability curves**: Fig. 13 shows that for the defenders' exploitability curve, both the baseline and the **DDPG-MFTG** algorithm converge to 0, which indicates the algorithms are learning a Nash equilibrium. For the attackers' exploitability curve, we can see a clear gap between the exploitability curve of our **DDPG-MFTG** algorithm and the baseline, which shows the improvement made by our algorithm. Also, both two curves tend to reach towards 0, which shows that the algorithms are converging to a Nash equilibrium. **Population Distribution curves**: Fig. 14 provides two examples of population distributions used for testing.

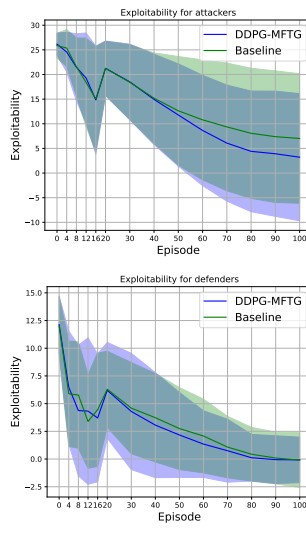

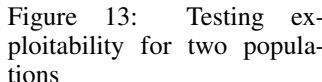

Figure 13: Testing exploitability for two populations

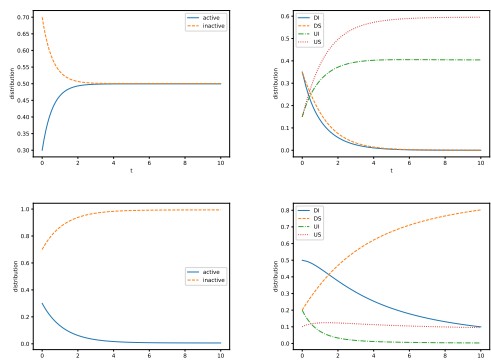

Figure 14: Two examples of population distribution evolution. The first row is for the initial distribution $\big((0.35, 0.35, 0.15, 0.15), (0.3, 0.7)\big)$, the second row is for $\big((0.5, 0.2, 0.2, 0.1), (0.3, 0.7)\big)$. The left column is for the distribution of attackers, and the right column for the distribution of defenders.

## F.6 SUMMARY OF IMPROVEMENTS

In Table 1, we summarize the improvement brought by our method compared with the corresponding baseline, in each example. The quantities are:

- **Baseline Exploitability:** The baseline's mean value (as described in the paper).
- **Our Exploitability:** Our method's mean value (as described in the paper).
- **Improvement:** The percentage improvement is calculated as:

$$\text{Improvement (percentage)} = \frac{\text{Baseline} - \text{Ours}}{\text{Baseline}} \times 100.$$

|  | Example 1 | Example 2 | Example 3 | Example 4 | Example 5 |
|---|---|---|---|---|---|
| **Baseline Exploitability** | 2355.35 | 3.13 | 131.43 | 2.69 | 6.93 |
| **Our Exploitability** | 471.40 | 2.16 | 38.75 | 1.39 | 3.14 |
| **Improvement** | 79.98% | 31.0% | 70.52% | 48.3% | 54.69% |

Table 1: Comparison of baseline and our exploitability metrics across the 5 examples described in the text, along with percentage improvement.

# G HYPERPARAMETERS SWEEP

We explore various batch sizes, actor learning rates, and standard deviations of Ornstein-Uhlenbeck noise (OU noise) across all numerical experiments. Heuristically, we set $\alpha_{\text{critic}} = 2 \times \alpha_{\text{actor}}$ and $\tau = 5 \times \alpha_{\text{actor}}$. Each hyperparameter group is evaluated during one player's exploitability computation stage, and the results are presented as follows:

## G.1 PREDATOR-PREY 2D WITH 4 GROUPS

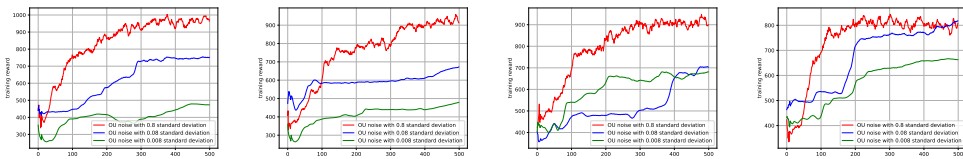

Figure 15: Exploitability computation training reward with $\alpha_{\text{actor}} = 5 \times 10^{-5}$. Batch size from left to right: 16, 32, 64, 128

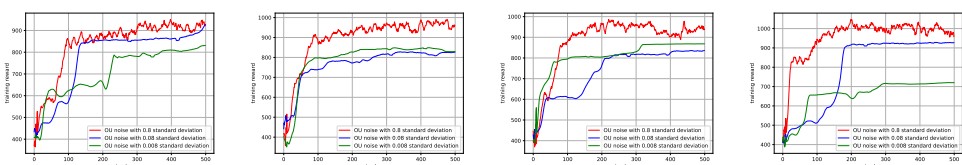

Figure 16: Exploitability computation training reward with $\alpha_{\text{actor}} = 0.0005$. Batch size from left to right: 16, 32, 64, 128

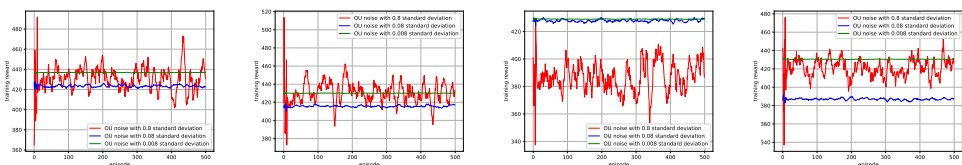

Figure 17: Exploitability computation training reward with $\alpha_{\text{actor}} = 0.005$. Batch size from left to right: 16, 32, 64, 128

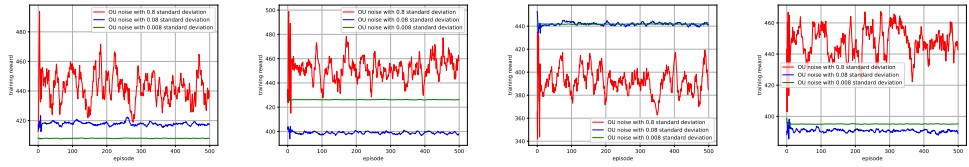

Figure 18: Exploitability computation training reward with $\alpha_{\text{actor}} = 0.05$. Batch size from left to right: 16, 32, 64, 128

## G.2 DISTRIBUTION PLANNING IN 2D

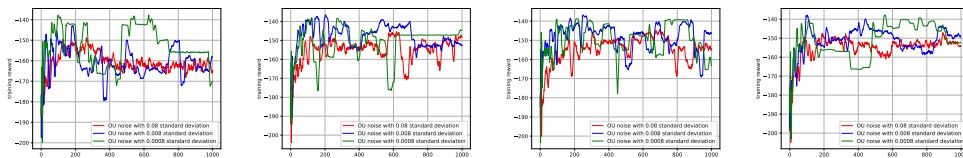

Figure 19: Exploitability computation training reward with $\alpha_{\text{actor}} = 0.0005$. Batch size from left to right: 16, 32, 64, 128

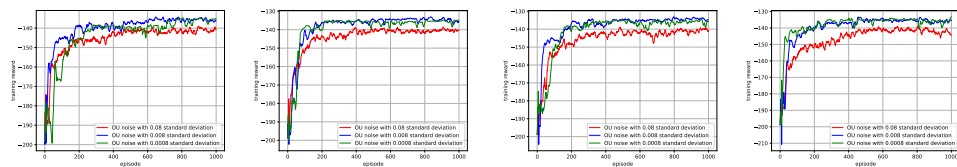

Figure 20: Exploitability computation training reward with $\alpha_{\text{actor}} = 5 \times 10^{-5}$. Batch size from left to right: 16, 32, 64, 128

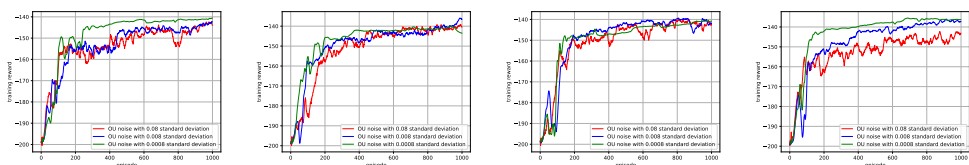

Figure 21: Exploitability computation training reward with $\alpha_{\text{actor}} = 5 \times 10^{-6}$. Batch size from left to right: 16, 32, 64, 128

### G.3 Four-room with crowd aversion

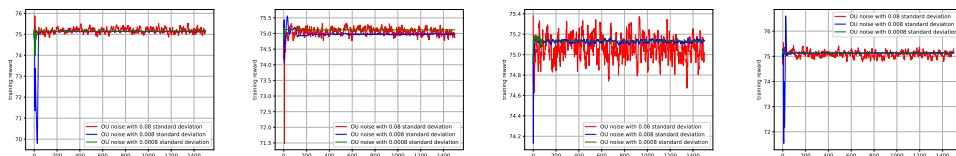

Figure 22: Exploitability computation training reward with $\alpha_{\text{actor}} = 0.005$. Batch size from left to right: 16, 32, 64, 128

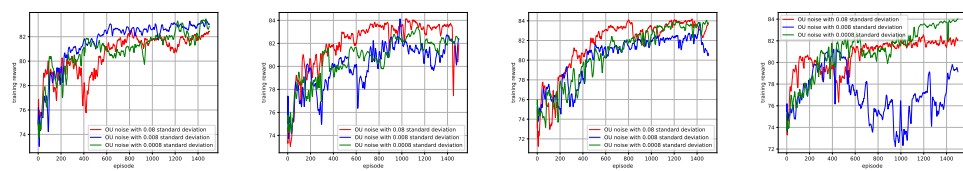

Figure 23: Exploitability computation training reward with $\alpha_{\text{actor}} = 0.0005$. Batch size from left to right: 16, 32, 64, 128

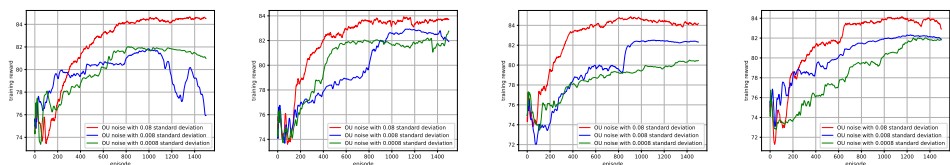

Figure 24: Exploitability computation training reward with $\alpha_{\text{actor}} = 5 \times 10^{-5}$. Batch size from left to right: 16, 32, 64, 128

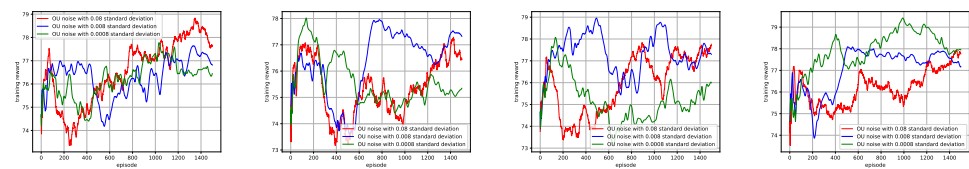

Figure 25: Exploitability computation training reward with $\alpha_{\text{actor}} = 5 \times 10^{-6}$. Batch size from left to right: 16, 32, 64, 128

### G.4 CYBER SECURITY

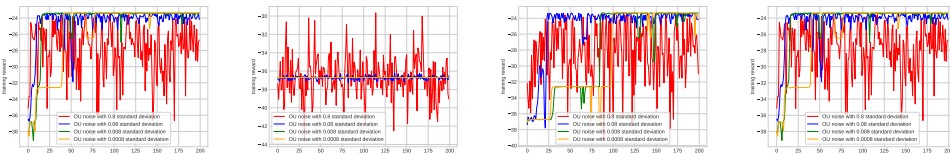

Figure 26: Exploitability computation training reward for defenders with $\alpha_{\text{actor}} = 0.06$. Batch size from left to right: 16, 32, 64, 128

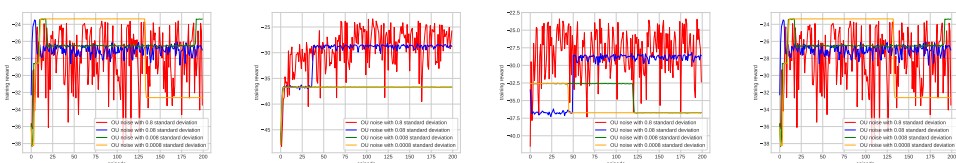

Figure 27: Exploitability computation training reward for defenders with $\alpha_{\text{actor}} = 0.006$. Batch size from left to right: 16, 32, 64, 128

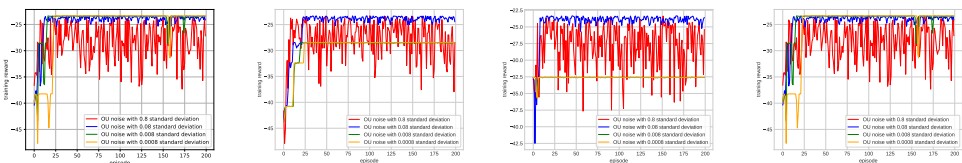

Figure 28: Exploitability computation training reward for defenders with $\alpha_{\text{actor}} = 0.0006$. Batch size from left to right: 16, 32, 64, 128

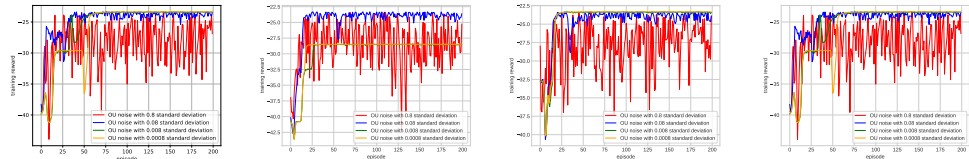

Figure 29: Exploitability computation training reward for defenders with $\alpha_{\text{actor}} = 6 \times 10^{-5}$. Batch size from left to right: 16, 32, 64, 128

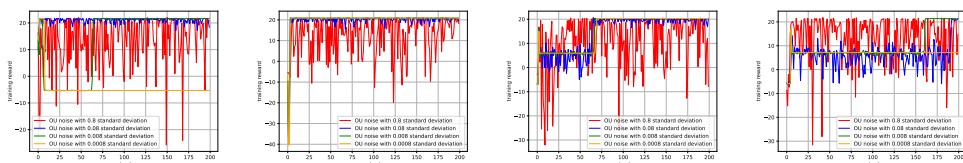

Figure 30: Exploitability computation training reward for attackers with $\alpha_{\text{actor}} = 0.006$. Batch size from left to right: 16, 32, 64, 128

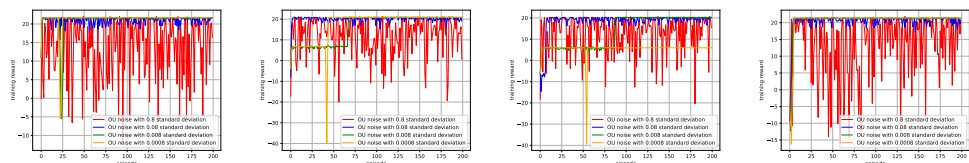

Figure 31: Exploitability computation training reward for attackers with $\alpha_{\text{actor}} = 0.0006$. Batch size from left to right: 16, 32, 64, 128

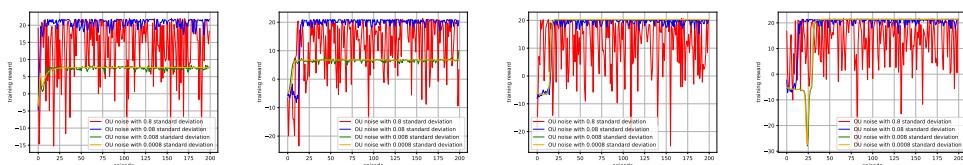

Figure 32: Exploitability computation training reward for attackers with $\alpha_{\text{actor}} = 6 \times 10^{-5}$. Batch size from left to right: 16, 32, 64, 128

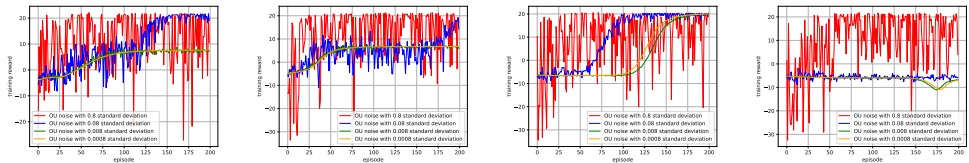

Figure 33: Exploitability computation training reward for attackers with $\alpha_{\text{actor}} = 6 \times 10^{-6}$. Batch size from left to right: 16, 32, 64, 128

