# OpenReview forum: "Reinforcement Learning for Finite Space Mean-Field Type Games"
_ICLR.cc/2025/Conference — Submitted to ICLR 2025_

### Official Review · Reviewer_RAvE · 2024-10-31

**Soundness:** 3
**Presentation:** 3
**Contribution:** 3
**Rating:** 8
**Confidence:** 3

**Summary:**

The paper considers mean field type games which combine cooperative and competitive characteristics, i.e. agents form multiple cooperative coalitions but there is competition between coalitions. The authors focus on a quite general finite space case where they derive novel theoretical results and propose different learning schemes. They conclude their work with an evaluation of the proposed learning algorithms on different examples.

**Strengths:**

I think that this paper is a nice contribution to the MF(T)G literature and closes a conceptual gap in the existing literature. This work is well-written and structured. The assumptions for the theoretical results are clearly stated and discussed, and the Nash Q-learning algorithm is also analyzed rigorously. The algorithms appear to perform well on the given examples, and I liked that the authors provide both a mathematically tractable learning approach (Nash Q) as well as a scalable algorithm based on deep learning. Finally, the authors did a good job at discussing limitations of their current work.

**Weaknesses:**

Overall, I think the paper is sound and I did not spot major weaknesses. One general criticism that I have is that the presentation and formatting of the figures can be improved, e.g. by increasing font sizes and making captions more informative (see the following comments for details). Here are my comments:

-	Lines 34-35: Mean field games were not (first) introduced in these references. I think the authors mixed these references up with the earlier works of Lasry and Lions (2007) and Huang, Malhame and Caines (2006), which would be the correct references here.
-	Line 111, a small question regarding notation: Why do we need all $N_1, …, N_m$ in the superscript for the MF of coalition $i$? Wouldn’t it suffice and simplify notations if the superscript would just be $i, N_i$?
-	Suggestion: One could define something like $\bar{N} \coloneqq (N_1, \ldots, N_m)$ or similar to abbreviate the following mathematical expressions.
-	Assumption 3, potential typo: “The policies … satisfies” should be “satisfy”.
-	The font size in most of the figures, especially Figures 1 and 3, is very small. Is it possible to increase the figure font size?
-	Line 478: It says “Fig. 5 (left) shows the testing rewards”. Since Figure 5 is just one plot, what does “left” refer to here? Also, Figure 5 shows exploitabilities but not testing rewards.
-	Line 481: There is a reference to Figure 1 here. Is this correct?
-	Some of the figure captions, e.g. Figures 3 and 5, are very short and uninformative. For example, the caption in Figure 3 could at least mention that the exploitability is averaged over testing sets and players. And it should also briefly state what the bands surrounding the curves depict. In general, it should be possible to get a basic understanding of the figures without reading all details in the main text.

**Questions:**

1. Theorem 2.4: Could the authors please discuss the condition on the discount factor $\gamma (1 +L_\pi + L_p) < 1$? Especially, how restrictive is this inequality?
2. Example 1: I am not sure whether Example 1 is an expressive game. To my understanding, the optimal solution is rather trivial (Coalition 1 stays where it is, Coalition 2 moves to the state of Coalition 1) and I don’t see any potential conflict of interest between the two coalitions. Thus, my question is if the authors consider this game to be competitive in a strict sense on the coalition level? If so, what exactly are the coalitions competing about?
3. How scalable are the proposed learning algorithms with respect to a growing number $m$ of coalitions?

---

> ### Author Response · Authors · 2024-11-18
> **Reply to Reviewer RAvE**
>
> We are grateful for your comments and interesting questions, and we thank you for the positive assessment of our work. We provide detailed answers to your questions below. We will also include all the changes in the revised PDF.
>
> **Weakness:**
> Thank you for pointing out the typos and suggesting changes. We will correct all these points in the revised paper.
> - Reference: Thank you and sorry for mixing up the references.
> - Superscripts: We used the superscript $N_1,N_2,...,N_m$ to emphasize that the number of agents in  each coalition might be different. But we agree with your suggestion to make the notation more concise. We will use it in the revised PDF.
> - Assumption 3: thank you, we corrected the typo.
> - Font size in figures: thank you, we will increase the font size in the revised version.
> - Figure reference: Sorry for the typo. The figure referenced in Line 478 should be Figure 11, which contains the testing rewards. We moved it to the Appendix due to space limit and because there are 4 populations in this example.
> - The figure mentioned in line 481 is Figure 5. We apologize for the confusion.
> - Captions: Thank you for the suggestions, we will improve the captions in the revised version.
>
> **Question 1 (Theorem 2.4 and discount):**
> Thank you for your question. Besides the information provided in point (3) of the common reply, we would like to clarify that:
> * The condition $\gamma(1+L_\pi+L_p)<1$ involves $\gamma$ because we consider the infinite horizon setting and it helps to control the propagation of errors in future time steps and give a **rate of convergence**.
> * If $\pi$ and $p$ do not depend on the mean field, i.e., $L_\pi=L_p=0$, this condition reduces to $\gamma<1$ which is standard in MDPs.
> * Other works often use similar conditions. For example Saldi et al. (2018) (for MFGs and not MFTGs) also have a restrictive condition on the discount factor ($\beta$ in their notation), namely assumption (g) on page 4261 requires $\alpha\beta\gamma < 1,$ with $\alpha$ and $\gamma$ defined on the same page. Their $\gamma$ is greater than $1$ and $\alpha$ would probably be larger than $1$ too in some cases (e.g., if the transition kernel $p$ gives more mass to regions where $w$ is large). In that case, $\alpha\gamma$ can be much larger than $1$, hence requiring the discount $\beta$ to be much smaller than $1$. Furthermore, Saldi et al. also have conditions on the moduli of continuity $w_p(r)$ and $w_c(r)$ of the transitions and the cost (see assumption (h) on page 4262 in their paper). While it is true that this is slightly less restrictive than Lischitz continuity, **their Theorem 4.1 only provides an asymptotic convergence and no convergence rate**.
>
> **Question 2 (Example 1 and competition):**
>     Thank you for this interesting question. We agree and since we got the reviews, we conducted new experiments with a more complex model for Example 1, in which we introduced a cross-population density term in coalition 1's reward function while keeping coalition 2's setting unchanged. With this modification, coalition 1 will learn to avoid coalition 2 while trying to maintain its position, whereas Coalition 2's objective is to catch coalition 1. In the notation of our paper, the reward function for coalition 1 is modified to include a term $-c_{2}(\bar{s}^{1} \times \bar{s}^{2})$ while for coalition 2, we keep the same reward. We observe numerically that the average exploitability of the baseline is above 1000 while the exploitability of our method is below 500. We will include the new setting and the new figures in the revised PDF.
>
> **Question 3 (Scalability in number of coalitions):**
> Increasing the number of coalitions is an interesting direction. In this work, we are mostly focused on settings with a relatively small number of coalitions. In Example 4, we have 4 coalitions and we solved it without using very large computing resources (a single RTX 8000 GPU) and we believe it should be possible to solve similar problems with more coalitions (e.g., of the order of 10 coalitions). When the number of coalitions is very large (e.g., hundreds or thousands), it is probably unrealistic to solve exactly the MFTG and we would recommend using other approximations. For instance, one could have a mean field of mean field coalitions, and the algorithm would learn the behavior of one typical agent in one typical coalition. Such models seem similar to graphon-based models (Caines & Huang, 2019) and mean field control games (Angiuli et al., 2022). However such approaches are relevant only for very large number of coalitions. In the regime with small or moderate number of coalitions, we still believe that our approach is more precise and practical. We will add comments about this in the revision.

---

> > ### Comment · Reviewer_RAvE · 2024-11-26
> >
> > I thank the authors for their detailed response and answering all of my questions. I would thus like to keep my original score.

---

### Official Review · Reviewer_iiZW · 2024-11-02

**Soundness:** 3
**Presentation:** 3
**Contribution:** 2
**Rating:** 8
**Confidence:** 4

**Summary:**

The paper considers an extension of mean field games to many agents in coalitions that compete against each other. In the case of many agents, the infinite-agent case reduces the otherwise difficult problem to a classical finite player game, albeit with high-dimensional state-actions. As a result, the problem is analyzed and solved using classical techniques (Nash Q-Learning) and discretization / deep RL. The proposed algorithms are verified on a variety of examples, and compared to basic independent RL.

**Strengths:**

- The paper is well written and clear.
- The work fuses existing ideas from MFGs with algorithms from both game theory and deep RL.
- The transfer of NashQ-based theoretical analysis and algorithms beyond classical mean-field RL (Yang et al., 2018) to mean-field games and mean-field-type games is interesting.
- The empirical evaluations and examples extensively demonstrate the potential of proposed algorithms.

**Weaknesses:**

- The empirical significance of the paper is somewhat limited, as the experiments show only limited improvement in terms of exploitability over baselines, while the setting sounds a bit niche to me. I wonder if one could discuss more the significance of the setting, as I believe the referenced works (Tembine et al.) do not consider coalitions, but rather refer to standard MFGs as mean-field-type games.
- The theoretical results are limited due to many assumptions (Assumptions 1-8): The approximate Nash property in Theorem 2.4 requires an unusually strong condition on $\gamma$, and the algorithmic convergence (understandably) requires strong conditions on the stage games. To improve the applicability of the theoretical results, perhaps some assumptions in Section 2.2 could be relaxed or discussed more.
- The model and NashQ-based analysis are not highly novel, as partially remarked by the authors. The setting is also related to graph-based works, e.g., [1] considering infinite numbers of infinitely large coalitions through a double limit.

[1] Caines, Peter E., and Minyi Huang. "Graphon mean field games and the GMFG equations: $\epsilon$-Nash equilibria." 2019 IEEE 58th Conference on Decision and Control (CDC). IEEE, 2019.

**Questions:**

- Is Assumption 3 required, or could one perhaps also consider smaller classes of policies, e.g., ones that do not depend on the deterministic-in-the-limit mean field?
- What is the difficulty in relaxing the strong condition on $\gamma$ in Theorem 2.4, as it is not required in classical MFGs [2]?
- Could Assumption 8 follow from the previous assumptions?

[2] Saldi, Naci, Tamer Basar, and Maxim Raginsky. "Markov--Nash equilibria in mean-field games with discounted cost." SIAM Journal on Control and Optimization 56.6 (2018): 4256-4287.

---

> ### Author Response · Authors · 2024-11-18
> **Reply to Reviewer iiZW (1/2: Weaknesses)**
>
> We thank you for all the comments. We hope that our answers below will answer your questions adequately and convince your of the merits of our paper. We would be grateful if you would consider raising your score.
>
>
> **Weakness 1 (empirical evidence):** Thank you for your question. Although the curves sometimes look close, the values actually show a clear improvement: our method brings an **improvement of at least 30%** in each example. We summarize the relative improvement in the table below, where:
> - **Baseline Exploitability**: The baseline's mean value (as described in the paper).
> - **Our Exploitability**: Our method's mean value (as described in the paper).
> - **Improvement**: The percentage improvement is calculated as:
>     $$
>   \text{Improvement (percentage)} = \frac{\text{Baseline} - \text{Ours}}{\text{Baseline}} \times 100
>     $$
>
> |                 | Example 1  | Example 2  | Example 3  | Example 4  | Example 5  |
> |-----------------|------------|------------|------------|------------|------------|
> | **Baseline Exploitability** | 2355.35         | 3.13         |  131.43      | 2.69         | 6.93         |
> | **Our Exploitability**     | 471.40         | 2.16         |  38.75        | 1.39         | 3.14         |
> | **Improvement**           | 79.98%        | 31.0%        | 70.52%        | 48.3%        | 54.69%
>
> We will include this table in the revision.
>
>
>
> **Weakness 2 (setting and "coalitions"):**
> *"I believe the referenced works [...] refer to standard MFGs as mean-field-type games."* We respectfully disagree:
> - What Tembine et al. call a mean-field-type game is not standard MFGs. For example in (Tembine, 2017), the author wrote on page 706-707 "*In contrast to mean-field games [2, 3] in which a single player does not influence of the mean-field terms, here, it is totally diﬀerent. In mean-field-type games, a single player may have a strong impact on the mean-field terms.*" Actually this is one of the **major challenges** that we tackle in our work and which justifies using **population-dependent policies**.
> - Furthermore, on page 707, "*Mean-field-type games with two or more players can be seen as the multi-agent generalization of the single agent mean-field-type control problem.*" And mean-field-type control problems can be interpreted as a cooperative game with players who try to jointly minimize a cost (or maximize a reward). This is what we call a "coalition".
> - The way we use "coalition" might be different from other authors and we are open to using a different word if you have suggestions. But we hope that you now agree with the interpretation and the distinction with MFGs.
>
> **Weakness 3 (assumptions and applicability):**
> Though there are 8 assumptions in our paper, they are for different theorems, please see point (3) in our common reply. We want to add that:
> * **Assumptions 1-3** are used only in Theorem 2.4 and are common in MFG/MFC literature.
> * The **Assumption on $\gamma$** in Theorem 2.4 is used to obtain a rate of convergence (see also point (3) in the common reply).
> * **Assumptions 4-6** are used in Theorem 3.1 and are classical conditions for the convergence of the Nash Q-learning algorithm. See Hu & Wellman (2003) and Yang et al. (2018).
> * **Assumptions 7-8** (Lipschitz continuity for the mean-field functions) are used in Theorem 3.2. These conditions are used to derive a rate of convergence of the error between the discretized MFTG and its original continuous version.
>
> **Weakness 4 (Nash-Q analysis):**
> Thank you but it should be noted that our contribution is not the convergence of Nash Q-learning for the discretized problem but the analysis of the Q-function discretization error, which was not at all discussed in the paper of Hu and Wellman because they focused on discrete spaces. In line 353, we wrote: *"We omit the proof of Theorem 3.1 as it is essentially the same as in (Hu & Wellman, 2003). We then focus on the difference between the approximated Nash Q-function [...] and the true Nash Q-function"*. The second result **(Theorem 3.2) is our main contribution in this direction**. With Theorem 3.1, they give the full picture of convergence of Nash Q-learning for MFTG.
>
> **Weakness 5 (graphon games):**
> Thank you for pointing this out. We will include this reference and clarify the connection. However, there is one important difference: when the number of coalitions is finite, each coalition's actions have **a non-negligible influence on each of the other coalitions** (as in a finite-player game). This is why we consider **policies that take as input the population distributions**, which are high-dimensional continuous vectors. It is a crucial difference with graphon games, in which the number of coalitions is very large (or, in the limit, infinite), so that one can consider decentralized policies (i.e., policies which depend only on the individual agent's state).  In that case, the policy's input is low-dimensional and value functions are easier to approximate.

---

> > ### Author Response · Authors · 2024-11-18
> > **Reply to Reviewer iiZW (2/2: Questions)**
> >
> > **Question 1 (Assumption 3):**
> > Thank you for your question. We want to clarify that:
> > * Our result **still holds** if we consider a *smaller* class of policies because it would be a special case.
> > * For example, Theorem 2.4 would still be true if the policies were not allowed to depend on the mean-field distribution: indeed, this is a special case of Lipschitz policies, with Lipschitz constant $L_\pi = 0$.
> > * However, our motivation to consider population-dependent policies (even if the mean fields are deterministic) is because this is a **game**. When population 1 changes her policy, this will change her own population distribution, which will affect other populations. So unilateral deviations in the class of population-independent policies and in the class of population-dependent policies are different. In this work, **we treat the most general case** (population-dependent policies).
> >
> > **Question 2 (Theorem 2.4):** Thank you for raising this interesting technical question. We would like to clarify the following points:
> > * This condition $\gamma(1+L_\pi+L_p)<1$ helps to control the propagation of errors in future time steps and to obtain a **rate of convergence** (not just an asymptotic convergence like Saldi et al., see their Theorem 4.1).
> > * If $\pi$ and $p$ do not depend on the mean field, i.e., $L_\pi=L_p=0$, this condition reduces to $\gamma<1$ which is standard in infinite-horizon discounted MDPs.
> > * Please note that Saldi et al. also have a restrictive condition on the discount factor ($\beta$ in their notation), namely assumption (g) on page 4261 requires $\alpha\beta\gamma < 1,$ with $\alpha$ and $\gamma$ defined on the same page. Their $\gamma$ is greater than $1$ and $\alpha$ would probably be larger than $1$ too for some choices of models. In that case, $\alpha\gamma$ can be much larger than $1$, hence requiring the discount $\beta$ to be much smaller than $1$. Furthermore, Saldi et al. also have conditions on the moduli of continuity $w_p(r)$ and $w_c(r)$ of the transitions and the cost (see assumption (h) on page 4262 in their paper). While it is true that this is slightly less restrictive than Lischitz continuity, **their Theorem 4.1 only provides an asymptotic convergence and no convergence rate**.
> >
> > **Question 3 (Assumption 8):**
> > Thank you for bringing up this point. In general, it is not straightforward to deduce the Lipschitz continuity of the value function based on the Lipschitz continuity of the reward function and the transition probability. For single MFMDPs, Proposition 4.2 of (Motte & Pham, 2022) shows that the value function is Holder continuous (and 1-Holder continuous is Lipschitz continuous). However, the extension to MFTGs (i.e., multiple *competing* MFMDPs) is non trivial. We will add a remark along these lines.

---

> > > ### Comment · Reviewer_iiZW · 2024-11-25
> > >
> > > Thank you for the detailed response. My raised weaknesses and questions have been fully addressed. For the improved discussions and clarified theoretical assumptions / experimental results, I have increased my score.

---

### Official Review · Reviewer_ijet · 2024-11-03

**Soundness:** 1
**Presentation:** 2
**Contribution:** 1
**Rating:** 3
**Confidence:** 3

**Summary:**

This paper explores the application of mean field games (MFGs) to learn approximate ϵ-Nash equilibria. The authors present several theoretical findings. One key result (Theorem 2.4) shows that, with sufficient Lipschitz constraints on the Markov game properties, the approximation error for the finite MFG can be bounded by $O(|S| \sqrt{|A|} / \sqrt{N})$. Another result (Theorem 3.2) demonstrates that, under similar Lipschitz bounds, the $\epsilon$-approximation error can be bounded by a linear function of these constraints, achieved via Nash Q-learning. In addition to the theoretical contributions, the authors present empirical results using the Deep Deterministic Policy Gradient (DDPG) algorithm. They demonstrate that, across multiple settings, the DDPG algorithm converges to an approximate Nash equilibrium.

**Strengths:**

- The paper addresses a relevant and practical problem, and the proposed technology could be applied to real-world scenarios such as cybersecurity and economics.

- The authors conduct a comprehensive empirical study, showing that the DDPG algorithm performs well in converging to an approximate Nash equilibrium.

**Weaknesses:**

- The proofs of error bounds rely heavily on ensuring all components of the MFTG (reward, transition, policy, etc.) satisfy Lipschitz conditions. While this approach is valid, proving these conditions poses no significant technical challenge and may be considered borderline trivial.

- The authors claim to be the first to apply deep reinforcement learning (RL) to MFTGs, yet there are recent works (e.g., [1], [2]) that also explore similar applications. The paper does not adequately position its contributions relative to these advancements.

- There is a disconnect between the theoretical framework (based on Nash Q-learning) and the empirical results (focused on DDPG). The theoretical guarantees are established specifically for Nash Q-learning, but no similar results are provided for DDPG, resulting in a gap between the theory and the experiments.

**Questions:**

- In Section 2.3, the authors show an equivalence between mean-field games and Markov Decision Processes (MDPs). It is well known that MFGs can be represented as MDPs; were any new or useful features introduced in this reformulation to a mean-field MDP?

- In line 384, the authors suggest that solving the one-stage game is already intractable. Are we primarily interested in analyzing the sequential game outcomes, or does the focus remain on the single-stage game? This distinction is crucial, as the results between single stage game and sequential game may differ drastically.

---

> ### Comment · Reviewer_ijet · 2024-11-12
>
> Quick, note my apologies, the references I mentioned in my review were:
>
> [1] Jusup, Matej, et al. "Safe model-based multi-agent mean-field reinforcement learning." arXiv preprint arXiv:2306.17052 (2023).
>
> [2] Bordelon, Blake, and Cengiz Pehlevan. "Dynamics of finite width kernel and prediction fluctuations in mean field neural networks." Advances in Neural Information Processing Systems 36 (2024).

---

> > ### Author Response · Authors · 2024-11-18
> > **Reply to Reviewer ijet**
> >
> > We thank you for your comments. We hope that our answers below will clarify any confusion and will convince you that our work deserves a better score.
> >
> > **Weakness 1 (Lipschitz assumption):**
> > We respectfully disagree and would like to convince you:
> > - First, this type of assumptions is **very common**, please see point (3) in our common reply for some references. The paper by Jusup et al. that you cite also uses this type of assumption (see their Assumptions 1-3). We do not believe that all these papers can be considered *"border line trivial"*.
> > - Second, the proof of our Theorem 2.4 ($\epsilon$-Nash approximation) requires a careful analysis, particularly because of the **structure** of the problem (which is different from MFG and MFC, see point (1) in the common reply). Notice that our $\epsilon-$Nash result provides not only an asymptotic convergence but also a **rate**, which is not always the case for convergence results in other papers.
> >
> >
> > **Weakness 2 (other references):**
> > We would like to draw your attention to the fact that these two papers are **not related to MFTGs**. Jusup et al. wrote "*We focus on MFCs in this work*" and as mentioned in point (1) of the common reply, MFC is very different from MFTG. Bordelon et al. wrote "*We analyze the dynamics of finite width effects in wide but finite feature learning neural networks*" and their paper does not discuss games, Nash equilibria or reinforcement learning at all. This seems totally unrelated to what we do. If you disagree, could you please clarify how the methods described in these two papers can solve MFTGs like our methods do? If they cannot, we hope that you agree that our methods are novel and interesting, compared with the existing literature.
> >
> >
> > **Weakness 3 (theory vs numerics):** Thank you for raising this question. We want to clarify the following 3 points:
> > 1. We do have some numerical experiments with Nash-Q learning (Example 1 on page 9). It shows that this algorithm works well for small-scale problems.
> > 2. Let us complement what we wrote in Section 4 (particularly lines 395-403) about the motivation for the DRL algorithm and why a proof of convergence is outside of the scope of the present paper. To the best of our knowledge, there are **no existing proofs of convergence of DDPG** under general conditions. For example, the recent paper of Xiong et al., (2022) states in the conclusion: *"Convergence of more popular algorithms such as DDPG is also interesting to study."* It does not seem reasonable to blame every paper using DDPG for not having a proof.
> > 3. Even if such a proof existed, it would only be for a standard MDP, but here we need to compute a **Nash equilibrium**, which further complicates the analysis. This is why we focused on numerical aspects, proving that the method works.
> > 4. From our point of view, our work has the merit of showing that this DRL method works empirically, which motivates a theoretical study to be done in future works.
> >
> > **Question 1 (MFG, MFMDP and MFTG):**
> > * *"the authors show an equivalence between mean-field games and Markov Decision Processes (MDPs)"*: We are sorry if there is a confusion but this is not correct. As we wrote at the beginning of Sec. 2.3, we "*rephrase the MFTG in the framework of Markov decision processes (MDPs)*". But **MFTGs are not MDPs** so there is no *"equivalence"*. Please check our paper's introduction, as well as point (1) of our common reply.
> > * *"were any new or useful features introduced in this reformulation to a mean-field MDP"*: We provided some information in point (2) of our common reply: So far, MFMDPs have been studied only in themselves, as optimization problems, without competition. Here, we consider a **game** where each player has an MFMDP.
> >
> > **Question 2 (sequential vs stage-game):** We want to stress that:
> > * The overarching goal is to solve sequential (mean field type) games. Solving the stage-game is only used as an **intermediate step in Nash-Q learning** (line 9 in Algo. 1 of the Appendix).
> > * On line 384, we meant that solving the stage-game is feasible for problems with a small number of states and actions, but it becomes intractable when this number grows. We will clarify this sentence in the revision.
> > * This is actually the motivation for introducing our **second algorithm**, which **does not require solving any stage-game** and directly aims for the global-in-time Nash equilibrium (see Algo. 2 in the appendix: no stage-game is ever solved).
> >
> > We thank you again for the comments and we hope that our answers will clarify any confusion, convince you of the merits of our paper, and that you will consider raising your score.
> >
> > **References:**
> >
> > Xiong, Huaqing, Tengyu Xu, Lin Zhao, Yingbin Liang, and Wei Zhang. "Deterministic policy gradient: Convergence analysis." In Uncertainty in Artificial Intelligence, pp. 2159-2169. PMLR, 2022.

---

> > > ### Comment · Reviewer_ijet · 2024-11-25
> > >
> > > The authors have highlighted a significant difference in complexity between MFTG (the paper) and previous work on MFG and MFC. After revisiting both the revised document and the referenced works, I am still unclear on where the significant additional complexity lies. However, for the sake of this discussion, let's assume that MFTG does indeed introduce a higher level of complexity compared to MFG or MFC, and that the differences between this work and [1] and [2] (which were cited only as examples) are substantial. I am willing to accept the author's claim that their work is significantly different that previous work the field of multi-player MDPs and MFGs. Yet despite this, I believe there are other key issues that need further consideration.
> > >
> > > **Lipschitz:** I recognize that many papers, including those cited here, use Lipschitz assumptions to offer guarantees on convergence and convergence rates in multi-agent settings. I appreciate the authors' investigation into the time it takes to prove Theorem 2.4, and I have reviewed the proofs. While the proof arguments are well-structured and standard, I didn't notice any parts which require tremendous innovation. However, I don’t oppose the arguments presented, and I think additional clarification or elaboration might help here, to discern where the technical challenges actually lie and what innovations were used to overcome them.
> > >
> > > **Theoretical Framework:** The most pressing concern for me remains the disconnect between the theory for Nash Q learning and DDPG, as presented in the paper, and the empirical results. The authors argue that there is no proof available for DDPG and therefore no need to provide one, suggesting that the theoretical work in the DDPG direction is complex. While I understand this point, I feel it does not fully address the core issue. We should aim to tackle complex problems, rather than avoid them because of their difficulty. The central question remains: *Why present Nash Q learning theory, but present empirical results from DDPG?* This question has yet to be fully addressed, and I believe further clarification would be beneficial.
> > >
> > > **Sequential vs. Stage-game:** I respectfully disagree with the assertion that solving a single stage game is an intermediate step toward solving the sequential game. Solving a repeated single-stage game does not always provide insights for solving a similar sequential game, as is evident in cases toy examples the Prisoner's Dilemma. I believe this assumption needs more rigorous justification, and it would be helpful to elaborate on this further.
> > >
> > > **Motivation and Key Contributions:** Lastly, I believe the paper could benefit from clearer communication regarding its key contributions. The focus seems to be on incremental progress within a narrow framework, rather than addressing more complex challenges via a general framework. This is only my impression from reading the paper thus far, and I suggest that a stronger connection between the framework and its broader applications (i.e. specific scenarios in economics and/or engineering) to enhance the overall impact of the paper.

---

> > > > ### Author Response · Authors · 2024-11-26
> > > > **Reply to Reviewer ijet**
> > > >
> > > > Thank you for your further comments and for acknowledging the novelty of our work compared with the literature on MFG and MFC/MFMDP. For the other comments, we reply point by point below.
> > > >
> > > > **1. Lipschitz:**
> > > > - **Technical challenges:**
> > > >     - Compared with MFGs, the state is not a single-agent state but a **mean-field (MF) state** (probability distribution). This MF state is an input of the policy (not decentralized as usual in MFG). So we need to evaluate the propagation of error for the MF dynamics with MF-dependent policies.
> > > >     - Compared with MFCs/MFMDPs, the goal is not just to evaluate the sub-optimality of a policy but the **"sub-Nashness"**. In other words: only one coalition deviates to optimize her cost, not all the coalitions together. Coalitions impact each other, while in an MFC/MFMDP, all the agents are treated as one group.
> > > > - **Innovations:**
> > > >     - Conceptually, we find a way to combine ideas from the two literatures (we study Nash equilibria with MF states).
> > > >     - Technically: First, we use the **state-action distribution** (p.16-17). Note the problem's definition involves only the state distribution. Let us know if you know papers using this technique. Second, the error propagation in the MF evolution is studied with MF in both the transitions and the policies. We need to find a **suitable decomposition** of the error in a specific way in order to be able to use the LLN and obtain (3).
> > > >
> > > > We stress again that this result is very important for the MFTG literature: it **motivates MFTG computations**and it does so by providing a **rate of convergence**.
> > > >
> > > > **2. Theoretical Framework:**
> > > > - We believe that **both have their own merits** and deserve to be discussed in the paper. This work will motivate more studies on scalable RL for MFTGs.
> > > > - **Nash Q-learning** is one of the most famous RL methods for games. It is thus natural to adapt it for MFTGs. We analyze it and show its good performance on a small-scale game (Ex. 1).
> > > > - However, it suffers from a lack of scalability. It is then natural to wonder how to combine the principle of Nash Q-learning with deep RL. Instead, MARL methods typically try other approaches, directly building on single-agent RL methods. Following this trend, we propose a **DDPG-based method** for MFTG and study it empirically.
> > > > - In short our answer is: *Both algorithms have their own advantages, which justify their presence in the paper: Nash Q-learning enjoys a strong theoretical foundation and works in small-scale examples; DDPG-MFTG is highly scalable.*
> > > >
> > > > **3. Sequential vs. Stage-game:** We are sorry to see that there is a disagreement  *"with the assertion that solving a single stage game is an intermediate step toward solving the sequential game".*
> > > > In fact, the core idea of the paper of Hu and Wellman is precisely to **reduce the infinite horizon problem to solving a sequence of stage-games**. Of course these policies are **not just greedy policies** for the one-step reward function. The stage-game depends on the **NashQ function**, which is learnt along the way. See in particular Hu and Wellman's Lemma 10 (which shows a general connection between infinite horizon Nash equilibria policies and stage-game policies) and Theorem 17, which shows that the limited of the stage-game solutions constructed by the algorithm, you obtain the solution to the whole game. This is the foundation of the Nash Q-learning method. Please let us know if this is not clear and you would like more explanations.
> > > >
> > > > **4. Motivation and Key Contributions:** Thank you for the suggestion to clarify this. Space is very limited so we decided to focus on the technical content rather than general motivations. As you may have noticed, ICLR has a strict page limit and our paper already has a long appendix so we did not want to make it even longer. But we are happy to provide more explanations. To be clear:
> > > > - **Framework:** We tackle **MFTGs with general transitions and reward functions** (in finite space) in the sense that we do not assume that the transitions are independent of the mean field (as is common in the MFG literature for instance) or that some of the functions are linear or quadratic.
> > > > - **Applications:** We have given some references in the introduction (third paragraph). Our paper does not tackle specific applications and instead focuses on developing a **general method**. Furthermore, we provide concrete **examples in the numerical section**. These examples are modeling crowd motion (Ex. 1, 2 and 4), predator-prey dynamics (Ex. 3), and cybersecurity (Ex. 5).
> > > >
> > > > To summarize, *our paper makes the first step in the development DRL methods for MFTGs, opening the door to the  range of applications covered by MFTGs.*
> > > >
> > > > Please let us know if you would like any further clarifications or if there is anything else that could help convincing you of the merits of our contributions.

---

> > > > > ### Comment · Reviewer_ijet · 2024-12-03
> > > > >
> > > > > **Minor Point:** While the authors make a compelling effort to demonstrate the increased complexity of MFTG compared to MFG or MFC, I feel the argument could be strengthened, as both I and reviewer 34g4 remain unconvinced. Revisiting the problem setting and positioning of the paper with a more focused explanation may help clarify this point and reinforce the paper's contributions.
> > > > >
> > > > > **Minor Point:** Upon examining Figure 1, the results for Nash Q-learning in MFTG suggest that the exploitability of player 2 appears constant, potentially indicating that an approximate Nash Equilibrium (NE) has been achieved. However, this gap remains constant throughout the learning phase, which raises concerns about convergence (It might also be worth clarifying whether this refers to coalition 1 and coalition 2, rather than player 1 and player 2.) Improving the empirical demonstration of learning behaviour for MFTG-NashQ could address these concerns and strengthen the results.
> > > > >
> > > > > **Minor Point:** Furthermore, I recommend that the authors put more effort into clarifying and proving NE or approximate NE of the game they are solving. For games with many NE, learning an approximate NE may be easier, but for games with unique, non-trivial NE, the problem could be combinatorially NP-hard, even to find an approximation. In the case of multiple NE's some solutions may be more desirable than others. Demonstrating the efficacy of their algorithm in these challenging environments would considerably strengthen their argument greatly. The MFG is a robust formulation, and solution concept, and it does have tangible application areas (for example as I've cited in [1]), however, this paper provides only very minimal evidence to support this.
> > > > >
> > > > > **Major Point:** The gap between DDPG and the theoretical results remains an area for improvement. While the DDPG-MFTG results are much stronger empirically, the theoretical underpinnings to support these findings are missing. The rebuttal and revisions have not sufficiently addressed this gap. I respectfully disagree with the authors, that theoretical convergence results for DDPG lies outside the scope of this paper, in fact, I believe the opposite. I believe that providing theoretical convergence results for DDPG is not only within scope but central to the value of this paper, as it holds significant relevance for the ML community. This addition would greatly enhance the paper's contributions and impact.
> > > > >
> > > > > Overall, the paper shows promise, particularly with the empirical results for DDPG-MFTG, and with some additional theoretical and empirical clarifications, it has the potential to make a valuable contribution to the field. Nevertheless, I am still somewhat reserved about the issues regarding the positioning of the paper, the rigour of equilibrium analysis, and the theory gap for DDPG-MFTG.

---

> ### Author Response · Authors · 2024-12-04
> **Reply to Reviewer ijet**
>
> Thank you for taking the time to reply. We still want to answer point by point, in case you, other reviewers or the Area Chair find these answers useful.
>
> **Minor Point 1**: The argument that MFTG is very different from MFG and MFMDP is standard in the literature on this topic. We tried our best to convince you, including by citing other works (e.g., by Tembine). Given the page limit, we cannot reproduce in our own paper the whole literature. The readers who want to know more are supposed to refer to the **references we cited**.
>
> **Minor Point 2 (Figure 1):** We are sorry but you are mistaken. We did **not** plot the exploitability of player 2. So there is no basis to say that *"the exploitability of player 2 appears constant"*. Please check again the figure: We plotted the **average exploitabilities** for each method (second to the left), and for our method it goes towards 0. You might have mistaken the **rewards** (left plot) for the exploitabilities. But even so, the reward of player 2 is not constant (while the reward of player 1 converges well), so we cannot understand what you meant. Also, we have explained the correspondence between "coalition" and "(central) player" e.g., in lines 101-102.
>
> **Minor Point 3 (NE):** Here again, we are sorry but the expression *"proving NE"* alone does not make sense. Maybe you forgot one word? Please note that we have proved **convergence of our Nash Q-learning algorithm to an approximate NE** (see our Theorems 3.1 and 3.2). This answers your question. Furthermore, we are not sure what you mean by *"very minimal evidence to support this"*. What does *"this"* refer to? As we have pointed out, MFTGs are different from MFGs. We have provided several references to motivate MFTGs and our Theorem 2.4 shows that MFTGs are a valuable approximation for applications to finite-player problems, which arise in practice.
>
>
> **Major Point (DDPG)** We would like to react to your comment *"theoretical convergence results for DDPG lies outside the scope of this paper"*.  According to Google Scholar, the DDPG paper [Lillicrap et al., 2016] is cited more than 17000 times. To the best of our knowledge a proof of convergence is still lacking, and the recent paper [Xiong et al., 2022] (see our earlier response) also mentions this. The same goes for many deep RL algorithms, which lack rigorous and generic proofs of convergence. Your comment seems to imply that all these works should be rejected. We disagree.

---

### Official Review · Reviewer_34g4 · 2024-11-05

**Soundness:** 3
**Presentation:** 4
**Contribution:** 2
**Rating:** 3
**Confidence:** 3

**Summary:**

This paper proposes a mean field type game (MFTG) with a finite number of noncooperative coalitions in which each coalition consists of a continuum of cooperative agents. Learning algorithms are proposed to solve approximate Nash equilibria in finite-size coalition games. Convergence and stability analysis are provided. Extensive numerical experiments are conducted.

**Strengths:**

Strength:
The paper is easy to follow with a clear logic from model formulation, equilibrium definition, algorithm implementation, to numerical experiments.

**Weaknesses:**

The novelty remains unclear. The authors are suggested to motivate readers in terms of what fundamental challenges an MFTG brings forth, of which learning algorithms and properties differ significantly from existing MFG and mean field control (MFC). The proposed two learning methods are incremental to the existing learning methods used for MFG. Reformulation with mean field MDPs is not new either.

Overall, this paper is more of a combination of existing concepts and methods for a mean field type game, rather than making fundamental contributions to the field of MFG or MFC.

**Questions:**

Could the authors provide more clarification in terms of why a finite space algorithm needs to presented first in Sec. 3 while having a DRL for Sec. 4? Using DRL for learning MFGs with large state and action spaces is widely used and well developed in MFG.

---

> ### Author Response · Authors · 2024-11-18
> **Reply to Reviewer 34g4**
>
> We are grateful for your comments. It seems that there are some confusions about what our paper is doing. We hope that our answers below will convince you that our paper deserves a better score.  If you would like more details, could you please provide specific references?
>
> **Weakness (Novelty):**  Thank you for giving us the opportunity to clarify this point:
> 1. *"incremental to the existing learning methods used for MFG."*: As explained in point **(1)** of the **common reply**, MFTG and MFG are very different. We propose the first DRL algorithm for MFTG. If you believe that an existing algorithm of the MFG literature can be applied to MFTG, could you please **provide a specific reference**? We will be glad to provide more explanations.
> 2. *"Reformulation with mean field MDPs is not new either."*: While MFMDPs are not new (we mentioned this on line 196), previous works only considered a **single MFMDP** while we study a **game** between MFMDP players. Please also check point **(2)** in our **common reply**. If this is not clear, please let us know.
> 3. *"Combination of existing methods and concepts"*: We respectfully disagree with the reviewer: in our paper, we clearly cite prior work but we go **beyond** these works to solve a **new type** of problems. As far as we know, there are no existing methods to solve MFTGs like the ones we solve.
>
> **Question 1 (finite space/DRL):**  *"why a finite space algorithm needs be to presented first in Sec. 3 while having a DRL for Sec. 4"*:
> - In an MFTG, the central player of each coalition has a value function and a policy. As explained in Section 2.4 (see in particular line 229), these are **functions of the population distributions**, which are represented as vectors (of dimension the number of possible states). So the value functions we want to compute are naturally taking high-dimensional, continuous inputs. This is why we decided to employ DRL.
> - As explained on page 8 of our initial submission the main advantage of this method compared with tabular RL *"is that it does not require discretizing the simplexes. The state and action distributions are represented as vectors (containing the probability mass functions) and passed as inputs to neural networks for the policies and the value function"*.
> - Please note that this actually a **key difference with RL for MFGs**, which usually focuses on decentralized value functions and policies that are functions of the individual state only and not functions of population distribution (see e.g. the references on line 60 of our paper:  Subramanian & Mahajan, 2019; Guo et al., 2019; Elie et al., 2020; Cui & Koeppl, 2021). In that case, tabular methods can be used when the state space is finite. This does not hold true in our case due to the presence of the mean-field as an input: even when the state space is finite, the mean field input is continuous.
>
> **Question 2 (difference with MFG):** *"Using DRL for learning MFGs with large state and action spaces is widely used and well developed in MFG"*:
> - First, we are **solving MFTGs and not MFGs**. Please see our answers to the above Weakness and Question 1.
> - Second, as far as we know, there are only a few papers on DRL for MFGs and they are usually restricted to very specific settings. In any case, these papers would not help to solve our MFTGs.
> - We hope that our answers help clarifying these points. But if you would like further clarifications, please provide specific references of DRL for MFGs so we can explain why these methods do not jeopardize our work.

---

> > ### Comment · Reviewer_34g4 · 2024-11-28
> >
> > I appreciate the authors’ efforts to clarify the novelty and explanation of details. However, I am still not convinced why MFTG is significantly more challenging than either MFG or MFC. The contributions of this paper are not clear either, primarily because of lack of sufficient theoretical justification of the solutions and algorithms. What are the key fundamental contributions of this paper? Does it lie in the modeling of MFTG using MFMDP or more in the learning algorithms? If the former, what is the existence and uniqueness of the Nash equilibrium. If the latter, what are the properties of the learning algorithms, such as the convergence rate and error of DDPG?

---

> > > ### Author Response · Authors · 2024-12-02
> > > **Reply**
> > >
> > > Thank you very much for taking the time to share your thoughts and questions. We hope that your concerns have already been addressed in earlier responses to your comments and to all the reviewers. But we want to take this chance to address your latest questions in detail.
> > >
> > > **1. "Why MFTG is significantly more challenging than either MFG or MFC":**
> > > - As we wrote on pages 1 and 2: *"MFTGs are different from MFGs because the agents are cooperative within coalitions, while MFGs are about purely non-cooperative agents. They are also different from MFC problems, in which the agents are purely cooperative."*  Solving an MFTG is significantly more complex as we need to consider a non-cooperative game where each central player has an MFMDP. So it combines both MFGs and MFMDPs.
> > >  -  In particular for MFG: (Tembine, 2017) wrote on page 706-707 *"In contrast to mean-field games [2, 3] in which a single player does not influence of the mean-field terms, here, it is totally diﬀerent. In mean-field-type games, a single player may have a strong impact on the mean-field terms."* Actually this is one of the major challenges that we tackle in our work and which justifies using **population-dependent policies**.
> > > -  For the comparison with MFMDPs, we have already answered earlier, see for instance point (2) in the common reply.
> > >
> > > **2. "The contributions of this paper are not clear either, primarily because of lack of sufficient theoretical justification of the solutions and algorithms":**
> > > The main contributions are stated in our paper, lines 78-88 in the revised PDF. To address your specific questions:
> > >
> > > - *"Does it lie in the modeling of MFTG using MFMDP or more in the learning algorithms?"*
> > > We study **both theoretical and numerical** aspects of MFTGs. From our point of view, the contributions are **complementary**. For the theoretical part, we first present a rigorous mathematical formulation of MFTGs and study the $\epsilon$-Nash approximation. We prove the theoretical convergence of Nash Q-learning and provide an error analysis. Numerically, we present two main algorithms to solve several MFTGs with general environments and rewards, one based on Nash Q-learning and the other based on DDPG.
> > >
> > > - *"What is the existence and uniqueness of the Nash equilibrium"*
> > > Existence holds under general conditions by following the literature on finite-player games, see e.g. [1]. Uniqueness is in general not guaranteed. Thank you for raising this point, we will add a comment about it in the final version.
> > >
> > > - *"What are the properties of the learning algorithms, such as the convergence rate and error of DDPG?"*
> > > While the convergence of DDPG is an interesting problem, it lies outside the scope of this paper. As far as DDPG is concerned, we focus on showing that the algorithm scales well and is able to solve complex MFTGs. We provided extensive numerical experiments. We hope that this numerical study will motivate follow-up works to study the theoretical convergence.
> > >
> > > Reference:
> > >
> > > [1] Fink, A. M. (1964). Equilibrium in a stochastic $n$-person game. Journal of science of the hiroshima university, series ai (mathematics), 28(1), 89-93.

---

### Author Response · Authors · 2024-11-18
**Common reply**

We thank all reviewers for their efforts and reviews. Their comments help increasing the quality of our paper. We are grateful for their acknowledgement that the paper is "easy to follow with a clear logic" and that our work "addresses a relevant and practical problem". We will soon upload a revised version of the paper with all the changes described in our responses, including new numerical results for a variant of Example 1. In the meantime, we provide detailed answers to all the questions and explain the changes we are making in the paper.

Before giving answers for each reviewer, we want to provide a common reply to a clarify a few important points.

**(1) MFTGs are not MFG nor MFC:** Existing algorithms for mean field games (MFG) and mean field control (MFC) **cannot be used** for the type of problems that our paper studies, namely, mean field type games (MFTGs). Although these topics share some key words, there are important differences. As we wrote on pages 1 and 2: *"MFTGs are different from MFGs because the agents are cooperative within coalitions, while MFGs are about purely non-cooperative agents. They are also different from MFC problems, in which the agents are purely cooperative. As a consequence, computational methods and learning algorithms for MFGs and MFC problems cannot be applied to compute Nash equilibria between mean-field coalitions in MFTGs"*. If the reviewers think that existing algorithms can solve our MFTGs, we encourage them to provide specific references from the literature so we can provide detailed explanations.


**(2) MFTGs = Nash equilibrium between MFMDPs:** Solving an MFTG is significantly more complex than solving a mean field MDP (MFMDP). Indeed, the concept of MFMDP has been introduced for discrete time MFC [Motte & Pham, 2022], [Carmona et al., 2023], which is a *fully cooperative* population. In this paper, we consider a **non-cooperative** game where each central player has an **MFMDP**. Our paper is to MFMDP what Markov games are to plain MDPs. Solving Nash equilibria is much more complex than solving MDPs; see for instance (Daskalakis et al., 2023).

**(3) Assumptions and theoretical results:**  In this paper, we assume that the environment satisfies certain Lipschitz conditions. These **assumptions are frequently used** in MFGs and MFC see e.g. (Anahtarci et al., 2023), (Cosso et al., 2019), (Guan et al., 2024). Notice that our theoretical results provide not only convergence but **rates of convergence** (both for $\epsilon$-Nash approximation and error analysis of Nash Q-learning), so we believe it is worth making these assumptions. It should also be noted that we **do not require all the assumptions at the same time**: we have 3 groups of assumptions, namely Assumptions 1-3, 2-5 and 6-7, which are used respectively in Theorems 2.4, 3.1 and 3.2.

In our replies below, we provide extra references when needed at the end of each reply. Other references are the same as in our paper.

**References:**

Anahtarci, Berkay, Can Deha Kariksiz, and Naci Saldi. "Q-learning in regularized mean-field games." Dynamic Games and Applications 13.1 (2023): 89-117.

Cosso, Andrea, and Huyên Pham. "Zero-sum stochastic differential games of generalized McKean–Vlasov type." Journal de Mathématiques Pures et Appliquées 129 (2019): 180-212.

Daskalakis, Constantinos, Noah Golowich, and Kaiqing Zhang. "The complexity of markov equilibrium in stochastic games." In The Thirty Sixth Annual Conference on Learning Theory, pp. 4180-4234. PMLR, 2023.

Guan, Yue, Mohammad Afshari, and Panagiotis Tsiotras. "Zero-Sum Games between Mean-Field Teams: Reachability-Based Analysis under Mean-Field Sharing." Proceedings of the AAAI Conference on Artificial Intelligence. Vol. 38. No. 9. 2024.

---

> ### Author Response · Authors · 2024-11-23
> **Revised PDF**
>
> We would like to thank again the reviewers for all their work and detailed comments, which greatly helped improve the quality of the paper. We have uploaded a **revised version** of the PDF. We hope that this new version will convince the reviewers to increase their score. If there are still questions or confusions, we will be glad to answer any follow-up questions.
>
> For convenience, we have highlighted the main changes in blue in the revised text. Furthermore, we summarized below the main changes:
>
> - Section 1:
>     - Corrected the initial MFG references (Lasry & Lions, 2007; Huang et al. 2006).
>     - Added references to graphon games and mixed mean field control games, with infinitely many mean-field groups, and explained why methods for such games cannot be used for MFTGs.
> - Section 2:
>     - Section 2.1 (and following): Changed $N_1,\dots,N_m$ to $\bar{N}$ in the superscripts for the populations.
>     - Section 2.2: Corrected typo in the former Assumption 3. We have also provided references for former Assumptions 1-3 (new Assumption 1).
>     - Assumptions: **Regrouped respectively Assumptions 1-3, 4-6 and 7-8 as the new Assumptions 1, 2 and 3.** They are used for different purposes so we do not need all of them at the same time.
>     - Section 2.2: Added a comment about a comparison with (Saldi et al., 2018) to explain that we provide a **rate of convergence** and not just asymptotic convergence (which justifies our assumptions are slightly stronger).
>     - Section 2.3: Clarified that we study **games between MFMDPs** and not simple MFMDPs.
> - Section 3: Clarified the connection between our assumptions and **existing assumptions or results in the literature.**
> - Section 4: Clarified the **two advantages of the DRL algorithm** over Nash Q-learning: no need to discretization of the simplexes and no need to solve any stage-game.
> - Section 5:
>     - Modified the **reward function of Example 1** (line 435 and then lines 1369 to 1371) to include **bi-directional interactions** between the two populations. Replotted Figures 1 and 2 with the new results (we still observe that our method does better than the baseline).
>     - Enlarged font size of most figures to make them easier to read.
>     - Merged the captions of side-by-side figures for better formatting and better space usage.
>     - Added more information in the captions (rewards and exploitabilities are averaged and colorbars are for densities).
>     - Corrected figure references.
> - Section 6:
>     - Added a paragraph on **related works**.
>     - Clarified the reproducibility statement to show that we have included a large amount of details in the appendices.
>
> To accommodate for these changes, we had to make minor adjustments to how some of the equations are displayed but we hope the reviewers will understand that we tried to our best to take their comments into consideration while respecting the page limit.

---

### Meta-Review · Area_Chair_kYEf · 2024-12-22

**Metareview:**

The paper explores reinforcement learning methods for mean-field type games (MFTGs), a setting where cooperative agents within coalitions compete non-cooperatively against other coalitions. The authors propose a Nash Q-learning algorithm with theoretical guarantees and a deep RL-based algorithm which appears to perform favorably.

A major concern that persisted throughout the discussion is that the paper’s contributions were seen as incremental relative to prior work.

In the words of Reviewer 34g4, “the novelty remains unclear” and that the work is “more of a combination of existing concepts and methods for a mean field type game, rather than making fundamental contributions.” Reviewer iiZW agreed (“the model and NashQ-based analysis are not highly novel”)

While the authors clarified that MFTGs differ significantly from MFGs and MFC, some reviewers remained unconvinced.

Reviewer iiZW, while positive on the paper, mentioned that the "The work fuses existing ideas from MFGs with algorithms from both game theory and deep RL". This gave credence to the fact that---while MFTGs are different from MFGs---the paper might be an application of well-understood ideas. I asked the negative reviewers to please comment on whether the authors' response had changed their mind after the discussion, but they mentioned being still convinced the paper was just lacking in novelty.

Lack of novelty per se need not be an insurmountable barrier: papers do not need to invent the wheel to be useful. So, the next question that was assessed was whether the problem tackles an important setting. Reviewer iiZW seemed the only to touch on this. While overall appreciative of the paper, they mentioned that "the setting sounds a bit niche to me. I wonder if one could discuss more the significance of the setting [..]".  So, unfortunately, this is not useful in dismissing the incrementality worry of the negative reviewers.

Overall, my impression is that the above concern with incrementality is the only real issue of the contribution (a concern of perceived disconnect between the theoretical results and experiments, by Reviewer ijet, was dismissed in the decision as not substantive enough).

I think the paper would benefit from addressing the above by clarifying the technical novelty.

**Additional Comments On Reviewer Discussion:**

In addition to the above account, another concern that was raised during the discussion was the perceived disconnect between the theoretical results and the experiments (Reviewer ijet). The authors responded that proving convergence for DDPG is a challenging open problem in the literature, but the reviewer remained unconvinced. I think the position of the reviewer might have been too intransigent on this one, and this concern was discounted.

---

### Decision · Program_Chairs · 2025-01-22

Reject